# REX: REVERSIBLE SOLVERS FOR DIFFUSION MODELS

## ABSTRACT

Diffusion models have quickly become the state-of-the-art for numerous generation tasks across many different applications. Encoding samples from the data distribution back into the model's underlying prior distribution, often called the *inversion* of diffusion models, is an important task that arises from many downstream applications. Prior approaches for solving this task, however, are often simple heuristic solvers that come with several drawbacks in practice. In this work, we propose a new family of solvers for diffusion models by exploiting the connection between this task and the broader study of *algebraically reversible* solvers for differential equations. In particular, we construct a family of reversible solvers using an application of Lawson methods to construct exponential Runge-Kutta methods for the diffusion models; we call this family of reversible exponential solvers *Rex*. In addition to a rigorous theoretical analysis of the proposed solvers, we also demonstrate the utility of the methods through a variety of empirical illustrations.

## 1 INTRODUCTION

Diffusion models have quickly become the state-of-the-art in generation tasks across many varied modalities from images (Rombach et al., 2022) and video (Blattmann et al., 2023) to protein generation (Skreta et al., 2025b) and biometrics (Blasingame & Liu, 2024d). The sampling process of diffusion models is done through numerically solving an Itô *stochastic differential equation* (SDE) or related *ordinary differential equation* (ODE) which describes the evolution of a sample drawn for some prior noise distribution to the data distribution. Inversion of the sampling procedure, *i.e.*, constructing a bijective map from the data distribution back to the prior distribution, is invaluable for many downstream applications.

While the true (stochastic) flow maps of diffusion models do provide such a bijection, in practice we need to solve such models numerically, thereby incurring truncation errors breaking the bijection. Thus to obtain the *exact inversion* of a diffusion model we are looking for a scheme which is algebraically reversible. *I.e.*, we would like a numerical scheme which enables us to move between the data and prior distribution without any reconstruction errors. Recently, several works have explored solving this problem for the probability flow ODE, namely, EDICT (Wallace et al., 2023), BDIA (Zhang et al., 2024), and BELM (Wang et al., 2024).

However, designing such inversion methods is very tricky, as such solvers are plagued by issues of low order of convergence, lack of stability, amongst other undesirable properties; moreover, it is even more difficult to construct such schemes for SDEs. To the best of our knowledge there does not currently exist a scheme for exact inversion for diffusion SDEs *without* storing the entire trajectory of the Brownian motion in memory *à la* Wu & la Torre (2023) which is trivially reversible, but not the type of reversibility we are interested with.

To address these issues we propose *Rex*, a family of reversible solvers for diffusion models which can

1. Work for both the probability flow ODE and reverse-time SDE with both data and noise prediction parameterizations,

2. Obtain an arbitrarily high order of convergence (in the ODE case), and

3. Exactly invert a diffusion SDE *without* storing the entire realization Brownian motion in memory.

## 2 PRELIMINARIES

**Diffusion models.** Diffusion models (Sohl-Dickstein et al., 2015; Ho et al., 2020; Song et al., 2021a;b) have quickly become one of the most popular paradigms for constructing *generative models*. Consider the following Itô *stochastic differential equation* (SDE) defined on time interval $[0, T]$:

$$\mathrm{d}\boldsymbol{X}_t = f(t)\boldsymbol{X}_t \, \mathrm{d}t + g(t) \, \mathrm{d}\boldsymbol{W}_t, \tag{1}$$

where $f, g \in \mathcal{C}^\infty([0, T])$[1] form the drift and diffusion coefficients of the SDE and where $\{\boldsymbol{W}_t\}_{t \in [0,T]}$ is the standard Brownian motion on the time interval. The coefficients $f, g$ are chosen such that the SDE maps clean samples from the data distribution $\boldsymbol{X}_0 \sim q(\boldsymbol{X})$ at time 0 to an isotropic Gaussian at time $T$. More specifically, for a *noise schedule* $\alpha_t, \sigma_t \in \mathcal{C}^\infty([0, T]; \mathbb{R}_{\geq 0})$ consisting of a strictly monotonically decreasing function $\alpha_t$ and strictly monotonically increasing function $\sigma_t$, the drift and diffusion coefficients are found to be

$$f(t) = \frac{\dot{\alpha}_t}{\alpha_t}, \qquad g^2(t) = \dot{\sigma}_t^2 - 2\frac{\dot{\alpha}_t}{\alpha_t}\sigma_t^2, \tag{2}$$

where with abuse of notation $\dot{\sigma}_t^2$ denotes the time derivative of the function $\sigma_t^2$ (Lu et al., 2022b; Kingma et al., 2021)—this ensures that $\boldsymbol{X}_t \sim \mathcal{N}(\alpha_t\boldsymbol{X}_0, \sigma_t^2\boldsymbol{I})$. However, we wish to map from *noise* back to *data*, as such we employ the result of Anderson (1982) to construct the *reverse-time* diffusion SDE of Equation (1), which is found to be

$$\mathrm{d}\boldsymbol{X}_t = [f(t)\boldsymbol{X}_t - g^2(t)\nabla_{\boldsymbol{x}} \log p_t(\boldsymbol{X}_t)] \, \mathrm{d}t + g(t) \, \mathrm{d}\overline{\boldsymbol{W}}_t, \tag{3}$$

where $\mathrm{d}t$ is a *negative* timestep, $\{\overline{\boldsymbol{W}}_t\}_{t \in [0,T]}$ is the standard Brownian motion in reverse-time, and $p_t(\boldsymbol{x}) := p(t, \boldsymbol{x})$ is the marginal density function. Then, if we can learn the *score function* $(t, \boldsymbol{x}) \mapsto \nabla_{\boldsymbol{x}} \log p_t(\boldsymbol{x})$ (Song et al., 2021b)—or some other *equivalent* reparameterization, *e.g.*, noise prediction (Song et al., 2021a; Ho et al., 2020) or data prediction (Kingma et al., 2021)—we can then draw samples from our data distribution $q(\boldsymbol{X})$ by first sampling some $\boldsymbol{X}_T \sim p(\boldsymbol{X})$ from the Gaussian prior and then employing a numerical SDE solver, *e.g.*, Euler-Maruyama, to solve Equation (3) in reverse-time. Notably, through careful massaging of the Fokker-Planck-Kolomogorov equation for the marginal density, one can construct an ODE which is equivalent in *distribution* to Equation (3) (Song et al., 2021b; Maoutsa et al., 2020), yielding the *highly* popular *probability flow ODE*

$$\frac{\mathrm{d}\boldsymbol{x}_t}{\mathrm{d}t} = f(t)\boldsymbol{x}_t - \frac{g^2(t)}{2}\nabla_{\boldsymbol{x}} \log p_t(\boldsymbol{x}_t). \tag{4}$$

**Reversible solvers for neural differential equations.** Recently, researchers studying *neural differential equations* have begun to propose several *algebraically reversible solvers* as an alternative to both traditional discretize-then-optimize and optimize-then-discretize (the continuous adjoint equations) (Kidger, 2022, Chapters 5.1 & 5.2) which are used to perform backpropagation through the neural differential quation. Consider some prototypical neural ODE of the form $\dot{\boldsymbol{x}}_t = \boldsymbol{u}_\theta(t, \boldsymbol{x}_t)$ with vector field $\boldsymbol{u}_\theta \in \mathcal{C}^r(\mathbb{R} \times \mathbb{R}^d; \mathbb{R}^d)$ which satisfies the usual regularity conditions. Then consider a single-step numerical scheme of the form $\boldsymbol{x}_{n+1} = \boldsymbol{x}_n + \boldsymbol{\Phi}_h(t_n, \boldsymbol{x}_n, \boldsymbol{u}_\theta)$. Every numerical scheme $\boldsymbol{\Phi}$ is reversible in the sense that we can rewrite the forward step as an implicit scheme of the form $\boldsymbol{x}_n = \boldsymbol{x}_{n+1} - \boldsymbol{\Phi}_h(t_n, \boldsymbol{x}_n, \boldsymbol{u}_\theta)$; however, this requires fixed point iteration[2] and is both *approximate* and computationally *expensive*. This type of reversibility is known as *analytic reversibility* within the neural differential equations community (Kidger, 2022, Section 5.3.2.1). What we would prefer, however, is a form of reversibility that can be expressed in *closed-form*.

Beyond symplectic solvers (Vogelaere, 1956) which are trivially reversible[3], several algebraically reversible solvers have been proposed in light of the large popularity of neural ODEs. Namely, the following methods have been proposed: the *asynchronous leapfrog method* (Mutze, 2013; Zhuang et al., 2021), *reversible Heun method* (Kidger et al., 2021), and *McCallum-Foster method* (McCallum & Foster, 2024). The last of these is of particular interest to us, as it is the *only* algebraically reversible ODE solver to have a non-trivially region of stability and arbitrarily high convergence order. As McCallum & Foster (2024) simply refer to their method as *reversible X* where *X* is the underlying single-step solver, we opt to refer to their method as the *McCallum-Foster method* which we summarize below in Definition 2.1.

---

[1]We let $\mathcal{C}^r(X; Y)$ denote the class of $r$-th differentiable functions from $X$ to $Y$. If $Y$ is omitted then $Y = \mathbb{R}$.

[2]If the step size $h$ is small enough.

[3]Due to symplectic integrators being developed for solving Hamiltonian systems, they are intrinsically reversible by construction (Greydanus et al., 2019).

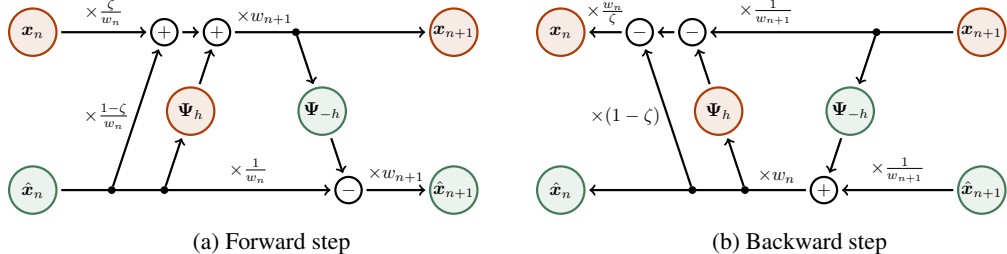

(a) Forward step            (b) Backward step

Figure 1: The computation graph of the Rex solver. Here $\boldsymbol{\Psi}_h$ denotes an exponentially weighted Runge-Kutta scheme (*cf*. Section 3.1) or exponential stochastic Runge-Kutta scheme (*cf*. Section 3.2), $\zeta \in (0, 1)$ is a coupling parameter, and $\{w_n\}_{n=1}^N$ denotes the set of weighting variables derived from the exponential schemes. The particular values of $w_n$ are discussed in Proposition 3.3. The visualization of the computation graph is inspired by McCallum & Foster (2024, Figure 2).

**Definition 2.1** (McCallum-Foster method). Initialize $\hat{\boldsymbol{x}}_0 = \boldsymbol{x}_0$ and let $\zeta \in (0, 1]$. Consider a step size of $h$, then a forward step of the McCallum-Foster method is defined as

$$
\begin{aligned}
\boldsymbol{x}_{n+1} &= \zeta \boldsymbol{x}_n + (1 - \zeta)\hat{\boldsymbol{x}}_n + \boldsymbol{\Phi}_h(t_n, \hat{\boldsymbol{x}}_n), \\
\hat{\boldsymbol{x}}_{n+1} &= \hat{\boldsymbol{x}}_n - \boldsymbol{\Phi}_{-h}(t_{n+1}, \boldsymbol{x}_{n+1}),
\end{aligned}
\tag{5}
$$

and the backward step is given as

$$
\begin{aligned}
\hat{\boldsymbol{x}}_n &= \hat{\boldsymbol{x}}_{n+1} + \boldsymbol{\Phi}_{-h}(t_{n+1}, \boldsymbol{x}_{n+1}), \\
\boldsymbol{x}_n &= \zeta^{-1}\boldsymbol{x}_{n+1} + (1 - \zeta^{-1})\hat{\boldsymbol{x}}_n - \zeta^{-1}\boldsymbol{\Phi}_h(t_n, \hat{\boldsymbol{x}}_n).
\end{aligned}
\tag{6}
$$

## 3 REX

In this section we introduce the *Rex* family of reversible solvers for diffusion models. Whilst one could straightforwardly apply a pre-existing reversible solver like asynchronous leapfrog, reversible Heun, or the McCallum-Foster method directly to the probability flow ODE in Equation (4), there are several reasons to consider an alternative approach. Stepping back from reversible solvers for a moment, we consider the broader literature of constructing numerical schemes for diffusion models. It is well known that we can exploit the structure of the drift and diffusion coefficients, *i.e.*, $f(t)$ and $g(t)$, to remove the discretization error from the linear term and transform the stiff ODE into a non-stiff form (Lu et al., 2022b; Zhang & Chen, 2023); a similar idea also holds for the reverse-time-diffusion SDE (see Lu et al., 2022a; Gonzalez et al., 2024; Blasingame & Liu, 2024a). Moreover, recall that the definitions of the drift and diffusion coefficients contain the time derivatives of the noise schedule ($\alpha_t, \sigma_t$), this structure enables us to greatly simplify the ODE/SDE and express a number of terms in closed-form again reducing approximation errors.

In Figure 1 we present an overview of the Rex computational graph. *N.B.*, the graph for both the ODE and SDE formulations are identical with the only difference being the weighting terms $\{w_n\}$ and the underlying numerical scheme $\boldsymbol{\Psi}_h$. The rest of this section is organized as follows: first we discuss applying the exponential integrators to the probability flow ODEs (see Section 3.1), then the reverse-time SDEs (see Section 3.2), and lastly we present the general Rex scheme (see Section 3.3).

### 3.1 PROBABILITY-FLOW ODE

Before constructing Rex we must first discuss the construction of $\boldsymbol{\Psi}_h$ from $\boldsymbol{\Phi}_h$ and how to derive the reparameterized ODE, *i.e.*, step 1 in Figure 2. In this section we review how to reparameterize the ODE in Equation (4) into this more convenient form.

**Generalized nomenclature for data and noise prediction models.** As alluded to earlier, there exist two popular reparameterizations of the score function which are used widely in practice, namely the noise prediction (Ho et al., 2020) and data prediction (Kingma et al., 2021) formulations. Following the conventions of Lipman et al. (2024) we write noise prediction model as $\boldsymbol{x}_{T|t}(\boldsymbol{x}) = \mathbb{E}[\boldsymbol{X}_T|\boldsymbol{X}_t =$

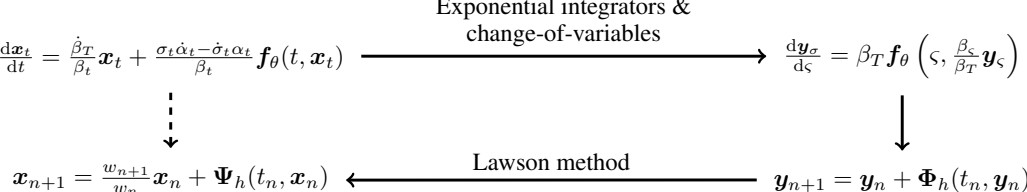

Figure 2: Overview of the construction of $\boldsymbol{\Psi}$ for the probability flow ODE from an underlying Runge-Kutta scheme $\boldsymbol{\Phi}$ for the reparameterized ODE in Equation (8). The parameters $\beta_t$ and $\varsigma_t$ are chosen to suit the data or noise prediction parameterizations (*cf*. Section 3.1). The graph holds for the SDE case *mutatis mutandis*.

$\boldsymbol{x}$] and write data prediction model as $\boldsymbol{x}_{0|t}(\boldsymbol{x}) = \mathbb{E}[\boldsymbol{X}_0|\boldsymbol{X}_t = \boldsymbol{x}]$. In this work we consider *both* a trained noise and data prediction model which we will denote generally by the neural network $\boldsymbol{f}_\theta(t, \boldsymbol{x})$. Additionally, we place the usual regularity constraints (*cf*. Lu et al., 2022b, Appendix B.1) on the model to ensure the existence and uniqueness of the ODE/SDE solutions. It is well known (Blasingame & Liu, 2025, Equation (19)) that the ODE in Equation (4) can be rewritten as

$$\frac{\mathrm{d}\boldsymbol{x}_t}{\mathrm{d}t} = \frac{\dot{\beta}_t}{\beta_t}\boldsymbol{x}_t + \frac{\sigma_t\dot{\alpha}_t - \dot{\sigma}_t\alpha_t}{\beta_t}\boldsymbol{f}_\theta(t, \boldsymbol{x}_t), \tag{7}$$

where $\beta_t = -\alpha_t$ for noise prediction with and $\beta_t = \sigma_t$ for target prediction. This choice of $\beta$ and $\boldsymbol{f}_\theta$ thus depends on the particulars of the noise or data reparameterization.

**Remark 3.1.** Without loss of generality any of the results for the probability flow ODE apply to any arbitrary flow model which models an *affine probability path* (Lipman et al., 2024) with the correct conversions to the flow matching conventions.[4]

It is well observed that the structure of the ODE in Equation (7) can be greatly simplified via *exponential integrators* (Lu et al., 2022b; Zhang & Chen, 2023; Blasingame & Liu, 2024a). We make use of this insight to rewrite the ODE in a form which eliminates the discretization error in the $f(t)\boldsymbol{x}_t$ linear term along with a time reparameterization which will simplify the construction of the reversible solver. To achieve the time reparameterization we introduce a new variable $\varsigma_t$ defined as the *signal-to-noise ratio* (SNR) $\alpha_t/\sigma_t$ for the data prediction formulation and defined as the inverse SNR $\sigma_t/\alpha_t$ for the noise prediction formulation. Using this time change we find Proposition 3.1, in Section C.1.1 we provide the full derivation of this result.

**Proposition 3.1** (Reparameterization of the probability flow ODE). *The probability flow ODE in Equation (7) can be rewritten in $\varsigma_t$ as*

$$\frac{\mathrm{d}\boldsymbol{y}_\varsigma}{\mathrm{d}\varsigma} = \beta_T\boldsymbol{f}_\theta\left(\varsigma, \frac{\beta_\varsigma}{\beta_\varsigma}\boldsymbol{y}_\varsigma\right), \tag{8}$$

*where $\boldsymbol{y}_t = \frac{\beta_T}{\beta_t}\boldsymbol{x}_t$.*

The remaining step to constructing Rex is to perform a similar process but for an underlying explicit Runge-Kutta scheme by making use of Lawson methods (a particular class of exponential integrators) (Lawson, 1967; Hochbruck et al., 2020). However, since *both* the ODE and SDE version of Rex share the same computational graph, we will delay this presentation until we have discussed the SDE case.

## 3.2 REVERSE-TIME DIFFUSION SDE

Unlike with the ODE scenario the forms of the data and noise prediction formulations differ more significantly. As such we opt to focus only on the data prediction formulation which slightly less complicated and leave the details on the noise prediction formulation to Appendix C.2. It is well known (Lu et al., 2022a) that the reverse-time diffusion SDE in Equation (3) can be rewritten in terms of the data prediction model as

$$\mathrm{d}\boldsymbol{X}_t = \left[\left(f(t) + \frac{g^2(t)}{\sigma_t^2}\right)\boldsymbol{X}_t - \frac{\alpha_t g^2(t)}{\sigma_t^2}\boldsymbol{x}_{0|t}^\theta(\boldsymbol{X}_t)\right]\mathrm{d}t + g(t)\,\mathrm{d}\overline{\boldsymbol{W}}_t. \tag{9}$$

---

[4]*I.e.*, sampling in forward-time such that $\boldsymbol{X}_1 \sim q(\boldsymbol{X})$ and $\boldsymbol{X}_0 \sim p(\boldsymbol{X})$.

Remarkably, following a similar derivation to the one above for the probability flow ODE yields a time-changed SDE with a very similar form to the one above, sans the Brownian motion term and different weighting terms. We present this result in Proposition 3.2 with the full proof in Section C.2.2.

**Proposition 3.2** (Time reparameterization of the reverse-time diffusion SDE). *The reverse-time SDE in Equation* (9) *can be rewritten in terms of the data prediction model as*

$$\mathrm{d}\boldsymbol{Y}_\varrho = \frac{\sigma_T}{\gamma_T} \boldsymbol{x}_{0|\varrho}^\theta \left( \frac{\gamma_T \sigma_\varrho}{\sigma_T \gamma_\varrho} \boldsymbol{Y}_\varrho \right) \mathrm{d}\varrho + \frac{\sigma_T}{\gamma_T} \mathrm{d}\boldsymbol{W}_\varrho, \tag{10}$$

*where* $\boldsymbol{Y}_t = \frac{\sigma_T^2 \alpha_t}{\sigma_t^2 \alpha_T} \boldsymbol{X}_t$ *and* $\varrho_t \coloneqq \frac{\alpha_t^2}{\sigma_t^2}$.

**Stochastic Runge-Kutta.** Before constructing a reversible solver for the reverse-time SDE in Equation (10), we will zoom out to contextualize the discussion within the study of neural SDEs and to introduce *stochastic Runge-Kutta (SRK)* methods. Constructing a numerical scheme for SDEs is greatly more complicated than ODEs due to the complexities of stochastic processes and in particular stochastic integrals. Unlike numerical schemes for ODEs which are usually built upon truncated Taylor expansions, SDEs require constructing truncated Itô or Stratonovich-Taylor expansions (Kloeden & Platen, 1991) which results in numerous iterated stochastic integrals. Approximating these iterated integrals, or equivalently Lévy areas, of Brownian motion is quite difficult (Clark & Cameron, 2005; Mrongowius & Rößler, 2022); however, SDEs with certain constraints on the diffusion term may use specialized solvers to further achieve a strong order of convergence with simple approximations of these iterated stochastic integrals. As such there are several ways to express SRK methods depending on the choice of approximating these iterated integrals. We choose to follow the work of Foster et al. (2024) which makes usage of the *space-time Lévy area* in constructing such methods. The space-time Lévy area (see Foster et al., 2020, Definition 3.5; *cf.* Rößler, 2010) is defined below in Definition 3.2.

**Definition 3.2** (Space-time Lévy area). The rescaled space-time Lévy area of a Brownian motion $\{W_t\}$ on the interval $[s, t]$ corresponds to the signed area of the associated bridge process

$$H_{s,t} \coloneqq \frac{1}{h} \int_s^t \left( W_{s,u} - \frac{u-s}{h} W_{s,t} \right) \mathrm{d}u, \tag{11}$$

where $h \coloneqq t - s$ and $W_{s,u} = W_u - W_s$ for $u \in [s, t]$.

In particular, for additive-noise SDEs which our SDE in Equation (10) is, the Itô and Stratonovich integrals coincide and the numerical scheme is significantly simpler, for more details we refer to Appendix B.

## 3.3 THE REX SOLVER

Equipped with both Proposition 3.1 and Proposition 3.2 we are now ready to construct Rex. The key idea is to construct a reversible scheme from an explicit (S)RK scheme (we provide more detail in Appendix B) for the reparameterized differential equation using the McCallum-Foster method and then apply Lawson methods to bring the scheme back to the original state variable, *cf.* Figure 2.

We present the full scheme for the Rex solver below in Proposition 3.3 with the full derivation found in Appendix C.

**Proposition 3.3** (Rex). *Without loss of generality let* $\boldsymbol{\Phi}$ *denote an explicit SRK scheme for the SDE in Equation* (10) *with extended Butcher tableau* $a_{ij}, b_i, c_i, a_i^W, a_i^H, b^W, b^H$. *Fix an* $\omega \in \Omega$ *and let* $\boldsymbol{W}$ *be the Brownian motion over time variable* $\varsigma$. *Then the reversible solver constructed from* $\boldsymbol{\Phi}$ *in terms of the underlying state variable* $\boldsymbol{X}_t$ *is given by the forward step*

$$\boldsymbol{X}_{n+1} = \frac{w_{n+1}}{w_n} \left( \zeta \boldsymbol{X}_n + (1 - \zeta) \hat{\boldsymbol{X}}_n \right) + w_{n+1} \boldsymbol{\Psi}_h(\varsigma_n, \hat{\boldsymbol{X}}_n, \boldsymbol{W}_n(\omega)),$$

$$\hat{\boldsymbol{X}}_{n+1} = \frac{w_{n+1}}{w_n} \hat{\boldsymbol{X}}_n - w_{n+1} \boldsymbol{\Psi}_{-h}(\varsigma_{n+1}, \boldsymbol{X}_{n+1}, \boldsymbol{W}_n(\omega)), \tag{12}$$

*and backward step*

$$\hat{\boldsymbol{X}}_n = \frac{w_n}{w_{n+1}} \hat{\boldsymbol{X}}_{n+1} + w_n \boldsymbol{\Psi}_{-h}(\varsigma_{n+1}, \boldsymbol{X}_{n+1}, \boldsymbol{W}_n(\omega)),$$

$$\boldsymbol{X}_n = \frac{w_n}{w_{n+1}} \zeta^{-1} \boldsymbol{X}_{n+1} + (1 - \zeta^{-1}) \hat{\boldsymbol{X}}_n - w_n \zeta^{-1} \boldsymbol{\Psi}_h(\varsigma_n, \hat{\boldsymbol{X}}_n, \boldsymbol{W}_n(\omega)), \tag{13}$$

*with step size $h := \varsigma_{n+1} - \varsigma_n$ and where $\boldsymbol{\Psi}$ denotes the following scheme*

$$\hat{\boldsymbol{Z}}_i = \frac{1}{w_n} \boldsymbol{X}_n + h \sum_{j=1}^{i-1} \left[ a_{ij} \boldsymbol{f}^\theta \left( \varsigma_n + c_j h, w_{\varsigma_n + c_j h} \hat{\boldsymbol{Z}}_j \right) \right] + a_i^W \boldsymbol{W}_n(\omega) + a_i^H \boldsymbol{H}_n(\omega),$$

$$\boldsymbol{\Psi}_h(\varsigma_n, \boldsymbol{X}_n, \boldsymbol{W}_\varrho(\omega)) = h \sum_{j=1}^{s} \left[ b_i \boldsymbol{f}^\theta \left( \varsigma_n + c_i h, w_{\varsigma_n + c_i h} \hat{\boldsymbol{Z}}_j \right) \right] + b^W \boldsymbol{W}_n(\omega) + b^H \boldsymbol{H}_n(\omega),$$

(14)

*where $\boldsymbol{f}^\theta$ denotes the data prediction model, $w_n = \frac{\sigma_n}{\gamma_n}$ and $\varsigma_t = \varrho_t$. The ODE case is recovered for an explicit RK scheme $\boldsymbol{\Phi}$ for the ODE in Equation (70) with $w_n = \sigma_n$ and $\varsigma_t = \gamma_t$ For noise prediction models we have $\boldsymbol{f}^\theta$ denoting the noise prediction model with $w_n = \alpha_n$ and $\varsigma_t = \frac{\sigma_n}{\alpha_n}$.*

We still have yet to address how to construct an *algebraically reversible* scheme for a *stochastic* process, but merely stated it above in Proposition 3.3, we will now, however, justify our design decisions above. The key idea is to use the *same* realization of the Brownian motion in both the forward pass or backward pass. This has been explored in prior works studying the continuous adjoint equations for neural SDEs (Li et al., 2020; Kidger et al., 2021) and essentially amounts to fixing the realization of the Brownian motion along with clever strategies for reconstructing the same realization. Formally, let $(\Omega, \mathcal{F}, \mathbb{P})$ be the probability space and let $\boldsymbol{W}_t : \Omega \to \mathbb{R}^{d_w}$ be the standard Brownian motion on $[0, T]$. Then for each reversible solve we fix an $\omega \in \Omega$. This can be justified if we view the SDE from a roughs path perspective, *i.e.*, the Itô-Lyons map (Lyons, 1998) provides a deterministic continuous map from the initial condition of the SDE and realization of the Brownian motion to the solution trajectory, see Appendix F for a more detailed explanation.

**Numerical simulation of the Brownian motion.** The naïve way to fix the realization of the Brownian motion for both the forward pass is to simply store the entire realization of the Brownian motion in system memory, *i.e.*, record $\{\boldsymbol{W}_n(\omega)\}_{n=1}^N$ *à la* Wu & la Torre (2023).[5] However, recent work by Li et al. (2020); Kidger et al. (2021); Jelinčič et al. (2024) have proposed much more elegant solutions which enable one to recalculate *any* realization of the Brownian motion from a single seed given access to a splittable *pseudo-random number generator* (PRNG) (Salmon et al., 2011). *N.B.*, we discuss the more nuanced technical details of such approaches in Appendix G, for now it suffices to say we adopt a more elegant solution to reconstructing the Brownian motion in the backward step.

## 4 THEORETICAL RESULTS

### 4.1 CONVERGENCE ORDER AND STABILITY

A nice property of the McCallum-Foster is that the the convergence order of the underlying explicit RK scheme $\boldsymbol{\Phi}$ is inherited by the resulting reversible scheme McCallum & Foster (2024, Theorem 2.1). However, does this property hold true for Rex? Fortunately, it does indeed hold true which we show in Theorem 4.1 with the proof provided in Appendix D.2.

**Theorem 4.1** (Rex is a $k$-th order solver). *Let $\boldsymbol{\Phi}$ be a $k$-th order explicit Runge-Kutta scheme for the reparameterized probability flow ODE in Equation (70) with variance preserving noise schedule $(\alpha_t, \sigma_t)$. Then Rex constructed from $\boldsymbol{\Phi}$ is a $k$-th order solver, i.e., given the reversible solution $\{\boldsymbol{x}_n, \hat{\boldsymbol{x}}_n\}_{n=1}^N$ and true solution $\boldsymbol{x}_{t_n}$ we have*

$$\|\boldsymbol{x}_n - \boldsymbol{x}_{t_n}\| \le Ch^k, \tag{15}$$

*for constants $C, h_{max} > 0$ and for step sizes $h \in [0, h_{max}]$.*

We can show a similar result for the underlying scheme $\boldsymbol{\Psi}$ constructed from an explicit SRK $\boldsymbol{\Phi}$ with the full proof provided in Appendix D.3.

**Theorem 4.2** (Convergence order for stochastic $\boldsymbol{\Psi}$). *Let $\boldsymbol{\Phi}$ be a SRK scheme with strong order of convergence $\xi > 0$ for the reparameterized reverse-time diffusion SDE in Equation (10) with variance preserving noise schedule $(\alpha_t, \sigma_t)$ and $\alpha_T > 0$. Then $\boldsymbol{\Psi}$ constructed from $\boldsymbol{\Phi}$ has strong order of convergence $\xi$.*

---

[5]This clearly prohibits the use of adaptive step-size solvers.

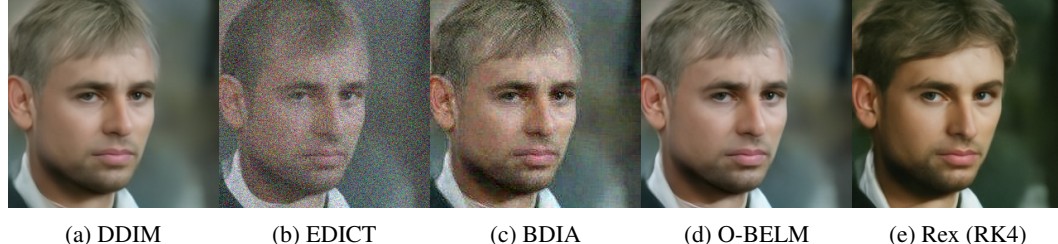

|        |        |        |        |            |
|:------:|:------:|:------:|:------:|:----------:|
| (a) DDIM | (b) EDICT | (c) BDIA | (d) O-BELM | (e) Rex (RK4) |

Figure 3: Qualitative comparison of unconditional sampling with different reversible solvers with a pre-trained DDPM model on CelebA-HQ ($256 \times 256$) with the non-reversible DDIM as a baseline. Each method used 10 discretization steps.

**Stability.** One drawback of reversible solvers is their rather unimpressive stability, in fact until the work of McCallum & Foster (2024) there were no reversible methods which had a non-trivial region of stability. We discuss this more in detail Appendix A.2 along with illustrating the poor stability characteristics of BDIA and O-BELM (see Corollaries A.4.1 and A.3.2). However, since Rex is built upon the McCallum-Foster method the ODE solver has some stability.[6]

### 4.2 RELATION TO EXISTING SOLVERS

Next we show that several variants of Rex are actually the *reversible versions* of several well-known solvers in the literature for diffusion models, *e.g.*, the DPM-Solvers (Lu et al., 2022b). We state this result below in Theorem 4.3 with the full details and proofs in Appendix E.

**Theorem 4.3** (Rex subsumes previous solvers). *The underlying scheme used $\Psi$ in Rex given by*

$$\hat{\boldsymbol{Z}}_i = \frac{1}{w_n} \boldsymbol{X}_n + h \sum_{j=1}^{i-1} \left[ a_{ij} \boldsymbol{f}^\theta \left( \varsigma_n + c_j h, w_{\varsigma_n + c_j h} \hat{\boldsymbol{Z}}_j \right) \right] + a_i^W \boldsymbol{W}_n(\omega) + a_i^H \boldsymbol{H}_n(\omega),$$

$$\boldsymbol{X}_{n+1} = \frac{w_{n+1}}{w_n} \boldsymbol{X}_n + w_{n+1} \left( h \sum_{j=1}^{s} \left[ b_i \boldsymbol{f}^\theta \left( \varsigma_n + c_i h, w_{\varsigma_n + c_i h} \hat{\boldsymbol{Z}}_j \right) \right] + b^W \boldsymbol{W}_n(\omega) + b^H \boldsymbol{H}_n(\omega) \right),$$

$$\tag{16}$$

*subsumes the following solvers for diffusion models*

1. *DDIM (Song et al., 2021a),*

2. *DPM-Solver-1, DPM-Solver-2, DPM-Solver-12 (Lu et al., 2022b),*

3. *DPM-Solver++1, DPM-Solver++(2S), SDE-DPM-Solver-1, SDE-DPM-Solver++1 (Lu et al., 2022a),*

4. *SEEDS-1 (Gonzalez et al., 2024), and*

5. *gDDIM (Zhang et al., 2023).*

**Corollary 4.3.1** (Rex is reversible version of previous solvers). *Rex is the reversible revision of the well-known solvers for diffusion models in Theorem 4.3.*

## 5 EMPIRICAL RESULTS

### 5.1 IMAGE GENERATION

**Unconditional image generation.** Following prior works (Wang et al., 2024; Wallace et al., 2023) we begin by exploring the ability of Rex to function as a traditionally solver for diffusion models. To evaluate this we drew 10,240 samples using a DDPM model (Ho et al., 2020) pretrained on the

---

[6]*I.e.*, in the sense of the linear test equation, see Appendix A.2 for more details.

Table 1: Quantitative comparison of different reversible solvers for unconditional image generation with a pre-trained DDPM model on CelebA-HQ ($256 \times 256$) with the non-reversible DDIM as a baseline. † denotes $\gamma = 0.5$ and ‡ denotes $\gamma = 1.0$ for BDIA hyperparameter.

| Steps | Solver | FD ($\downarrow$) | FD$_\infty$ ($\downarrow$) | Precision ($\uparrow$) | Recall ($\uparrow$) | Density ($\uparrow$) | Coverage ($\uparrow$) |
|---|---|---|---|---|---|---|---|
| 10 | EDICT | 1042.89 | 1034.82 | 0.49 | 0.10 | 0.19 | 0.11 |
| | BDIA† | 900.95 | 894.23 | 0.61 | 0.10 | 0.28 | 0.14 |
| | BDIA‡ | 1284.48 | 1274.46 | 0.41 | 0.00 | 0.14 | 0.05 |
| | O-BELM | **605.52** | **596.47** | 0.78 | 0.18 | 0.56 | 0.34 |
| | Rex (RK4) | 633.90 | 617.11 | **0.81** | **0.22** | **0.64** | 0.36 |
| | Rex (Midpoint) | **607.20** | **597.04** | 0.78 | 0.21 | 0.60 | **0.37** |
| | Rex (Euler-Maruyama) | **610.16** | **598.56** | 0.79 | 0.10 | 0.61 | **0.37** |
| | DDIM | 727.75 | 716.41 | 0.75 | 0.14 | 0.49 | 0.27 |
| 20 | EDICT | 752.68 | 743.89 | 0.68 | 0.15 | 0.36 | 0.21 |
| | BDIA† | 611.47 | 601.37 | 0.76 | 0.19 | 0.50 | 0.30 |
| | BDIA‡ | 982.30 | 968.62 | 0.54 | 0.10 | 0.22 | 0.10 |
| | O-BELM | 489.94 | 477.82 | 0.82 | 0.23 | 0.71 | 0.43 |
| | Rex (RK4) | 547.24 | 533.30 | 0.82 | **0.27** | 0.71 | 0.43 |
| | Rex (Midpoint) | 539.96 | 527.85 | 0.81 | 0.26 | 0.66 | 0.41 |
| | Rex (Euler-Maruyama) | **460.42** | **447.01** | **0.86** | 0.21 | **0.91** | **0.51** |
| | DDIM | 570.11 | 555.26 | 0.79 | 0.20 | 0.62 | 0.38 |
| 50 | EDICT | 551.13 | 534.73 | 0.78 | 0.24 | 0.60 | 0.37 |
| | BDIA† | 500.79 | 489.24 | 0.82 | 0.27 | 0.70 | 0.44 |
| | BDIA‡ | 798.47 | 790.17 | 0.71 | 0.12 | 0.39 | 0.18 |
| | O-BELM | 476.29 | 463.07 | 0.84 | **0.29** | 0.77 | 0.45 |
| | Rex (RK4) | 511.17 | 498.94 | 0.80 | 0.27 | 0.69 | 0.44 |
| | Rex (Midpoint) | 505.67 | 494.94 | 0.81 | **0.29** | 0.70 | 0.44 |
| | Rex (Euler-Maruyama) | **391.93** | **381.01** | **0.87** | 0.28 | **0.98** | **0.56** |
| | DDIM | 490.88 | 479.87 | 0.80 | 0.26 | 0.67 | 0.45 |

CelebA-HQ (Karras et al., 2018) dataset with the various solvers each using the same fixed seed. Following Stein et al. (2023), we report the *Fréchet distance* (FD) with DINOv2 (Oquab et al., 2023) feature extractor along with FD$_\infty$ (Chong & Forsyth, 2020). We also report the precision and recall metrics (Kynkäänniemi et al., 2019); along with density and coverage metrics (Naeem et al., 2020) which serve as a proxy for fidelity and sample diversity respectively. We provide more details on these metrics in Section I.1.2. In Table 1 we compare pre-existing methods for exact inversion with diffusion models against Rex, along with including the non-reversible DDIM solver as a baseline. We observe that the Rex family of reversible solvers performs exceedingly well, surpassing the baseline non-reversible DDIM scheme, handily beating EDICT and BDIA, and often outperforming O-BELM. We observe that our reversible SDE scheme consistently performs quite well outside of the very few step-size regime (a well known limitation of SDE schemes). *N.B.*, that unlike the results reported for the other reversible solvers we did not search for the optimal hyperparameters for Rex for the sampling task. In Figure 3 we present a visual qualitative comparison of the different solvers using the same initial noise. We provide additional experimental details in Appendix I.1.

Table 2: Quantitative comparison of different reversible solvers in terms of average CLIP score, Image Reward, and PickScore. for conditional text-to-image generation with Stable Diffusion v1.5 ($512 \times 512$) with the non-reversible DDIM as a baseline.

| Solver / Steps | CLIP score ($\uparrow$) | | | Image Reward ($\uparrow$) | | | PickScore ($\uparrow$) | | |
|---|---|---|---|---|---|---|---|---|---|
| | 10 | 20 | 50 | 10 | 20 | 50 | 10 | 20 | 50 |
| EDICT | 27.97 | 31.04 | 31.17 | -1.219 | -0.134 | -0.055 | 19.52 | 20.84 | 21.05 |
| BDIA $\gamma = 0.96$ | 31.11 | 31.52 | 31.54 | -0.111 | 0.067 | 0.087 | 20.52 | 21.01 | 21.19 |
| BDIA $\gamma = 0.5$ | 31.57 | 31.48 | 31.48 | -0.006 | 0.055 | 0.066 | 20.98 | 21.16 | 21.21 |
| O-BELM | 31.47 | 31.43 | 31.51 | 0.051 | 0.105 | 0.160 | 20.88 | 21.00 | 21.16 |
| Rex (Midpoint) | 31.62 | **31.64** | **31.60** | 0.119 | 0.179 | 0.198 | 21.28 | 21.38 | 21.41 |
| Rex (RK4) | **31.69** | 31.60 | 31.57 | 0.156 | 0.187 | 0.195 | 21.35 | 21.40 | 21.41 |
| Rex (Euler-Maruyama) | **31.68** | 31.56 | 31.33 | 0.222 | 0.239 | **0.264** | **21.50** | **21.66** | 21.70 |
| Rex (ShARK) | 31.55 | 31.56 | 31.39 | **0.239** | **0.249** | 0.263 | **21.51** | **21.66** | **21.72** |
| DDIM | 31.78 | 31.76 | 31.24 | 0.033 | 0.136 | 0.247 | 21.06 | 21.29 | 21.04 |

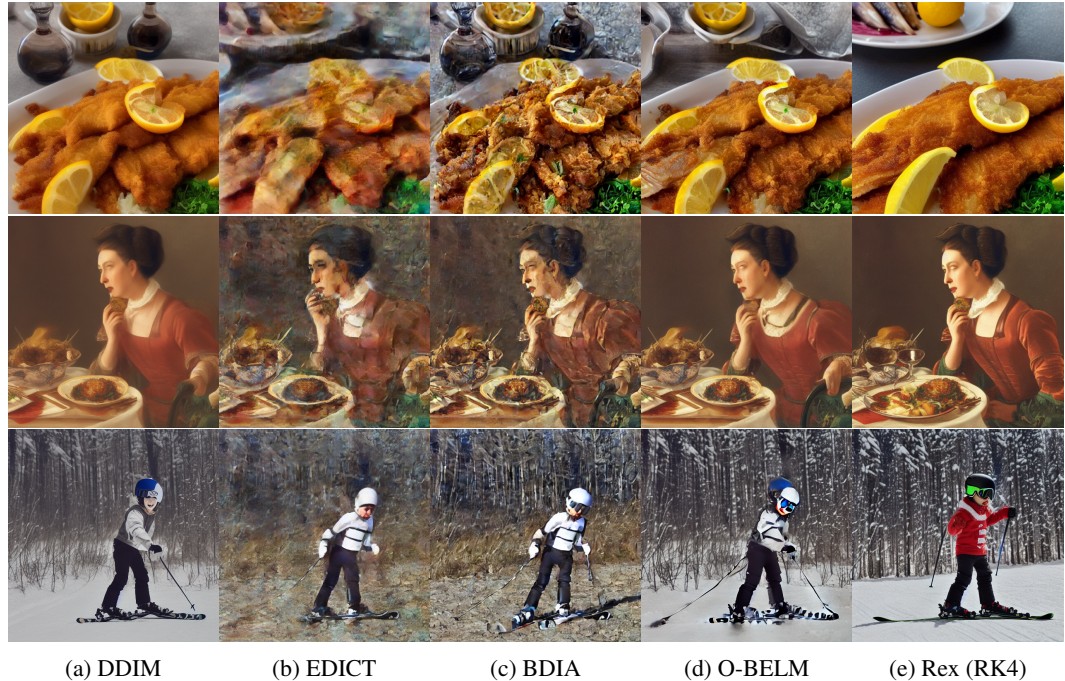

|  (a) DDIM | (b) EDICT | (c) BDIA | (d) O-BELM | (e) Rex (RK4) |

Figure 4: Qualitative comparison of text-to-image conditional sampling with different reversible solvers with Stable Diffusion v1.5 ($512 \times 512$) and 10 discretization steps. Prompts from top to bottom are: "White plate with fried fish and lemons sitting on top of it.", "A lady enjoying a meal of some sort.", and "A young boy riding skis with ski poles.".

**Conditional image generation.** To further evaluate Rex we drew text-conditioned samples using Stable Diffusion v1.5 (Rombach et al., 2022) with a set of 1000 randomly selected captions from COCO (Lin et al., 2014) with the various solvers each using the same fixed seed. We report performance in terms of the CLIP Score (Hessel et al., 2021); in terms of the state-of-the-art text-to-image scoring function PickScore (Kirstain et al., 2023); and in terms of the state-of-the-art Image Reward metric (Xu et al., 2023) which assigns a score that reflects human preferences, namely, aesthetic quality and prompt adherence. The later metric was recently become a popular metric for evaluating the performance of diffusion models (Skreta et al., 2025a). In Table 2 we compare pre-existing methods for exact inversion with diffusion models against Rex, along with including the non-reversible DDIM solver as a baseline. We observe that Rex does very well compared to other reversible solvers, and in particular the stochastic variants of Rex perform *extremely* well. In Figure 4 we present a visual qualitative comparison of the different solvers using the same initial noise. We provide additional experimental details in Appendix I.2.

## 5.2 IMAGE INTERPOLATION

We explore interpolating between the inversions of two images, a difficult problem as the inverted space is often non-Gaussian (Blasingame & Liu, 2024b). We illustrate an example of this in Figure 5 exploring interpolation with an unconditional DDPM model. We notice the that stochastic Rex has much better interpolations properties than both ODE inversions corroborating with Nie et al. (2024). Both ODE variants seem to fail quite noticeably, unable to smoothly interpolate between the two samples. *N.B.*, we noticed that the inverted samples with ShARK had variance much closer to one, whereas the other inverted samples had much larger variance, likely contributing to the distortions, we discuss this more in Appendix K.

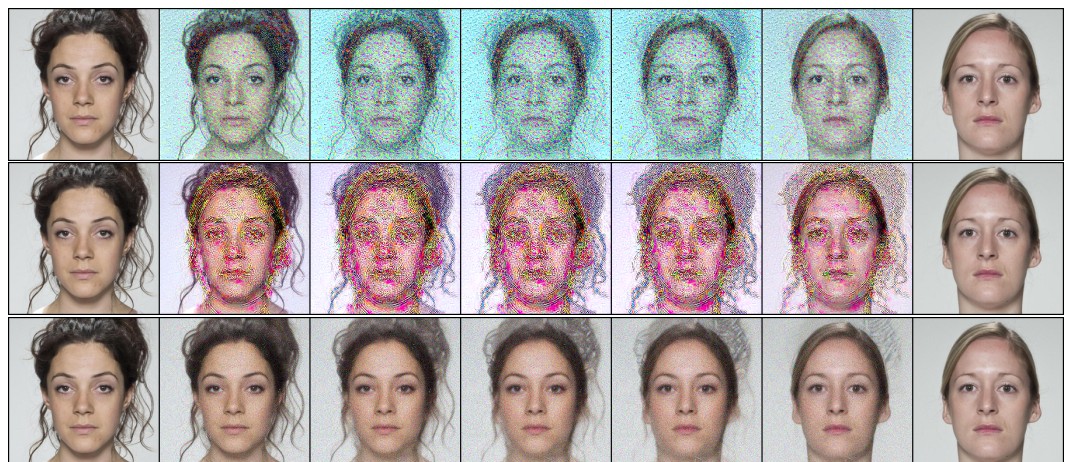

Figure 5: Unconditional interpolation between two real images from FRLL (DeBruine & Jones, 2017) with a DDPM model trained on CelebA-HQ. Top row is BELM, middle is Rex (Euler), and bottom is Rex (ShARK). 50 steps used for each method.

## 6  CONCLUSION

We propose *Rex* a family of algebraically reversible solvers for diffusion models which can obtain arbitrarily a high order of convergence (for the ODE case). Moreover, we propose (to the best of our knowledge) the first method for exact inversion for diffusion SDEs without storing the entire trajectory of the Brownian motion. Our empirical illustrations show that not only does Rex have nice theoretical properties but it also functions as a capable numerical scheme for sampling with diffusion models. The proposed method can be incorporated into preexisting applications wherein preserving the bijections of flow maps is important, leading to many exciting possible applications.

### ETHICS STATEMENT

We recognize that Rex as numerical scheme for sampling with diffusion models could potentially be misused used for malicious applications particularly when used in editing pipelines.

### REPRODUCIBILITY STATEMENT

To aid with reproducibility we include detailed derivations of Rex in Appendix C along with additional proofs in Appendix D. We draw connections between Rex and other solver for diffusion models in Appendix E. We include through implementation details in Appendix H and experimental details in Appendix I; in particular, we mention all code repositories and datasets we used in Appendix I.5. Moreover, we provide code illustrations of the core components of Rex in Appendix J.

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

APPENDICES

OVERVIEW OF THEORETICAL RESULTS

## A    RELATED WORKS

In this section we provide a detailed comparison with relevant related works. We begin in Appendix A.1 by providing an overview of algebraically reversible solvers. Then in Appendix A.2 we introduce the stability of an ODE solver, a helpful tool in comparing reversible solvers. Using this tool along with examining the convergence order we compare a variety of reversible solvers for diffusion models in Appendix A.3. Lastly, in Appendix A.4 we explore related work on constructing SDE solvers for diffusion models.

### A.1    REVERSIBLE SOLVERS

The earliest work on reversible solvers can be traced back to the pioneering work on symplectic integrators by Vogelaere (1956); Ruth (1983); Feng (1984). Due to symplectic integrators being developed for solving Hamiltonian systems they are intrinsically reversible by construction (Greydanus et al., 2019). More recently, Matsubara et al. (2021) explored the use of symplectic solvers for solving the continuous adjoint equations. Likewise, work by Pan et al. (2023) extended this idea, making use of symplectic solvers for solving the continous adjoint equations for diffusion models. However, in this section we will focus on non-symplectic reversible solvers.

Throughout this section we consider solving the following $d$-dimensional IVP:

$$\boldsymbol{x}(0) = \boldsymbol{x}_0, \qquad \frac{\mathrm{d}\boldsymbol{x}}{\mathrm{d}t}(t) = \boldsymbol{f}(t, \boldsymbol{x}(t)), \tag{17}$$

over the time interval $[0, T]$ with numerical solution $\{\boldsymbol{x}_n\}_{n=0}^N$.

#### A.1.1    ASYNCHRONOUS LEAPFROG METHOD

To the best of our knowledge the *asynchronous leapfrog definition* was the first algebraically reversible non-symplectic solver, initially proposed by Mutze (2013) and popularized in a modern deep learning context by Zhuang et al. (2021). The asynchronous leapfrog method is a modification of the leapfrog method which converts it from a multi-step to single-step method. The method keeps track of a second state, $\{\boldsymbol{v}_n\}$ which is supposed to be *sufficiently close* to the value of the vector field. We define the method below in Definition A.1.

**Definition A.1** (Asynchronous leapfrog method). Initialize $\boldsymbol{v}_0 = \boldsymbol{f}(0, \boldsymbol{x}_0)$. Consider a step size of $h$ and let $\hat{t}_n = t_n + h/2$, then a forward step of the asynchronous leapfrog method is defined as

$$\begin{aligned}
\hat{\boldsymbol{x}}_n &= \boldsymbol{x}_n + \frac{1}{2}\boldsymbol{v}_n h, \\
\boldsymbol{v}_{n+1} &= 2\boldsymbol{f}(\hat{t}_n, \hat{\boldsymbol{x}}_n) - \boldsymbol{v}_n, \\
\boldsymbol{x}_{n+1} &= \boldsymbol{x}_n + \boldsymbol{f}(\hat{t}_n, \hat{\boldsymbol{x}}_n)h,
\end{aligned} \tag{18}$$

and the backward step is given as

$$\begin{aligned}
\hat{\boldsymbol{x}}_n &= \boldsymbol{x}_{n+1} - \frac{1}{2}\boldsymbol{v}_{n+1} h, \\
\boldsymbol{x}_n &= \boldsymbol{x}_{n+1} - \boldsymbol{f}(\hat{t}_n, \hat{\boldsymbol{x}}_n)h, \\
\boldsymbol{v}_n &= 2\boldsymbol{f}(\hat{t}_n, \hat{\boldsymbol{x}}_n) - \boldsymbol{v}_{n+1}.
\end{aligned} \tag{19}$$

**Remark A.2.** The method is a second-order solver (Zhuang et al., 2021, Theorem 3.1).

#### A.1.2    REVERSIBLE HEUN METHOD

Later work by Kidger et al. (2021) proposed the *reversible Heun method*, a general purpose reversible solver which is symmetric and is an algebraically reversible SDE solver in addition to being a reversible ODE solver. This solver keeps track of an auxiliary state variable $\hat{\boldsymbol{x}}_n$ and an extra copy of previous evaluations of the drift and diffusion coefficients. We present this method below in Definition A.3.

**Definition A.3** (Reversible Heun method for ODEs). Initialize $\hat{\boldsymbol{x}}_0 = \boldsymbol{x}_0$. Consider a step size of $h$, then a forward step of the reversible Heun method is defined as

$$
\begin{aligned}
\hat{\boldsymbol{x}}_{n+1} &= 2\boldsymbol{x}_n - \hat{\boldsymbol{x}}_n + \boldsymbol{f}(t_n, \hat{\boldsymbol{x}}_n)h, \\
\boldsymbol{x}_{n+1} &= \boldsymbol{x}_n + \frac{1}{2}\left(\boldsymbol{f}(t_{n+1}, \hat{\boldsymbol{x}}_{n+1}) + \boldsymbol{f}(t_n, \hat{\boldsymbol{x}}_n)\right)h.
\end{aligned}
\tag{20}
$$

and the backward step is given as

$$
\begin{aligned}
\hat{\boldsymbol{x}}_n &= 2\boldsymbol{x}_{n+1} - \hat{\boldsymbol{x}}_{n+1} - \boldsymbol{f}(t_{n+1}, \hat{\boldsymbol{x}}_{n+1})h, \\
\boldsymbol{x}_n &= \boldsymbol{x}_{n+1} - \frac{1}{2}\left(\boldsymbol{f}(t_{n+1}, \hat{\boldsymbol{x}}_{n+1}) + \boldsymbol{f}(t_n, \hat{\boldsymbol{x}}_n)\right)h.
\end{aligned}
\tag{21}
$$

**Remark A.4.** This method is a second-order solver (Kidger, 2022, Theorem 5.18).

Recall that simulating SDEs in reverse-time is much trickier than simulating ODEs in reverse-time. This observation is even more true of algebraically reversible methods for SDEs. To the best of our knowledge, the only general reversible solver for SDEs is the reversible Heun method. The main idea of the SDE formulation of the reversible Heun method is to extend the Euler-Heun method[7] like how Heun's method was extended to the reversible Heun solver for ODEs. We define the method in Kidger et al. (2021, Algorithm 1) below in Definition A.5.

**Definition A.5** (Reversible Heun method for SDEs). Initialize $\hat{\boldsymbol{x}}_0 = \boldsymbol{x}_0$. Consider a step size of $h$ and let $\boldsymbol{W}_h := \boldsymbol{W}_{t_{n+1}} - \boldsymbol{W}_{t_n}$, then a forward step of the reversible Heun method is defined as

$$
\begin{aligned}
\hat{\boldsymbol{x}}_{n+1} &= 2\boldsymbol{x}_n - \hat{\boldsymbol{x}}_n + \boldsymbol{\mu}(t_n, \hat{\boldsymbol{x}}_n)h + \boldsymbol{\sigma}(t_n, \hat{\boldsymbol{x}}_n)\boldsymbol{W}_h, \\
\boldsymbol{x}_{n+1} &= \boldsymbol{x}_n + \frac{1}{2}\left(\boldsymbol{\mu}(t_{n+1}, \hat{\boldsymbol{x}}_{n+1}) + \boldsymbol{\mu}(t_n, \hat{\boldsymbol{x}}_n)\right)h \\
&\quad + \frac{1}{2}\left(\boldsymbol{\sigma}(t_{n+1}, \hat{\boldsymbol{x}}_{n+1}) + \boldsymbol{\sigma}(t_n, \hat{\boldsymbol{x}}_n)\right)\boldsymbol{W}_h.
\end{aligned}
\tag{22}
$$

and the backward step is given as

$$
\begin{aligned}
\hat{\boldsymbol{x}}_n &= 2\boldsymbol{x}_{n+1} - \hat{\boldsymbol{x}}_{n+1} - \boldsymbol{\mu}(t_{n+1}, \hat{\boldsymbol{x}}_{n+1})h - \boldsymbol{\sigma}(t_n, \hat{\boldsymbol{x}}_n)\boldsymbol{W}_h, \\
\boldsymbol{x}_n &= \boldsymbol{x}_{n+1} - \frac{1}{2}\left(\boldsymbol{\mu}(t_{n+1}, \hat{\boldsymbol{x}}_{n+1}) + \boldsymbol{\mu}(t_n, \hat{\boldsymbol{x}}_n)\right)h \\
&\quad - \frac{1}{2}\left(\boldsymbol{\sigma}(t_{n+1}, \hat{\boldsymbol{x}}_{n+1}) + \boldsymbol{\sigma}(t_n, \hat{\boldsymbol{x}}_n)\right)\boldsymbol{W}_h.
\end{aligned}
\tag{23}
$$

**Remark A.6.** This method requires some tractable solution for recalculating the Brownian motion from a splittable PRNG.

### A.1.3 MCCALLUM-FOSTER METHOD

Recent work by McCallum & Foster (2024) created a general method for constructing $n$-th order solvers from preexisting explicit single-step solvers while also addressing the stability issues that earlier methods suffered from. As McCallum & Foster (2024) simply refer to their method as *reversible X* where $X$ is the underlying single-step solver we opt to refer to their method as the *McCallum-Foster method*. We restate the definition below.

**Definition 2.1** (McCallum-Foster method). Initialize $\hat{\boldsymbol{x}}_0 = \boldsymbol{x}_0$ and let $\zeta \in (0, 1]$. Consider a step size of $h$, then a forward step of the McCallum-Foster method is defined as

$$
\begin{aligned}
\boldsymbol{x}_{n+1} &= \zeta\boldsymbol{x}_n + (1 - \zeta)\hat{\boldsymbol{x}}_n + \boldsymbol{\Phi}_h(t_n, \hat{\boldsymbol{x}}_n), \\
\hat{\boldsymbol{x}}_{n+1} &= \hat{\boldsymbol{x}}_n - \boldsymbol{\Phi}_{-h}(t_{n+1}, \boldsymbol{x}_{n+1}),
\end{aligned}
\tag{5}
$$

and the backward step is given as

$$
\begin{aligned}
\hat{\boldsymbol{x}}_n &= \hat{\boldsymbol{x}}_{n+1} + \boldsymbol{\Phi}_{-h}(t_{n+1}, \boldsymbol{x}_{n+1}), \\
\boldsymbol{x}_n &= \zeta^{-1}\boldsymbol{x}_{n+1} + (1 - \zeta^{-1})\hat{\boldsymbol{x}}_n - \zeta^{-1}\boldsymbol{\Phi}_h(t_n, \hat{\boldsymbol{x}}_n).
\end{aligned}
\tag{6}
$$

**Remark A.7.** *N.B.*, the $\zeta$ and $\zeta^{-1}$ terms in the forward and backward steps determine the stability of the system.

---

[7]This converges with strong order $\frac{1}{2}$ in the Stratonovich sense (Rüemelin, 1982).

Interestingly, McCallum & Foster (2024, Theorem 2.1) showed that this reversible method inherits the convergence order of single-step solver $\mathbf{\Phi}_h$ enabling the construction of an arbitrarily high-order reversible solver. We restate this result below in Theorem A.1.

**Theorem A.1** (Convergence order of the McCallum-Foster method). *Consider the ODE in Equation (17) over $[0, T]$ with fixed time horizon $T > 0$. Let $T = Nh$ where $N > 0$ is the number of discretization steps and $h > 0$ is the step size. Let $\mathbf{\Phi}$ be a $k$-th order ODE solver such that it satisfies the Lipschitz condition*

$$\|\mathbf{\Phi}_\eta(\cdot, \boldsymbol{a}) - \mathbf{\Phi}_\eta(\cdot, \boldsymbol{b})\| \leq L|\eta|\|\boldsymbol{a} - \boldsymbol{b}\|, \tag{24}$$

*for all $\boldsymbol{a}, \boldsymbol{b} \in \mathbb{R}^d$ and $\eta \in [-h_{max}, h_{max}]$ for some $h_{max} > 0$. Consider the reversible solution $\{\boldsymbol{x}_n, \hat{\boldsymbol{x}}_n\}_{n \in \mathbb{N}}$ admitted by Equation (5). Then there exists constants $h_{max} > 0$, $C > 0$, such that, for $h \in (0, h_{max}]$,*

$$\|\boldsymbol{x}_n - \boldsymbol{x}(t_n)\| \leq Ch^k. \tag{25}$$

## A.2 A NOTE ON STABILITY

Historically, the stability properties of reversible solvers has been one of their weakest attributes (Kidger, 2022), limiting their use in practical applications. We formally introduce the notation of stability following Kidger (2022, Definition C.39), which we rewrite below in Definition A.8.

**Definition A.8** (Region of stability). Fix some numerical differential equation solver and let $\{\boldsymbol{x}_n^{\lambda,h}\}_{n \in \mathbb{N}}$ be the solution admitted by the numerical scheme solving the linear (or Dahlquist) test equation

$$\boldsymbol{x}(0) = \boldsymbol{x}_0, \qquad \frac{\mathrm{d}\boldsymbol{x}}{\mathrm{d}t} = \lambda\boldsymbol{x}(t), \tag{26}$$

where $\lambda \in \mathbb{C}$, $h > 0$ is the step size, and $\boldsymbol{x}_0 \in \mathbb{R}^d$ is a non-zero initial condition. The region of stability is defined as

$$\{h\lambda \in \mathbb{C} : \{\boldsymbol{x}_n^{\lambda,h}\}_{n \in \mathbb{N}} \text{ is uniformly bounded over } t_n\}. \tag{27}$$

*I.e.*, there exists a constant $C$ depending on $\lambda$ and $h$ but independent of $t_n$ such that $\|\boldsymbol{x}_n^{\lambda,h}\| < C$.

With the linear test equation Equation (26) the ODE converges asymptotically when $\Re(\lambda) \leq 0$,[8] and thus we are interested in numerical schemes which are bounded when the underlying analytical solution converges. Ideally, a numerical scheme would converge for all $h\lambda$ with $\Re(\lambda) < 0$.[9] Thus, the larger the region of stability the larger the step size we can take, wherein the numerical scheme still converges.

**Remark A.9.** Regrettably, the reversible Heun, leapfrog, and asynchronous leapfrog methods have poor stability properties. Specifically, the region of stability for all the methods is the complex interval $[-i, i]$, see Kidger (2022, Theorem 5.20) for reversible Heun, Shampine (2009, Section 2) for leapfrog, and Zhuang et al. (2021, Appendix A.4) for asynchronous leapfrog.

In other words, all previous reversible solvers are nowhere linearly stable for any step size $h$.[10] The instability in both asynchronous leapfrog and reversible Heun can be attributed to a step of general form $2A - B$, *i.e.*, we can write the source of instability as

$$2\boldsymbol{f}(\hat{t}_n, \hat{\boldsymbol{x}}_n) - \boldsymbol{v}_n, \qquad \text{(asynchronous leapfrog)}$$
$$2\boldsymbol{x}_{n+1} - \hat{\boldsymbol{x}}_{n+1}. \qquad \text{(reversible Heun)}$$

Thus the instability in these reversible schemes is caused by a decoupling between $\boldsymbol{v}_n$ and $\boldsymbol{f}(t_n, \boldsymbol{x}_n)$ (asynchronous leapfrog); and $\boldsymbol{x}_n$ and $\hat{\boldsymbol{x}}_n$ (reversible Heun). The strategy of McCallum & Foster (2024) is to couple $\boldsymbol{x}_n$ and $\hat{\boldsymbol{x}}_n$ together with the coupling parameter $\zeta$. Using this strategy, they showed that it was possible to construct a reversible solver with a non-trivial region of convergence. Let $\mathbf{\Phi}_h(t_n, \boldsymbol{x}_n) = R(h\lambda)\boldsymbol{x}_n$ and let $R(h\lambda)$ denote the *transfer function* used in analysis of Runge-Kutta methods with step size $h$ (see Stewart, 2022). We restate McCallum & Foster (2024, Theorem 2.3) below.

---

[8]The ODE converges to 0 when $\Re(\lambda) < 0$.

[9]A region of stability which satisfies is known as a region of absolute stability.

[10]Linearly stability refers to stability for linear test equations with $\Re(\lambda) < 0$.

**Theorem A.2** (Region of stability for the McCallum-Foster method). *Let $\boldsymbol{\Phi}$ be given by an explicit Runge-Kutta solver. Then the reversible numerical solution $\{\boldsymbol{x}_n, \hat{\boldsymbol{x}}_n\}_{n \in \mathbb{N}}$ given by Equation (5) is linearly stable iff*

$$|\Gamma| < 1 + \zeta, \tag{28}$$

*where*

$$\Gamma = 1 + \zeta - (1 - \zeta)R(-h\lambda) - R(-h\lambda)R(h\lambda). \tag{29}$$

**Remark A.10.** The McCallum-Foster method when constructed from explicit Runge-Kutta methods have a *non-trivial* region of stability. Note, however, that this region of stability is smaller than the original region of stability from the original Runga-Kutta method.

### A.3 EXACT INVERSION OF DIFFUSION MODELS

Independent of the work on reversible solvers for neural ODEs several researchers have developed reversible methods for solving the probability flow ODE—often in the literature on diffusion models this is called the *exact inversion* of diffusion models.

#### A.3.1 EDICT SAMPLER

The first work to explore this topic of exact inversion with diffusion models was that of Wallace et al. (2023), who inspired by coupling layers in normalizing flows (Dinh et al., 2015) proposed a reversible solver which they refer to as *exact diffusion inversion via coupled transformations* (EDICT). Like all reversible solvers this method keeps track of an extra state, denoted by $\{\boldsymbol{y}_n\}_{n \in \mathbb{N}}$, with $\boldsymbol{y}_0 = \boldsymbol{x}_0$. Letting $a_n = \frac{\alpha_{n+1}}{\alpha_n}$ and $b_n = \sigma_{n+1} - \frac{\alpha_{n+1}}{\alpha_n}\sigma_n$, this numerical scheme can be described as

$$
\begin{aligned}
\boldsymbol{x}_n^{\text{inter}} &= a_n \boldsymbol{x}_n + b_n \boldsymbol{x}_{T|t_n}^\theta(\boldsymbol{y}_n), \\
\boldsymbol{y}_n^{\text{inter}} &= a_n \boldsymbol{y}_n + b_n \boldsymbol{x}_{T|t_n}^\theta(\boldsymbol{x}_n^{\text{inter}}), \\
\boldsymbol{x}_{n+1} &= \xi \boldsymbol{x}_n^{\text{inter}} + (1 - \xi)\boldsymbol{y}_n^{\text{inter}} \\
\boldsymbol{y}_{n+1} &= \xi \boldsymbol{x}_n^{\text{inter}} + (1 - \xi)\boldsymbol{x}_{n+1},
\end{aligned}
\tag{30}
$$

where $\xi \in (0, 1)$ is a mixing parameter.[11] This method can be inverted to obtain a closed form expression for backward step:

$$
\begin{aligned}
\boldsymbol{y}_n^{\text{inter}} &= \frac{\boldsymbol{y}_{n+1} - (1 - \xi)\boldsymbol{x}_{n+1}}{\xi}, \\
\boldsymbol{x}_n^{\text{inter}} &= \frac{\boldsymbol{y}_{n+1} - (1 - \xi)\boldsymbol{y}_n^{\text{inter}}}{\xi}, \\
\boldsymbol{y}_n &= \frac{\boldsymbol{y}_n^{\text{inter}} - b_n \boldsymbol{x}_{T|t_n}^\theta(\boldsymbol{x}_n^{\text{inter}})}{a_n}, \\
\boldsymbol{x}_n &= \frac{\boldsymbol{x}_n^{\text{inter}} - b_n \boldsymbol{x}_{T|t_n}^\theta(\boldsymbol{y}_n)}{a_n}.
\end{aligned}
\tag{31}
$$

Notably, the EDICT solver was developed in the context of discrete-time diffusion models and the connection to reversible solvers for ODEs was not considered in the original work. *N.B.*, to the best of our knowledge our work is the first to draw the connection between the work on reversible ODE solvers and exact inversion with diffusion models. Unfortunately, this method suffers from poor convergence issues (see Remark A.11) and generally has poor performance when used to perform sampling with diffusion models, thereby limiting its utility in practice (Zhang et al., 2024; Wang et al., 2024).

**Remark A.11.** Later work by Wang et al. (2024, Proposition 6) showed that EDICT is actually a zero-order method, *i.e.*, the local truncation error is $\mathcal{O}(h)$, making it generally unsuitable in practice.

---

[11]In practice, when used for image editing the authors found that the parameter $\xi$ controlled how closely the EDICT sampler aligned with the original sample, with lower values corresponding to higher agreement with the original sample.

### A.3.2 BDIA SAMPLER

Later work by Zhang et al. (2024) proposed a reversible solver for the probability flow ODE which they call *bidirectional integration approximation* (BDIA). The core idea is to use both single-step methods $\boldsymbol{\Phi}_{t_n,t_{n-1}}$ and $\boldsymbol{\Phi}_{t_n,t_{n+1}}$ to induce reversibility.[12] Then using these two approximations—both of which are computed from a discretization centered around $\boldsymbol{x}_n$—the process is update via a multistep process with a forward step of[13]

$$\boldsymbol{x}_{n+1} = \boldsymbol{x}_{n-1} - \boldsymbol{\Phi}_{t_n,t_{n-1}}(\boldsymbol{x}_n) + \boldsymbol{\Phi}_{t_n,t_{n+1}}(\boldsymbol{x}_n). \tag{32}$$

The backwards step can easily be expressed as

$$\boldsymbol{x}_{n-1} = \boldsymbol{x}_{n+1} + \boldsymbol{\Phi}_{t_n,t_{n-1}}(\boldsymbol{x}_n) + \boldsymbol{\Phi}_{t_n,t_{n+1}}(\boldsymbol{x}_n). \tag{33}$$

In practice, BDIA uses the DDIM solver (*i.e.*, Euler) for $\boldsymbol{\Phi}$, but in theory one could use a higher-order method—this was not explored in Zhang et al. (2024).

**Proposition A.3** (BDIA is the leapfrog/midpoint method). *The BDIA method described in Equation* (32) *is the leapfrog/midpoint method when* $\boldsymbol{\Phi}_h(t, \boldsymbol{x}) = h\boldsymbol{u}_t^\theta(\boldsymbol{x})$, *i.e., the Euler step.*

*Proof.* This can be shown rather straightforwardly by substitution, *i.e.*,

$$\boldsymbol{x}_{n+1} = \boldsymbol{x}_{n-1} + 2h\boldsymbol{u}_{t_n}^\theta(\boldsymbol{x}_n). \tag{34}$$

$\square$

**Corollary A.3.1** (BDIA is a first-order method). *BDIA is first-order method,* i.e.*, the local truncation error is* $\mathcal{O}(h^2)$.

**Remark A.12.** This result was also observed in Wang et al. (2024, Proposition 6).

**Corollary A.3.2** (BDIA is nowhere linearly stable). *BDIA is nowhere linearly stable,* i.e.*, the region of stability is the complex interval* $[-i, i]$.

*Proof.* This follows straightforwardly from Proposition A.3 and Shampine (2009, Section 2). $\square$

Zhang et al. (2024) introduce a hyperparameter $\gamma \in [0, 1]$ which is used below

$$\hat{\boldsymbol{\Phi}}_{t_n,t_{n-1}}(\boldsymbol{x}_n) = (1-\gamma)(\boldsymbol{x}_{n-1} - \boldsymbol{x}_n) + \gamma\boldsymbol{\Phi}_{t_n,t_{n-1}}(\boldsymbol{x}_n), \tag{35}$$

to modify the BDIA update rule in Equation (32). Thus, $\gamma$ can be viewed as a parameter which interpolates between the midpoint and Euler schemes. For image editing applications the authors found this parameter to control how closely the BDIA sampler aligned with the original image, with lower values corresponding to higher agreement with the original image (making it similar to the $\xi$ parameter from BDIA).

### A.3.3 BELM SAMPLER

Recently, Wang et al. (2024) proposed a linear multi-step reversible solver for the probability flow ODE called the *bidirectional explicit linear multi-step* (BELM) sampler. First, they reparameterize the probability flow ODE as

$$\mathrm{d}\overline{\boldsymbol{x}}(t) = \overline{\boldsymbol{x}}_{T|\overline{\sigma}_t}^\theta(\overline{\boldsymbol{x}}(t))\,\mathrm{d}\overline{\sigma}_t, \tag{36}$$

where $\overline{\boldsymbol{x}}(t) \coloneqq \boldsymbol{x}(t)/\alpha_t$, $\overline{\sigma}(t) \coloneqq \sigma_t/\alpha_t$, and $\overline{\boldsymbol{x}}_{T|\overline{\sigma}_t}^\theta(\overline{\boldsymbol{x}}(t)) = \boldsymbol{x}_{T|t}^\theta(\boldsymbol{x}(t))$.[14] The BELM sampler makes use of the variable-stepsize-variable-formula (VSVF) linear multi-step methods (Crouzeix

---

[12]*N.B.*, in the original paper, Zhang et al. (2024) use quite different notation for explaining their idea; however, we find our presentation to be simpler for the reader as it more easily enables comparison to other methods.

[13]In some sense, this is reminiscent of the idea from the more general McCallum-Foster method; however, this approach results in a multi-step method unlike the single-step method of McCallum & Foster (2024).

[14]*N.B.*, this is a popular parameterization of diffusion models and affine conditional flows. This can be done *mutatis mutandis* for target prediction models retrieving (Blasingame & Liu, 2025, Proposition D.2).

& Lisbona, 1984) to construct the numerical solver. The $k$-step VSVF linear multi-step method for solving the reparameterized probability flow ODE in Equation (36) is given by

$$\overline{\boldsymbol{x}}_{n+1} = \sum_{m=1}^{k} a_{n,m} \overline{\boldsymbol{x}}_{n+1-m} \tag{37}$$

$$+ \sum_{m=1}^{k-1} b_{n,m} h_{n+1-m} \overline{\boldsymbol{x}}^{\theta}_{T|\overline{\sigma}_{n+1-m}} (\overline{\boldsymbol{x}}_{n+1-m}). \tag{38}$$

where $a_{n,m} \neq 0,$[15] and $b_{n,m}$ are coefficients chosen using dynamic multi-step formulæ to find the coefficients (Crouzeix & Lisbona, 1984); and $h_n$ are step sizes chosen beforehand. This scheme can be reversed via the backward step

$$\overline{\boldsymbol{x}}_{n+1-k} = \frac{1}{a_{n,k}} \overline{\boldsymbol{x}}_{n+1} - \sum_{m=1}^{k-1} \frac{a_{n,m}}{a_{n,k}} \overline{\boldsymbol{x}}_{n+1-m} \tag{39}$$

$$- \sum_{m=1}^{k-1} \frac{b_{n,m}}{a_{n,k}} h_{n+1-m} \overline{\boldsymbol{x}}^{\theta}_{T|\overline{\sigma}_{n+1-m}} (\overline{\boldsymbol{x}}_{n+1-m}). \tag{40}$$

**Remark A.13.** The BELM samplers require $k-1$ extra to be stored in memory in order to be reversible. In contrast, McCallum & Foster (2024) only requires storing one extra states, irregardless of the desired convergence order. Additionally, poor stability is a concern with such linear multi-step methods (see Kidger, 2022, Remark 5.24).

**Remark A.14.** Interestingly, the earlier EDICT and BDIA methods can be viewed as instances of the BELM method (Wang et al., 2024, Appendicies A.7 and A.8).

By solving the multi-step formulæ to minimize the local truncation error Wang et al. (2024) propose an instance of the BELM solver which they refer to as *O-BELM* defined as[16]

$$\overline{\boldsymbol{x}}_{n+1} = \frac{h_n^2}{h_{n-1}^2} \overline{\boldsymbol{x}}_{n-1} + \frac{h_{n-1}^2 + h_n^2}{h_{n-1}^2} \overline{\boldsymbol{x}}_n - \frac{h_n(h_n + h_{n+1})}{h_{n+1}} \overline{\boldsymbol{x}}_{0|\overline{\sigma}_n}(\overline{\boldsymbol{x}}_n). \tag{41}$$

Notably, the O-BELM sampler can also be viewed as instance of the leapfrog/midpoint method.

**Theorem A.4** (O-BELM is the leapfrog/midpoint method)**.** *Fix a step size $h_n = h$ for all $n$, then O-BELM is the leapfrog/midpoint method.*

*Proof.* This follows from substitution of $h_n = h$. □

**Corollary A.4.1** (O-BELM is nowhere linearly stable)**.** *Fix a step size $h_n = h$, then O-BELM is nowhere linearly stable, i.e., the region of stability is the complex interval $[-i, i]$.*

### A.3.4 CYCLEDIFFUSION

To our knowledge, the *only* other work to propose exact inversion with the SDE formulation of the diffusion models is the work of Wu & la Torre (2023). However, there a *several* noticeable distinctions, the largest being that they store the entire solution trajectory in memory. Given a particular realization of the Wiener process that admits $\boldsymbol{x}_t \sim \mathcal{N}(\alpha_t \boldsymbol{x}_0 \mid \sigma_t^2 \mathbf{I})$, then given $\boldsymbol{x}_s$ and noise $\boldsymbol{\epsilon}_s \sim \mathcal{N}(\mathbf{0}, \mathbf{I})$ we can calculate

$$\boldsymbol{x}_t = \frac{\alpha_t}{\alpha_s} \boldsymbol{x}_s + 2\sigma_t(e^h - 1)\hat{\boldsymbol{x}}_{T|s}(\boldsymbol{x}_s) + \sigma_t \sqrt{e^{2h} - 1} \boldsymbol{\epsilon}_s. \tag{42}$$

Wu & la Torre (2023) propose to invert this by first calculating, for two samples $\boldsymbol{x}_t$ and $\boldsymbol{x}_s$, the noise $\boldsymbol{\epsilon}_s$. This can be calculated by rearranging the previous equation to find

$$\boldsymbol{\epsilon}_s = \frac{\boldsymbol{x}_t - \frac{\alpha_t}{\alpha_s} \boldsymbol{x}_s + 2\sigma_t(e^h - 1)\boldsymbol{\epsilon}_\theta(\boldsymbol{x}_s, \boldsymbol{z}, s)}{\sigma_t \sqrt{e^{2h} - 1}} \tag{43}$$

With this the sequence $\{\boldsymbol{\epsilon}_{t_i}\}_{i=1}^{N}$ of added noises can be calculated which can be used to reconstruct the original input from the initial realization of the Wiener process. However, unlike our approach, this process requires storing the entire realization in memory.

---

[15]This is to ensure that the method is reversible.

[16]*N.B.*, the original equation in Wang et al. (2024, Equation (18)) had a sign difference for the coefficient of $b_{i,1}$; however, this is due to differences in convention in handling integration in reverse-time.

Table 3: Comparison of different (non-symplectic) reversible ODE solvers. We note that some of the solvers were developed particularly for the probability flow ODE (an affine conditional flow) whilst others work for general ODEs. In the first column we denote the number of extra states the numerical scheme needs to keep in memory to ensure algebraic reversibility. For BELM $k$ denotes the number of steps and for McCallum-Foster $k$ denotes the convergence order of the underlying single-step solver. For the column labeled *region of linear stability* we mean there exists some subset of $\mathbb{C}$ which is the region of stability and the set is not a null set. The proof of convergence for BELM is only provided for the special case (called *O-BELM* in Wang et al. (2024)) with $k = 2$.

| Solver | Number of extra states | Local truncation error | Region of linear stability | Proof of convergence |
|---|---|---|---|---|
| **Probability flow ODEs** | | | | |
| EDICT | 1 | $\mathcal{O}(h)$ | ✗ | ✗ |
| BDIA | 1 | $\mathcal{O}(h^2)$ | ✗ | ✗ |
| BELM | $k-1$ | $\mathcal{O}(h^{k+1})$ | ✗ | $\sim$ |
| Rex | 1 | $\mathcal{O}(h^{k+1})$ | ✓ | ✓ |
| **General ODEs** | | | | |
| Asynchronous leapfrog | 1 | $\mathcal{O}(h^3)$ | ✗ | ✓ |
| Reversible Heun | 1 | $\mathcal{O}(h^3)$ | ✗ | ✓ |
| McCallum-Foster | 1 | $\mathcal{O}(h^{k+1})$ | ✓ | ✓ |

### A.3.5 SUMMARY

We present a summary of related works on either *exact inversion* or *reversible solvers* below in Table 3. *N.B.*, we omit *CycleDiffusion* because it is more orthogonal to the general concept of a reversible solver and is only reversible in the trivial sense.

### A.4 SDE SOLVERS FOR DIFFUSION MODELS

Next we discuss related works on SDE solvers for the reverse-time diffusion SDE in Equation (3). Now there are numerous *stochastic Runge-Kutta* (SRK) methods in the literature all tailor to specific types of SDEs, which we can distinguish by the their strong order of convergence (see Definition D.1) and strong order conditions. For example the classic Euler-Maruyama scheme (Kloeden & Platen, 1992) has strong order of convergence of 0.5 and was straightforwardly applied to the reverse-time diffusion SDE in Jolicoeur-Martineau et al. (2021) as a baseline. Song et al. (2021b) proposed an ancestral sampling scheme for a discretization of the forward-time diffusion SDE in Equation (1) with additional Langevin dynamics; likewise, the DDIM solver from Song et al. (2021a) can be viewed a sort of Euler-Maruyama scheme. Other classic SDE schemes like SRA1/SRA2/SRA3 schemes (Rößler, 2010) all have strong order of convergence 1.5 for additive noise SDEs and were tested for diffusion models in Jolicoeur-Martineau et al. (2021).

More recently, researchers have explored exponential solvers for SDEs, *e.g.*, the exponential Euler-Maruyama method (Komori et al., 2017) and the *stochastic Runge-Kutta Lawson* (SRKL) schemes (Debrabant et al., 2021). From an initial inspection the SRKL schemes of Debrabant et al. (2021, Algorithm 1) is somewhat similar to our method for constructing $\boldsymbol{\Psi}$; however, upon closer inspection they are some key fundamental differences.[17] The largest of these is how the underlying SRK schemes are represented. In particular the SRKL schemes choose to follow the conventions of Burrage & Burrage (2000) (for Stratonovich SDEs) in constructing the underlying SRK schemes; whereas we follow the SRK schemes outlined by Foster et al. (2024) (*cf*. Appendix B). These differences stem from how one chooses to handle the the iterated stochastic integrals from the Stratonovich-Taylor (or Itô-Taylor) expansions.

---

[17]*N.B.*, in general Debrabant et al. (2021) consider full stochastic Lawson schemes where the integrating factor is a stochastic process given by the matrix exponential applied to linear terms in the drift and diffusion coefficients; conversely, the drift stochastic Lawson schemes are more similar to what we study.

### A.4.1 COMPARISON WITH SEEDS

Mostly directly relevant to our work on constructing a stochastic $\mathbf{\Psi}$ is the SEEDS family of solvers proposed by Gonzalez et al. (2024). Similar to us, they also approach using exponential methods to simplify the expression of diffusion models Gonzalez et al. (2024, Appendix B.1). There are two *key* distinctions, namely, 1) that they use the *stochastic exponential time differencing* (SETD) method (Adamu, 2011), whereas, we construct stochastic Lawson schemes;[18] and 2) that they use a different technique for modeling the iterated stochastic integrals for high-order solvers. In particular, SEEDS introduces a decomposition for the iterated stochastic integrals produced by the Itô-Taylor expansions of Equation (3) such that the decomposition preserves the Markov property, *i.e.*, the random variables used to construct model the Brownian increments from iterated integrals are independent on non-overlapping intervals and dependent on overlapping intervals (see Gonzalez et al., 2024, Proposition 4.3). By making use of the SRK schemes of Foster et al. (2024) developed from using the space-time Lévy area to construct high-order splitting methods we have an alternative method for ensuring this property. This results in our solver based on ShARK (see Appendix B.3, *cf*. Theorem 4.2) having a strong order of convergence of 1.5; whereas, SEEDS-3 only achieves a *weak* order of convergence of 1.

This brings us to another large difference, the SEEDS solvers focus on the *weak* approximation to Equation (3); whereas, as we are concerned with the *strong* approximation to Equation (3). The difference between these two is that the weak convergence is considered with the precisions of the *moments*; whereas, strong convergence is concerned with the precision of the *path*. Moreover, by definition a strong order of convergence implies a weak order of convergence, the converse is not true. In particular, for our application of developing *reversible* schemes this strong order of convergence is particularly important as we care about the path. Thus the technique SEEDS uses to replace iterated Itô integrals with other random variables with equivalent moment conditions is *wholly unsuitable* for our purposes as we desire a *strong* approximation.

## B STOCHASTIC RUNGE-KUTTA METHODS

Recall that the general Butcher tableau for a $s$-stage explicit RK scheme (Stewart, 2022, Section 6.1.4) for a generic ODE is written as

$$
\begin{array}{c|ccccc}
c_1 & & & & & \\
c_2 & a_{21} & & & & \\
c_3 & a_{31} & a_{32} & & & \\
\vdots & \vdots & \vdots & \ddots & & \\
c_s & a_{s1} & a_{s2} & \cdots & a_{s(s-1)} & \\
\hline
& b_1 & b_2 & \cdots & b_{s-1} & b_s
\end{array}
= \frac{c\;\;\vert\;\; a}{b}. \tag{44}
$$

*E.g.*, the famous 4-th order Runge-Kutta (RK4) method is given by

$$
\begin{array}{c|cccc}
0 & & & & \\
\frac{1}{2} & \frac{1}{2} & & & \\
\frac{1}{2} & 0 & \frac{1}{2} & & \\
1 & 0 & 0 & 1 & \\
\hline
& \frac{1}{6} & \frac{1}{3} & \frac{1}{3} & \frac{1}{6}
\end{array}
. \tag{45}
$$

However, for SDEs this is much trickier due to the presense of iterated stochastic integrals in the Itô-Taylor or Stratonovich-Taylor expansions (Kloeden & Platen, 1992). Consider a $d$-dimensional *Stratonovich* SDE driven by $d_w$-dimensional Brownian motion $\{\boldsymbol{W}_t\}_{t \in [0,T]}$ defined as

$$
\mathrm{d}\boldsymbol{X}_t = \boldsymbol{\mu}_\theta(t, \boldsymbol{X}_t)\,\mathrm{d}t + \boldsymbol{\sigma}_\theta(t, \boldsymbol{X}_t) \circ \mathrm{d}\boldsymbol{W}_t, \tag{46}
$$

---

[18]*N.B.*, for certain scenarios these two different viewpoints converge, particularly, in the deterministic case. See our discussion on the family of DPM-Solvers which also use (S)ETD in Appendix E.

where $\boldsymbol{\mu}_\theta \in \mathcal{C}^2(\mathbb{R} \times \mathbb{R}^d; \mathbb{R}^d)$ and $\boldsymbol{\sigma}_\theta \in \mathcal{C}^3(\mathbb{R} \times \mathbb{R}^d; \mathbb{R}^{d \times d_w})$ satisfy the usual regularity conditions for Stratonovich SDEs (Øksendal, 2003, Theorem 5.2.1) and where $\circ \mathrm{d} \boldsymbol{W}_t$ denotes integration in the Stratonovich sense.

Rößler (2025) write one such class of an $s$-stage explicit SRK methods (*cf.* Burrage & Burrage, 2000; Rößler, 2010) for Equation (46) as

$$\boldsymbol{Z}_i^{(0)} = \boldsymbol{X}_n + h \sum_{j=1}^{i-1} a_{ij}^{(0)} \boldsymbol{\mu}_\theta(t_n + c_j^{(0)}, \boldsymbol{Z}_j^{(0)}),$$

$$\boldsymbol{Z}_i^{(k)} = \boldsymbol{X}_n + h \sum_{j=1}^{i-1} a_{ij}^{(1)} \boldsymbol{\mu}_\theta(t_n + c_j^{(0)}, \boldsymbol{Z}_j^{(0)}) + \sum_{j=1}^{i-1} \sum_{l=1}^{d_w} a_{ij}^{(2)} \boldsymbol{I}_{(l,k),n} \boldsymbol{\sigma}_\theta(t_n + c_j^{(1)}, \boldsymbol{Z}_i^{(l)}),$$

$$\boldsymbol{X}_{n+1} = \boldsymbol{X}_n + h \sum_{i=1}^{s} b_i^{(0)} \boldsymbol{\mu}_\theta(t_n + c_i^{(0)}, \boldsymbol{Z}_j^{(0)}) + \sum_{i=1}^{s} \sum_{k=1}^{d_w} \left( b_i^{(1)} \boldsymbol{I}_{(k),n} + b_i^{(2)} \right) \boldsymbol{\sigma}_\theta(t_n + c_j^{(1)}, \boldsymbol{Z}_i^{(k)}),$$

$$\tag{47}$$

for $k = 1, \ldots, d_w$ and where

$$\boldsymbol{I}_{(k),n} = \int_{t_n}^{t_{n+1}} \circ \mathrm{d} \boldsymbol{W}_u^k = \boldsymbol{W}_{t_{n+1}}^k - \boldsymbol{W}_{t_n}^k, \tag{48}$$

$$\boldsymbol{I}_{(l,k),n} = \int_{t_n}^{t_{n+1}} \int_{t_n}^{u} \circ \mathrm{d} \boldsymbol{W}_v^l \circ \mathrm{d} \boldsymbol{W}_u^k, \tag{49}$$

let $\hat{\boldsymbol{I}}$ denote the iterated integrals for the Itô case *mutatis mutandis*. This scheme is described the by the *extended* Butcher tableau (Rößler, 2025)

$$\begin{array}{c|c|c}
c^{(0)} & a^{(0)} & \\
\hline
c^{(1)} & a^{(1)} & a^{(2)} \\
\hline
& b^{(0)} & b^{(1)} \quad b^{(2)}
\end{array}. \tag{50}$$

These iterated integrals $\boldsymbol{I}_{(l,k),n}$ are very tricky to work with and can raise up many practical concerns. As alluded to earlier (*cf.* Section A.4.1) it is common to use a weak approximation of such integrals via a random variables with corresponding moments. This results in two drawbacks: 1) the resulting SDE scheme only converges in the *weak* sense and 2) the solution yielding by the scheme is not a Markov chain in general. SEEDS overcomes the second issue by using a special decomposition to preserve the Markov property, see the ablations in Gonzalez et al. (2024) for more details on this topic in practice.

### B.1 FOSTER-REIS-STRANGE SRK SCHEME

Conversely, Foster et al. (2024) propose another SRK scheme based on higher-order splitting methods for Stratonovich SDEs. For the Stratonovich SDE in Equation (46) Foster et al. (2024) write an $s$-stage SRK as

$$\boldsymbol{\mu}_\theta^i = \boldsymbol{\mu}_\theta(t_n + c_i h, \boldsymbol{Z}_i),$$

$$\boldsymbol{\sigma}_\theta^i = \boldsymbol{\sigma}_\theta(t_n + c_i h, \boldsymbol{Z}_i),$$

$$\boldsymbol{Z}_i = \boldsymbol{X}_n + h \left( \sum_{j=1}^{i-1} a_{ij} \boldsymbol{\mu}_\theta^j \right) + \boldsymbol{W}_n \left( \sum_{j=1}^{i-1} a_{ij}^W \boldsymbol{\sigma}_\theta^j \right) + \boldsymbol{H}_n \left( \sum_{j=1}^{i-1} a_{ij}^H \boldsymbol{\sigma}_\theta^j \right), \tag{51}$$

$$\boldsymbol{X}_{n+1} = \boldsymbol{X}_n + h \left( \sum_{i=1}^{s} b_i \boldsymbol{\mu}_\theta^i \right) + \boldsymbol{W}_n \left( \sum_{i=1}^{s} b_i^W \boldsymbol{\sigma}_\theta^i \right) + \boldsymbol{H}_n \left( \sum_{i=1}^{s} b_i^H \boldsymbol{\sigma}_\theta^i \right),$$

where $h = t_{n+1} - t_n$ is the step size and $\boldsymbol{W}_n := \boldsymbol{W}_{t_n, t_{n+1}}$ and $\boldsymbol{H}_n := \boldsymbol{H}_{t_n, t_{n+1}}$ are the Brownian and space-time Lévy increments (*cf.* Definition 3.2) respectively; and where $a_{ij}, a_{ij}^W, a_{ij}^H \in \mathbb{R}^{s \times s}$, $b_i, b_i^W, b_i^H \in \mathbb{R}^s$, and $c_i \in \mathbb{R}^s$ for the coefficients for an *extended* Butcher tableau (Foster et al., 2024) which is given as

$$\begin{array}{c|c|c|c}
c & a & a^W & a^H \\
\hline
& b & b^W & b^H
\end{array}. \tag{52}$$

*E.g.*, we can write the famous Euler-Maruyama scheme as

$$
\begin{array}{c|cc|c}
0 & 0 & 0 & 0 \\
\hline
 & 1 & 1 & 0
\end{array}.
\tag{53}
$$

## B.2 INDEPENDENCE OF THE BROWNIAN AND LÉVY INCREMENTS

Remarkably, in Foster et al. (2020, Theorem 2.2) present a polynomial Karhunen-Loève theorem for the Brownian bridge (*cf*. Definition G.1)—picture an stochastic analogue to the Fourier series of a function on a bounded interval—which leads to a most useful remark (Foster et al., 2020, Remark 3.6) which we restate below.

**Remark B.1.** We have that $H_{s,t} \sim \mathcal{N}(0, \frac{1}{12}h)$ is independent of $W_{s,t}$ when $d = 1$, likewise, since the coordinate processes of a Brownian motion are independent, one can write $\boldsymbol{W}_{s,t} \sim (\boldsymbol{0}, h\boldsymbol{I})$ and $\boldsymbol{H}_{s,t} \sim \mathcal{N}(\boldsymbol{0}, \frac{1}{12}h\boldsymbol{I})$ are independent.

Thus we have found another remedy to the problem of independent increments, whilst still being able to obtain a *strong* approximation of the SDE.

## B.3 SHARK

Recently, Foster et al. (2024) developed *shifted additive-noise Runge-Kutta* (ShARK) for additive noise SDEs which is based on Foster et al. (2024, Equation (6.1)). This scheme has converges strongly with order 1.5 for additive-noise SDEs and makes two evaluations of the drift and diffusion per step.

ShARK is described via the following extended Butcher tableau

$$
\begin{array}{c|cc|c|c}
0 & & & 0 & 1 \\
\frac{5}{6} & \frac{5}{6} & & \frac{5}{6} & 1 \\
\hline
 & 0.4 & 0.6 & 1 & 0 \\
 & -0.6 & 0.6 & &
\end{array}.
\tag{54}
$$

The second row for the $b$ variable describes the coefficients used for adaptive-step size solvers to approximate the error at each step. The Butcher tableau for this scheme can be found here: `https://github.com/patrick-kidger/diffrax/blob/main/diffrax/_solver/shark.py`.

# C DERIVATION OF REX

We derive the Rex scheme presented in Proposition 3.3 in the main paper.

## C.1 REX (ODE)

In this section we derive the Rex scheme for the probability flow ODE. We present derivations for both the data prediction and noise prediction formulations.

### C.1.1 PROOF OF PROPOSITION 3.1

We restate Proposition 3.1 below.

**Proposition 3.1** (Reparameterization of the probability flow ODE)**.** *The probability flow ODE in Equation* (7) *can be rewritten in $\varsigma_t$ as*

$$
\frac{\mathrm{d}\boldsymbol{y}_\varsigma}{\mathrm{d}\varsigma} = \beta_T \boldsymbol{f}_\theta \left( \varsigma, \frac{\beta_\varsigma}{\beta_\varsigma} \boldsymbol{y}_\varsigma \right),
\tag{8}
$$

*where $\boldsymbol{y}_t = \frac{\beta_T}{\beta_t} \boldsymbol{x}_t$.*

*Proof.* Recall that from Equation (7) we have that the ODE is given by

$$\frac{\mathrm{d}\boldsymbol{x}_t}{\mathrm{d}t} = \frac{\dot{\beta}_t}{\beta_t}\boldsymbol{x}_t + \frac{\sigma_t\dot{\alpha}_t - \dot{\sigma}_t\alpha_t}{\beta_t}\boldsymbol{f}_\theta(t, \boldsymbol{x}_t). \tag{55}$$

We can use the technique of exponential integrators to rewrite the ODE as

$$\frac{\mathrm{d}}{\mathrm{d}t}\left[e^{\int_T^t -\frac{\dot{\beta}_u}{\beta_u}\,\mathrm{d}u}\boldsymbol{x}_t\right] = e^{\int_T^t -\frac{\dot{\beta}_u}{\beta_u}\,\mathrm{d}u}\frac{\sigma_t\dot{\alpha}_t - \dot{\sigma}_t\alpha_t}{\beta_t}\boldsymbol{f}_\theta(t, \boldsymbol{x}_t), \tag{56}$$

recalling that we integrate from initial time $T$ in reverse-time. Then the exponential terms simplify to

$$e^{\int_T^t -\frac{\dot{\beta}_u}{\beta_u}\,\mathrm{d}u} = \frac{\beta_0}{\beta_T}. \tag{57}$$

We introduce a *change-of-variables* $\boldsymbol{y}_t = \frac{\beta_0}{\beta_T}\boldsymbol{x}_t$ to rewrite the ODE as

$$\frac{\mathrm{d}\boldsymbol{y}_t}{\mathrm{d}t} = \underbrace{\frac{\beta_T}{\beta_t}\frac{\sigma_t\dot{\alpha}_t - \dot{\sigma}_t\alpha_t}{\beta_t}}_{=\kappa_t}\boldsymbol{f}_\theta\left(t, \frac{\beta_T}{\beta_t}\boldsymbol{y}_t\right). \tag{58}$$

Next we define

$$\dot{\varsigma}_t = \mathrm{sgn}(\beta_T)\frac{\sigma_t\dot{\alpha}_t - \sigma\dot{\alpha}_t}{\beta_t^2}, \tag{59}$$

which we will now justify. Now recall that $\beta_t$ is either $-\alpha_t$ or $\sigma_t$ depending on the whether $\boldsymbol{f}_\theta$ denotes the data or noise prediction model. Moreover we know that $\alpha_t$ is a strictly monotonically decreasing in $t$ and that $\sigma_t$ is a strictly monotonically increasing in $t$. We will now prove that there exists and inverse function for $\varsigma_t$ such that $t_\varsigma(\varsigma_t) = t$ for both cases.

**Case $\beta_t = -\alpha_t$.** We can write $\kappa_t$ as

$$\kappa_t = \alpha_T\frac{\dot{\sigma}_t\alpha_t - \sigma_t\dot{\alpha}_t}{\alpha_t^2}, \tag{60}$$

$$\overset{(i)}{=} \alpha_T\frac{\mathrm{d}}{\mathrm{d}t}\left(\frac{\sigma_t}{\alpha_t}\right), \tag{61}$$

where (i) holds by the quotient role. Clearly, we have that

$$\dot{\varsigma}_t = \frac{\mathrm{d}}{\mathrm{d}t}\left(\frac{\sigma_t}{\alpha_t}\right), \tag{62}$$

$$\varsigma_t = \frac{\sigma_t}{\alpha_t}, \tag{63}$$

It follows from $(\alpha_t, \sigma_t)$ that $\varsigma_t$ is strictly monotonically increasing in $t$ and thus we can construct its inverse.

**Case $\beta_t = \sigma_t$.** We can write $\kappa_t$ as

$$\kappa_t = \sigma_T\frac{\sigma_t\dot{\alpha}_t - \dot{\sigma}_t\alpha_t}{\sigma_t^2}, \tag{64}$$

$$\overset{(i)}{=} \sigma_T\frac{\mathrm{d}}{\mathrm{d}t}\left(\frac{\alpha_t}{\sigma_t}\right), \tag{65}$$

where (i) holds by the quotient role. Clearly, we have that

$$\dot{\varsigma}_t = \frac{\mathrm{d}}{\mathrm{d}t}\left(\frac{\alpha_t}{\sigma_t}\right), \tag{66}$$

$$\varsigma_t = \frac{\alpha_t}{\sigma_t}, \tag{67}$$

It follows from $(\alpha_t, \sigma_t)$ that $\varsigma_t$ is strictly monotonically decreasing in $t$ and thus we can construct its inverse.

Thus we can rewrite the ODE via a time-change to find

$$\frac{\mathrm{d}\boldsymbol{y}_\varsigma}{\mathrm{d}\varsigma} = \beta_0\boldsymbol{f}_\theta\left(\varsigma, \frac{\beta_T}{\beta_\varsigma}\boldsymbol{y}_\varsigma\right), \tag{68}$$

with the usual *abuse-of-notation* $\boldsymbol{y}_\varsigma := \boldsymbol{y}_{t_\varsigma(\varsigma)}$, $\beta_\varsigma := \beta_{t_\varsigma(\varsigma)}$, &c. □

**Remark C.1.** When in the noise prediction formulation with Proposition 3.1 we recover the following reparameterization of Equation (7)

$$\frac{\mathrm{d}\boldsymbol{z}_\chi}{\mathrm{d}\chi} = \alpha_T \boldsymbol{x}^\theta_{T|\chi} \left( \frac{\alpha_\chi}{\alpha_T} \boldsymbol{z}_\chi \right), \tag{69}$$

where $\alpha_T > 0$, $\boldsymbol{z}_t = \frac{\alpha_T}{\alpha_t} \boldsymbol{x}_t$ and $\chi_t = \frac{\sigma_t}{\alpha_t}$, which has been observed by numerous prior works (see Song et al., 2021a, Equation (14); Pan et al., 2023, Equation (11); Wang et al., 2024, Equation (6)).

**Remark C.2.** When in the data prediction formulation, Proposition 3.1 recovers Blasingame & Liu (2025, Proposition D.2) which states that Equation (7) can be written as

$$\frac{\mathrm{d}\boldsymbol{y}_\gamma}{\mathrm{d}\gamma} = \sigma_T \boldsymbol{x}^\theta_{0|\gamma} \left( \frac{\sigma_\gamma}{\sigma_T} \boldsymbol{y}_\gamma \right), \tag{70}$$

where $\boldsymbol{y}_t = \frac{\sigma_T}{\sigma_t} \boldsymbol{x}_t$ and $\gamma_t = \frac{\alpha_t}{\sigma_t}$.

### C.1.2 DATA PREDICTION

We present this derivation in the form of Lemma C.1 below.

**Lemma C.1** (Rex (ODE) for data prediction models). *Let $\boldsymbol{\Phi}$ be an explicit Runge-Kutta solver for the ODE in Equation (70) with Butcher tableau $a_{ij}$, $b_i$, $c_i$. The reversible solver for $\boldsymbol{\Phi}$ in terms of the original state $\boldsymbol{x}_t$ is given by the forward step*

$$\boldsymbol{x}_{n+1} = \frac{\sigma_{n+1}}{\sigma_n} \left( \zeta \boldsymbol{x}_n + (1-\zeta)\hat{\boldsymbol{x}}_n \right) + \sigma_{n+1} \boldsymbol{\Psi}_h(\gamma_n, \hat{\boldsymbol{x}}_n),$$

$$\hat{\boldsymbol{x}}_{n+1} = \frac{\sigma_{n+1}}{\sigma_n} \hat{\boldsymbol{x}}_n - \sigma_{n+1} \boldsymbol{\Psi}_{-h}(\gamma_{n+1}, \boldsymbol{x}_{n+1}), \tag{71}$$

*and backward step*

$$\hat{\boldsymbol{x}}_n = \frac{\sigma_n}{\sigma_{n+1}} \hat{\boldsymbol{x}}_{n+1} + \sigma_n \boldsymbol{\Psi}_{-h}(\gamma_{n+1}, \boldsymbol{x}_{n+1}),$$

$$\boldsymbol{x}_n = \frac{\sigma_n}{\sigma_{n+1}} \zeta^{-1} \boldsymbol{x}_{n+1} + (1-\zeta^{-1})\hat{\boldsymbol{x}}_n - \sigma_n \zeta^{-1} \boldsymbol{\Psi}_h(\gamma_n, \hat{\boldsymbol{x}}_n), \tag{72}$$

*with step size $h \coloneqq \gamma_{n+1} - \gamma_n$ and where $\boldsymbol{\Psi}$ denotes the following scheme*

$$\hat{\boldsymbol{z}}_i = \frac{1}{\sigma_n} \boldsymbol{x}_n + h \sum_{j=1}^{i-1} a_{ij} \boldsymbol{x}^\theta_{0|\gamma_n + c_j h}(\sigma_{\gamma_n + c_j h} \hat{\boldsymbol{z}}_j),$$

$$\boldsymbol{\Psi}_h(\gamma_n, \boldsymbol{x}_n) = h \sum_{i=1}^{s} b_i \boldsymbol{x}^\theta_{0|\gamma_n + c_i h}(\sigma_{\gamma_n + c_i h} \hat{\boldsymbol{z}}_i), \tag{73}$$

*Proof.* Recall that the forward step of the McCallum-Foster method for Equation (70) given $\boldsymbol{\Phi}$ is given as

$$\boldsymbol{y}_{n+1} = \zeta \boldsymbol{y}_n + (1-\zeta)\hat{\boldsymbol{y}}_n + \boldsymbol{\Phi}_h(\gamma_n, \hat{\boldsymbol{y}}_n),$$

$$\hat{\boldsymbol{y}}_{n+1} = \hat{\boldsymbol{y}}_n - \boldsymbol{\Phi}_{-h}(\gamma_{n+1}, \boldsymbol{y}_{n+1}), \tag{74}$$

with step size $h = \gamma_{n+1} - \gamma_n$. We use the definition of $\boldsymbol{y}_t = \frac{\sigma_T}{\sigma_t} \boldsymbol{x}_t$ to rewrite the forward pass as

$$\boldsymbol{x}_{n+1} = \frac{\sigma_{n+1}}{\sigma_n} \left( \zeta \boldsymbol{x}_n + (1-\zeta)\hat{\boldsymbol{x}}_n \right) + \frac{\sigma_{n+1}}{\sigma_T} \boldsymbol{\Phi}_h \left( \gamma_n, \frac{\sigma_T}{\sigma_n} \hat{\boldsymbol{x}}_n \right),$$

$$\hat{\boldsymbol{x}}_{n+1} = \frac{\sigma_{n+1}}{\sigma_n} \hat{\boldsymbol{x}}_n - \frac{\sigma_{n+1}}{\sigma_T} \boldsymbol{\Phi}_{-h} \left( \gamma_{n+1}, \frac{\sigma_T}{\sigma_{n+1}} \boldsymbol{x}_{n+1} \right). \tag{75}$$

*Mutatis mutandis* we find the backward step in $\boldsymbol{x}_t$ to be given as

$$\hat{\boldsymbol{x}}_n = \frac{\sigma_n}{\sigma_{n+1}} \hat{\boldsymbol{x}}_{n+1} + \frac{\sigma_n}{\sigma_T} \boldsymbol{\Phi}_{-h} \left( \gamma_{n+1}, \frac{\sigma_T}{\sigma_{n+1}} \boldsymbol{x}_{n+1} \right),$$

$$\boldsymbol{x}_n = \frac{\sigma_n}{\sigma_{n+1}} \zeta^{-1} \boldsymbol{x}_{n+1} + (1-\zeta^{-1})\hat{\boldsymbol{x}}_n - \frac{\sigma_n}{\sigma_T} \zeta^{-1} \boldsymbol{\Phi}_h \left( \gamma_n, \frac{\sigma_T}{\sigma_n} \hat{\boldsymbol{x}}_n \right), \tag{76}$$

Next we simplify the explicit RK scheme $\boldsymbol{\Phi}(\gamma_n, \boldsymbol{y}_n)$ for the time-changed probability flow ODE in Equation (70). Recall that the RK scheme can be written as

$$
\boldsymbol{z}_i = \boldsymbol{y}_n + h \sum_{j=1}^{i-1} a_{ij} \sigma_T \boldsymbol{x}_{0|\gamma_n + c_j h} \left( \frac{\sigma_{\gamma_n + c_j h}}{\sigma_T} \boldsymbol{z}_j \right),
$$

$$
\boldsymbol{\Phi}_h(\gamma_n, \boldsymbol{y}_n) = h \sum_{i=1}^{s} b_i \sigma_T \boldsymbol{x}_{0|\gamma_n + c_i h} \left( \frac{\sigma_{\gamma_n + c_i h}}{\sigma_T} \boldsymbol{z}_i \right).
$$

(77)

Next, we replace $\boldsymbol{y}_t$ back with $\boldsymbol{x}_t$ which yields

$$
\boldsymbol{z}_i = \sigma_T \left( \frac{1}{\sigma_n} \boldsymbol{x}_n + h \sum_{j=1}^{i-1} a_{ij} \boldsymbol{x}_{0|\gamma_n + c_j h} \left( \frac{\sigma_{\gamma_n + c_j h}}{\sigma_T} \boldsymbol{z}_j \right) \right),
$$

$$
\boldsymbol{\Phi}_h \left( \gamma_n, \frac{\sigma_T}{\sigma_n} \boldsymbol{x}_n \right) = \sigma_T h \sum_{i=1}^{s} b_i \boldsymbol{x}_{0|\gamma_n + c_i h} \left( \frac{\sigma_{\gamma_n + c_i h}}{\sigma_T} \boldsymbol{z}_i \right).
$$

(78)

To further simplify let $\sigma_T \hat{\boldsymbol{z}}_i = \boldsymbol{z}_i$ and define $\boldsymbol{\Psi}_h(\gamma_n, \boldsymbol{x}_n) := \sigma_T \boldsymbol{\Phi}(\gamma_n, \frac{\sigma_T}{\sigma_n} \boldsymbol{x}_n)$.

Thus we can write the following reversible scheme with forward step

$$
\boldsymbol{x}_{n+1} = \frac{\sigma_{n+1}}{\sigma_n} (\zeta \boldsymbol{x}_n + (1 - \zeta) \hat{\boldsymbol{x}}_n) + \sigma_{n+1} \boldsymbol{\Psi}_h(\gamma_n, \hat{\boldsymbol{x}}_n),
$$

$$
\hat{\boldsymbol{x}}_{n+1} = \frac{\sigma_{n+1}}{\sigma_n} \hat{\boldsymbol{x}}_n - \sigma_{n+1} \boldsymbol{\Psi}_{-h}(\gamma_{n+1}, \boldsymbol{x}_{n+1}),
$$

(79)

and the backward step

$$
\hat{\boldsymbol{x}}_n = \frac{\sigma_n}{\sigma_{n+1}} \hat{\boldsymbol{x}}_{n+1} + \sigma_n \boldsymbol{\Psi}_{-h}(\gamma_{n+1}, \boldsymbol{x}_{n+1}),
$$

$$
\boldsymbol{x}_n = \frac{\sigma_n}{\sigma_{n+1}} \zeta^{-1} \boldsymbol{x}_{n+1} + (1 - \zeta^{-1}) \hat{\boldsymbol{x}}_n - \sigma_n \zeta^{-1} \boldsymbol{\Psi}_h(\gamma_n, \hat{\boldsymbol{x}}_n),
$$

(80)

with the numerical scheme

$$
\hat{\boldsymbol{z}}_i = \frac{1}{\sigma_n} \boldsymbol{x}_n + h \sum_{j=1}^{i-1} a_{ij} \boldsymbol{x}^{\theta}_{0|\gamma_n + c_j h}(\sigma_{\gamma_n + c_j h} \hat{\boldsymbol{z}}_j),
$$

$$
\boldsymbol{\Psi}_h(\gamma_n, \boldsymbol{x}_n) = h \sum_{i=1}^{s} b_i \boldsymbol{x}^{\theta}_{0|\gamma_n + c_i h}(\sigma_{\gamma_n + c_i h} \hat{\boldsymbol{z}}_i).
$$

(81)

$\square$

### C.1.3 NOISE PREDICTION

We present this derivation in the form of Lemma C.2 below.

**Lemma C.2** (Rex (ODE) for noise prediction models). *Let $\boldsymbol{\Phi}$ be an explicit Runge-Kutta solver for the ODE in Equation (69) with Butcher tableau $a_{ij}, b_i, c_i$. The reversible solver for $\boldsymbol{\Phi}$ in terms of the original state $\boldsymbol{x}_t$ is given by the forward step*

$$
\boldsymbol{x}_{n+1} = \frac{\alpha_{n+1}}{\alpha_n} (\zeta \boldsymbol{x}_n + (1 - \zeta) \hat{\boldsymbol{x}}_n) + \alpha_{n+1} \boldsymbol{\Psi}_h(\chi_n, \hat{\boldsymbol{x}}_n),
$$

$$
\hat{\boldsymbol{x}}_{n+1} = \frac{\alpha_{n+1}}{\alpha_n} \hat{\boldsymbol{x}}_n - \alpha_{n+1} \boldsymbol{\Psi}_{-h}(\chi_{n+1}, \boldsymbol{x}_{n+1}),
$$

(82)

*and backward step*

$$
\hat{\boldsymbol{x}}_n = \frac{\alpha_n}{\alpha_{n+1}} \hat{\boldsymbol{x}}_{n+1} + \alpha_n \boldsymbol{\Psi}_{-h}(\chi_{n+1}, \boldsymbol{x}_{n+1}),
$$

$$
\boldsymbol{x}_n = \frac{\alpha_n}{\alpha_{n+1}} \zeta^{-1} \boldsymbol{x}_{n+1} + (1 - \zeta^{-1}) \hat{\boldsymbol{x}}_n - \alpha_n \zeta^{-1} \boldsymbol{\Psi}_h(\chi_n, \hat{\boldsymbol{x}}_n),
$$

(83)

*with step size $h \coloneqq \chi_{n+1} - \chi_n$ and where $\mathbf{\Psi}$ denotes the following scheme*

$$\hat{z}_i = \frac{1}{\alpha_n} x_n + h \sum_{j=1}^{i-1} a_{ij} x^\theta_{T|\chi_n + c_j h}(\alpha_{\chi_n + c_j h} \hat{z}_j),$$

$$\mathbf{\Psi}_h(\chi_n, \boldsymbol{x}_n) = h \sum_{i=1}^{s} b_i x^\theta_{T|\chi_n + c_i h}(\alpha_{\chi_n + c_i h} \hat{z}_i), \tag{84}$$

*Proof.* Recall that the forward step of the McCallum-Foster method for Equation (69) given $\mathbf{\Phi}$ is given as

$$\boldsymbol{z}_{n+1} = \zeta \boldsymbol{z}_n + (1 - \zeta)\hat{z}_n + \mathbf{\Phi}_h(\chi_n, \hat{z}_n),$$
$$\hat{z}_{n+1} = \hat{z}_n - \mathbf{\Phi}_{-h}(\chi_{n+1}, \boldsymbol{z}_{n+1}), \tag{85}$$

with step size $h = \chi_{n+1} - \chi_n$. We use the definition of $\boldsymbol{z}_t = \frac{\alpha_T}{\alpha_t} \boldsymbol{x}_t$ to rewrite the forward pass as

$$\boldsymbol{x}_{n+1} = \frac{\alpha_{n+1}}{\alpha_n}\left(\zeta \boldsymbol{x}_n + (1 - \zeta)\hat{x}_n\right) + \frac{\alpha_{n+1}}{\alpha_T}\mathbf{\Phi}_h\left(\chi_n, \frac{\alpha_T}{\alpha_n}\hat{x}_n\right),$$

$$\hat{x}_{n+1} = \frac{\alpha_{n+1}}{\alpha_n}\hat{x}_n - \frac{\alpha_{n+1}}{\alpha_T}\mathbf{\Phi}_{-h}\left(\chi_{n+1}, \frac{\alpha_T}{\alpha_{n+1}}\boldsymbol{x}_{n+1}\right). \tag{86}$$

*Mutatis mutandis* we find the backward step in $\boldsymbol{x}_t$ to be given as

$$\hat{x}_n = \frac{\alpha_n}{\alpha_{n+1}}\hat{x}_{n+1} + \frac{\alpha_n}{\alpha_T}\mathbf{\Phi}_{-h}\left(\chi_{n+1}, \frac{\alpha_T}{\alpha_{n+1}}\boldsymbol{x}_{n+1}\right),$$

$$\boldsymbol{x}_n = \frac{\alpha_n}{\alpha_{n+1}}\zeta^{-1}\boldsymbol{x}_{n+1} + (1 - \zeta^{-1})\hat{x}_n - \frac{\alpha_n}{\alpha_T}\zeta^{-1}\mathbf{\Phi}_h\left(\chi_n, \frac{\alpha_T}{\alpha_n}\hat{x}_n\right), \tag{87}$$

Next we simplify the explicit RK scheme $\mathbf{\Phi}(\chi_n, \boldsymbol{z}_n)$ for the time-changed probability flow ODE in Equation (70). Recall that the RK scheme can be written as

$$\boldsymbol{z}_i = \boldsymbol{z}_n + h \sum_{j=1}^{i-1} a_{ij} \alpha_T x_{0|\chi_n + c_j h}\left(\frac{\alpha_{\chi_n + c_j h}}{\alpha_T} \boldsymbol{z}_j\right),$$

$$\mathbf{\Phi}_h(\chi_n, \boldsymbol{z}_n) = h \sum_{i=1}^{s} b_i \alpha_T x_{0|\chi_n + c_i h}\left(\frac{\alpha_{\chi_n + c_i h}}{\alpha_T} \boldsymbol{z}_i\right). \tag{88}$$

Next, we replace $\boldsymbol{z}_t$ back with $\boldsymbol{x}_t$ which yields

$$\boldsymbol{z}_i = \alpha_T\left(\frac{1}{\alpha_n} x_n + h \sum_{j=1}^{i-1} a_{ij} x_{0|\chi_n + c_j h}\left(\frac{\alpha_{\chi_n + c_j h}}{\alpha_T} \boldsymbol{z}_j\right)\right),$$

$$\mathbf{\Phi}_h\left(\chi_n, \frac{\alpha_T}{\alpha_n}\boldsymbol{x}_n\right) = \alpha_T h \sum_{i=1}^{s} b_i x_{0|\chi_n + c_i h}\left(\frac{\alpha_{\chi_n + c_i h}}{\alpha_T} \boldsymbol{z}_i\right). \tag{89}$$

To further simplify let $\alpha_T \hat{z}_i = \boldsymbol{z}_i$ and define $\mathbf{\Psi}_h(\chi_n, \boldsymbol{x}_n) \coloneqq \alpha_T \mathbf{\Phi}(\chi_n, \frac{\alpha_T}{\alpha_n}\boldsymbol{x}_n)$.

Thus we can write the following reversible scheme with forward step

$$\boldsymbol{x}_{n+1} = \frac{\alpha_{n+1}}{\alpha_n}\left(\zeta \boldsymbol{x}_n + (1 - \zeta)\hat{x}_n\right) + \alpha_{n+1}\mathbf{\Psi}_h(\chi_n, \hat{x}_n),$$

$$\hat{x}_{n+1} = \frac{\alpha_{n+1}}{\alpha_n}\hat{x}_n - \alpha_{n+1}\mathbf{\Psi}_{-h}(\chi_{n+1}, \boldsymbol{x}_{n+1}), \tag{90}$$

and the backward step

$$\hat{x}_n = \frac{\alpha_n}{\alpha_{n+1}}\hat{x}_{n+1} + \alpha_n\mathbf{\Psi}_{-h}(\chi_{n+1}, \boldsymbol{x}_{n+1}),$$

$$\boldsymbol{x}_n = \frac{\alpha_n}{\alpha_{n+1}}\zeta^{-1}\boldsymbol{x}_{n+1} + (1 - \zeta^{-1})\hat{x}_n - \alpha_n\zeta^{-1}\mathbf{\Psi}_h(\chi_n, \hat{x}_n), \tag{91}$$

with the numerical scheme

$$\hat{\boldsymbol{z}}_i = \frac{1}{\alpha_n}\boldsymbol{x}_n + h\sum_{j=1}^{i-1} a_{ij}\boldsymbol{x}^\theta_{T|\chi_n+c_jh}(\alpha_{\chi_n+c_jh}\hat{\boldsymbol{z}}_j),$$

$$\boldsymbol{\Psi}_h(\chi_n, \boldsymbol{x}_n) = h\sum_{i=1}^{s} b_i \boldsymbol{x}^\theta_{T|\chi_n+c_ih}(\alpha_{\chi_n+c_ih}\hat{\boldsymbol{z}}_i).$$

(92)

$\square$

## C.2 REX (SDE)

In this section we derive the Rex scheme for the reverse-time diffusion SDE along with several helper derivations. We begin by deriving the reparameterization of Equation (9) in Section C.2.2 and then performing an analogous derivation for the noise prediction scenario in Section C.2.3.

### C.2.1 TIME-CHANGED BROWNIAN MOTION

Before detailing this proof we first review some necessary preliminary results about continuous local martingales and Brownian motion. In particular we will show that we can simplify the stochastic integrals in Equation (9) and the corresponding reparameterization with noise prediction models.

**Dambis-Dubins-Schwarz representation theorem.** We restate the Dambis-Dubins-Schwarz representation theorem (Dubins & Schwarz, 1965) which shows that continuous local martingales can be represented as time-changed Brownian motions.

**Theorem C.3** (Dambis-Dubins-Schwarz representation theorem). *Let $M$ be a continuous local martingale adapted to a filtration $\{\mathcal{F}_t\}_{t\geq 0}$ beginning at 0 (i.e., $M_0 = 0$) such that $\langle M\rangle_\infty = \infty$ almost surely. Define the random variables $\{\tau_t\}_{t\geq 0}$ by*

$$\tau_t = \inf\{s \geq 0 : \langle M\rangle_s > t\} = \sup\{s \geq 0 : \langle M\rangle_s = t\}. \quad (93)$$

*Then for any given $t$ the random variable $\tau_t$ is an almost surely finite stopping time, and the process[19] $B_t = M_{\tau_t}$ is a Brownian motion w.r.t. the filtration $\{\mathcal{G}_t\}_{t\geq 0} = \{\mathcal{F}_{\tau_t}\}_{t\geq 0}$. Moreover,*

$$M_t = B_{\langle M\rangle_t}. \quad (94)$$

**A multi-dimensional version of the Dambis-Dubins-Schwarz representation theorem.** In our work we are interested in a $d$-dimensional local martingale $\boldsymbol{M} := (M^1, \dots M^d)$. As such we discuss a multi-dimensional extension of Theorem C.3 which requires that the $d$-dimensional continuous local martingale if the quadratic (covariation) matrix $\langle\boldsymbol{M}\rangle_t^{ij} = \langle M^i, M^j\rangle_t$ is proportional to the identity matrix. We adapt the following theorem from Lowther (2010, Theorem 2) and Bourgade (2010, Theorem 4.13) (*cf.* Revuz & Yor, 2013).

**Theorem C.4** (Multi-dimensional Dambis-Dubins-Schwarz representation theorem). *Let $\boldsymbol{M} = (M^1, \dots, M^d)$ be a collection of continuous local martingales with $\boldsymbol{M}_0 = \boldsymbol{0}$ such that for any $1 \leq 1 \leq d, \langle\boldsymbol{M}\rangle_\infty^{ii} = \infty$ almost surely. Suppose, furthermore, that $\langle M^i, M^j\rangle_t = \delta_{ij}A_t$, where $\delta$ denotes the Kronecker delta, for some process $A$ and all $1 \leq i, j \leq d$ and $t \geq 0$. Then there is a $d$-dimensional Brownian motion $\boldsymbol{B}$ w.r.t. a filtration $\{\mathcal{G}_t\}_{t\geq 0}$ such that for each $t \geq 0$, $\omega \mapsto A_t(\omega)$ is a $\mathcal{G}$-stopping time and*

$$\boldsymbol{M}_t = \boldsymbol{B}_{A_t}. \quad (95)$$

**Enlargement of the probability space.** Recall that in Theorems C.3 and C.4 we stated that quadratic variation of the continuous local martingale needed to tend towards infinity as $t \to \infty$. What when $\langle M\rangle_\infty$ has a nonzero probability of being finite? It can be shown that Theorems C.3 and C.4 holds under an enlargement of the probability space (not the filtration). Consider both our original probability space $(\Omega, \mathcal{F}, P)$ and another probability space $(\Omega', \mathcal{F}', P')$ along with a measurable surjection $f : \Omega' \to \Omega$ preserving probabilities such that $P(A) = P'(f^{-1}(A))$ for all $A \in \mathcal{F}$, *i.e.*, $f_*P'$ is a pushforward measure. Thus any process on the original probability space

---

[19]Defined up to a null set.

can be *lifted* to $(\Omega', \mathcal{F}', P')$ and likewise the filtration is also lifted to $\mathcal{F}'_t = \{f^{-1}(A) : A \in \mathcal{F}_t\}$. Therefore, it is possible to enlarge the probability space so that Brownian motion is defined. *E.g.*, if $(\Omega'', \mathcal{F}'', P'')$ is probability space on which there is a Brownian motion defined, we can take $\Omega' = \Omega \times \Omega''$, $\mathcal{F}' = \mathcal{F} \otimes \mathcal{F}''$, and $P' = P \otimes P''$ for the enlargement, and $f :' \Omega \to \Omega$ is just the projection onto $\Omega$.

We now present a lemma for rewriting the stochastic differential in Equation (9) using the Dambis-Dubins-Schwarz representation theorem. Recall that in Equation (9) we denote the reverse-time $d$-dimensional Brownian motion as $\overline{W}_t$, *i.e.*, by Lévy's characterization we have $\overline{W}_T = \mathbf{0}$ and

$$\overline{W}_t - \overline{W}_s \sim -\mathcal{N}(\mathbf{0}, (t-s)\boldsymbol{I}) = \mathcal{N}(\mathbf{0}, (t-s)\boldsymbol{I}), \tag{96}$$

for $0 \le t < s \le T$. With this in mind we present Lemma C.5 below.

**Lemma C.5.** *The stochastic differential* $\sqrt{-\frac{\mathrm{d}\varrho_t}{\mathrm{d}t}}\,\mathrm{d}\overline{W}_t$ *can be rewritten as a time-changed Brownian motion of the form*

$$\sqrt{-\frac{\mathrm{d}\varrho_t}{\mathrm{d}t}}\,\mathrm{d}\overline{W}_t = \mathrm{d}W_\varrho, \tag{97}$$

*where* $\varrho_t = \gamma_t^2$.

*Proof.* To simplify the stochastic integral term we first define a continuous local martingale $M_t$ via the stochastic integral

$$M_t := \int_T^t \sqrt{-\frac{\mathrm{d}\varrho}{\mathrm{d}t}}\,\mathrm{d}\overline{W}_t. \tag{98}$$

We choose time $T$ as our starting point for the martingale rather than 0 and then integrate in *reverse-time*. However, due to the negative sign within the square root term it is more convenient to work with $W_t$, *i.e.*, the standard $d$-dimensional Brownian motion defined in forward time. Recall that the standard $d$-dimensional Brownian motion in *reverse-time* with starting point $T$ is defined as

$$\overline{W}_t = W_T - W_t \tag{99}$$

which is distributed like $W_t$ in time $T - t$. Define the function $\boldsymbol{f}(t, W_t) = \overline{W}_t$. Then by Itô's lemma we have

$$\mathrm{d}\boldsymbol{f}(t, W_t) = \partial_t \boldsymbol{f}(t, W_t)\,\mathrm{d}t + \sum_{i=1}^d \partial_{\boldsymbol{x}_i}\boldsymbol{f}(t, W_t)\,\mathrm{d}W_t^i + \sum_{i,j=1}^d \partial_{\boldsymbol{x}_i,\boldsymbol{x}_j}\boldsymbol{f}(t, W_t)\,\mathrm{d}\left\langle W^i, W^j \right\rangle_t, \tag{100}$$

which simplifies to

$$\mathrm{d}\boldsymbol{f}(t, W_t) = \mathrm{d}\overline{W}_t = -\mathrm{d}W_t. \tag{101}$$

Thus we can rewrite Equation (98) as

$$M_t = -\int_T^t \sqrt{-\frac{\mathrm{d}\varrho}{\mathrm{d}t}}\,\mathrm{d}W_t. \tag{102}$$

Next we establish a few properties of this martingale. First, $M_T = \mathbf{0}$ by construction. Second, since the integral consists of scalar noise we have that $\langle M^i, M^j \rangle_t = 0$ for all $i \ne j$. Thus, the quadratic variation of $\langle M_t \rangle^{ii}$ for each $i$ is found to be

$$\langle M \rangle_t^{ii} = A_t = -\int_T^t \left(\sqrt{-\frac{\mathrm{d}\varrho_\tau}{\mathrm{d}\tau}}\right)^2 \mathrm{d}\tau, \tag{103}$$

$$= \int_T^t \frac{\mathrm{d}\varrho_\tau}{\mathrm{d}\tau}\,\mathrm{d}\tau, \tag{104}$$

$$= \varrho_t - \varrho_T = \frac{\alpha_t^2}{\sigma_t^2} - \frac{\alpha_T^2}{\sigma_T^2}. \tag{105}$$

Now we have a deterministic mapping from the original time to our new time via $A_t$. Now in general for any valid choice of $(\alpha_t, \sigma_t)$ we don't necessarily have that $\langle M \rangle_\infty^{ii} = \infty$ almost surely and as such we may need to enlarge the underlying probability space. Our constructed martingale can be

expressed as time-changed Brownian motion, see Theorem C.4, such that $M_t = W_{A_t}$ were $W_\varrho$ is the standard $d$-dimensional Brownian motion with time variable $\varrho$.

Now we can rewrite Equation (102) in differential form as

$$\mathrm{d}M_t = \mathrm{d}W_{A_t}. \tag{106}$$

Because Brownian motion is time-shift invariant we can then write

$$\mathrm{d}M_t = \mathrm{d}W_{\varrho_t}. \tag{107}$$

$\square$

**Remark C.3.** Lemma C.5 can similarly be found via Øksendal (2003, Theorem 8.5.7) and Kobayashi (2011, Lemma 2.3); however, do to the oddness of the *reverse-time* integration we found it easier to tackle the problem via the Dambis-Dubins-Schwarz theorem.

**Remark C.4.** Under the common scenario where $\sigma_0 = 0$ then we have that $\langle M \rangle^{ii}_\infty = \infty$ almost surely.

**Lemma C.6.** *Let $\alpha_T > 0$. Then the stochastic differential $\sqrt{\frac{\mathrm{d}}{\mathrm{d}t}(\chi_t^2)}\,\mathrm{d}\overline{W}_t$ can be rewritten as a time-changed Brownian motion of the form*

$$\sqrt{\frac{\mathrm{d}}{\mathrm{d}t}(\chi_t^2)}\,\mathrm{d}\overline{W}_t = \mathrm{d}\overline{W}_{\chi^2}, \tag{108}$$

*where $\chi_t = \frac{\sigma_t}{\alpha_t}$.*

*Proof.* To simplify the stochastic integral term we first define a continuous local martingale $M_t$ via the stochastic integral

$$M_t := \int_T^t \sqrt{\frac{\mathrm{d}}{\mathrm{d}t}(\chi_t^2)}\,\mathrm{d}\overline{W}_t. \tag{109}$$

We choose time $T$ as our starting point for the martingale rather than 0 and then integrate in *reverse-time*, hence the negative sign. Next we establish a few properties of this martingale. First, $M_T = \mathbf{0}$ by construction. Second, since the integral consists of scalar noise we have that $\langle M^i, M^j \rangle_t = 0$ for all $i \neq j$. Thus, the quadratic variation of $\langle M_t \rangle^{ii}$ for each $i$ is found to be

$$\langle M \rangle^{ii}_t = A_t = \int_T^t \left( \sqrt{\frac{\mathrm{d}}{\mathrm{d}\tau}(\chi_\tau^2)} \right)^2 \mathrm{d}\tau, \tag{110}$$

$$= \int_T^t \frac{\mathrm{d}}{\mathrm{d}\tau}(\chi_t^2)\,\mathrm{d}\tau, \tag{111}$$

$$= \chi_t^2 - \chi_T^2 = \frac{\sigma_t^2}{\alpha_t^2} - \frac{\sigma_T^2}{\alpha_T^2}. \tag{112}$$

Now we have a deterministic mapping from the original time to our new time via $A_t$. Now in general for any valid choice of $(\alpha_t, \sigma_t)$ we don't necessarily have that $\langle M \rangle^{ii}_\infty = \infty$ almost surely and as such we may need to enlarge the underlying probability space. Our constructed martingale can be expressed as time-changed Brownian motion, see Theorem C.4, such that $M_t = \overline{W}_{A_t}$ were $\overline{W}_{\chi^2}$ is the standard $d$-dimensional Brownian motion with time variable $\chi^2$ in reverse-time.

Now we can rewrite Equation (98) in differential form as

$$\mathrm{d}M_t = \mathrm{d}\overline{W}_{A_t}. \tag{113}$$

Because Brownian motion is time-shift invariant we can then write

$$\mathrm{d}M_t = \mathrm{d}\overline{W}_{\chi_t^2}. \tag{114}$$

$\square$

**Remark C.5.** The constraint of $\alpha_T > 0$ is important to ensure that $\chi_T$ is finite which is necessary due

$$\overline{W}_{\chi^2} = W_{\chi_T^2} - W_{\chi_t^2}. \tag{115}$$

In practice this is satisfied with a number of noise schedules of diffusion models (*cf*. Appendix H.1).

### C.2.2 PROOF OF PROPOSITION 3.2

In this section we provide the proof for Proposition 3.2 along with associated derivations. We restate Proposition 3.2 below.

**Proposition 3.2** (Time reparameterization of the reverse-time diffusion SDE)**.** *The reverse-time SDE in Equation* (9) *can be rewritten in terms of the data prediction model as*

$$\mathrm{d}\boldsymbol{Y}_\varrho = \frac{\sigma_T}{\gamma_T} \boldsymbol{x}_{0|\varrho}^\theta \left( \frac{\gamma_T \sigma_\varrho}{\sigma_T \gamma_\varrho} \boldsymbol{Y}_\varrho \right) \mathrm{d}\varrho + \frac{\sigma_T}{\gamma_T} \mathrm{d}\boldsymbol{W}_\varrho, \tag{10}$$

*where* $\boldsymbol{Y}_t = \frac{\sigma_T^2 \alpha_t}{\sigma_t^2 \alpha_T} \boldsymbol{X}_t$ *and* $\varrho_t := \frac{\alpha_t^2}{\sigma_t^2}$.

*Proof.* We rewrite Equation (3) in terms of the data prediction model, using the identity

$$\nabla_{\boldsymbol{x}} \log p_t(\boldsymbol{x}) = -\frac{1}{\sigma_t^2} \boldsymbol{x} + \frac{\alpha_t}{\sigma_t^2} \boldsymbol{x}_{0|t}(\boldsymbol{x}), \tag{116}$$

to find

$$\mathrm{d}\boldsymbol{X}_t = \left[ \underbrace{\left( f(t) + \frac{g^2(t)}{\sigma_t^2} \right)}_{=a(t)} \boldsymbol{X}_t + \underbrace{\left( -\frac{\alpha_t g^2(t)}{\sigma_t^2} \right)}_{=b(t)} \boldsymbol{x}_{0|t}(\boldsymbol{X}_t) \right] \mathrm{d}t + g(t) \, \mathrm{d}\overline{\boldsymbol{W}}_t, \tag{117}$$

where

$$f(t) = \frac{\dot{\alpha}_t}{\alpha_t}, \quad g^2(t) = \dot{\sigma}_t^2 - 2\frac{\dot{\alpha}_t}{\alpha_t}\sigma_t^2 = -2\sigma_t^2 \frac{\mathrm{d}\log\gamma_t}{\mathrm{d}t}. \tag{118}$$

Next we find the integrating factor $\Xi_t = \exp - \int_T^t a(u) \, \mathrm{d}u$,

$$\Xi_t = \exp\left( \int_t^T \frac{\mathrm{d}\log\alpha_u}{\mathrm{d}u} + \frac{g^2(u)}{\sigma_u^2} \, \mathrm{d}u \right), \tag{119}$$

$$= \exp\left( \int_t^T \frac{\mathrm{d}\log\alpha_u}{\mathrm{d}u} - 2\frac{\mathrm{d}\log\gamma_u}{\mathrm{d}u} \, \mathrm{d}u \right), \tag{120}$$

$$= \exp\left( \int_t^T \frac{\mathrm{d}\log\alpha_u}{\mathrm{d}u} - 2\left[ \frac{\mathrm{d}\log\alpha_u}{\mathrm{d}u} - \frac{\mathrm{d}\log\sigma_u}{\mathrm{d}u} \right] \mathrm{d}u \right), \tag{121}$$

$$= \exp\left( \int_t^T \frac{\mathrm{d}\log\sigma_u^2}{\mathrm{d}u} - \frac{\mathrm{d}\log\alpha_u}{\mathrm{d}u} \, \mathrm{d}u \right), \tag{122}$$

$$= \exp\left( \log\sigma_T^2 - \log\sigma_t^2 - (\log\alpha_T - \log\alpha_t) \right), \tag{123}$$

$$= \frac{\sigma_T^2 \alpha_t}{\sigma_t^2 \alpha_T}. \tag{124}$$

We can write the integrating factor in terms of $\gamma_t$ as

$$\Xi_t = \frac{\sigma_T \gamma_t}{\sigma_t \gamma_T}. \tag{125}$$

Moreover we can further simplify $b(t)$ as

$$b(t) = \frac{-\alpha_t g^2(t)}{\sigma_t^2}, \tag{126}$$

$$= 2\alpha_t \frac{\mathrm{d}\log\gamma_t}{\mathrm{d}t}. \tag{127}$$

Thus we can rewrite the SDE in Equation (117) as

$$
\mathrm{d}\left[\frac{\sigma_T}{\gamma_T}\frac{\gamma_t}{\sigma_t}\boldsymbol{X}_t\right] = 2\frac{\sigma_T}{\gamma_T}\frac{\alpha_t}{\sigma_t}\gamma_t\frac{\mathrm{d}\log\gamma_t}{\mathrm{d}t}\boldsymbol{x}_{0|t}(\boldsymbol{X}_t)\ \mathrm{d}t + \frac{\sigma_T}{\gamma_T}\frac{\gamma_t}{\sigma_t}\sqrt{-2\sigma_t^2\frac{\mathrm{d}\log\gamma_t}{\mathrm{d}t}}\ \mathrm{d}\overline{\boldsymbol{W}}_t, \tag{128}
$$

$$
\mathrm{d}\boldsymbol{Y}_t \overset{(i)}{=} 2\frac{\sigma_T}{\gamma_T}\frac{\alpha_t}{\sigma_t}\gamma_t\frac{\mathrm{d}\log\gamma_t}{\mathrm{d}t}\boldsymbol{x}_{0|t}\left(\frac{\gamma_T\sigma_t}{\sigma_T\gamma_t}\boldsymbol{Y}_t\right)\ \mathrm{d}t + \frac{\sigma_T}{\gamma_T}\frac{\gamma_t}{\sigma_t}\sqrt{-2\sigma_t^2\frac{\mathrm{d}\log\gamma_t}{\mathrm{d}t}}\ \mathrm{d}\overline{\boldsymbol{W}}_t, \tag{129}
$$

$$
\mathrm{d}\boldsymbol{Y}_t = \frac{\sigma_T}{\gamma_T}\frac{\mathrm{d}\gamma_t^2}{\mathrm{d}t}\boldsymbol{x}_{0|t}\left(\frac{\gamma_T\sigma_t}{\sigma_T\gamma_t}\boldsymbol{Y}_t\right)\ \mathrm{d}t + \frac{\sigma_T}{\gamma_T}\sqrt{-\gamma_t^2\frac{\mathrm{d}\log\gamma_t^2}{\mathrm{d}t}}\ \mathrm{d}\overline{\boldsymbol{W}}_t, \tag{130}
$$

$$
\mathrm{d}\boldsymbol{Y}_t = \frac{\sigma_T}{\gamma_T}\frac{\mathrm{d}\gamma_t^2}{\mathrm{d}t}\boldsymbol{x}_{0|t}\left(\frac{\gamma_T\sigma_t}{\sigma_T\gamma_t}\boldsymbol{Y}_t\right)\ \mathrm{d}t + \frac{\sigma_T}{\gamma_T}\sqrt{-\frac{\mathrm{d}\gamma_t^2}{\mathrm{d}t}}\ \mathrm{d}\overline{\boldsymbol{W}}_t, \tag{131}
$$

$$
\mathrm{d}\boldsymbol{Y}_\varrho \overset{(ii)}{=} \frac{\sigma_T}{\gamma_T}\boldsymbol{x}_{0|\varrho}\left(\frac{\gamma_T\sigma_\varrho}{\sigma_T\gamma_\varrho}\boldsymbol{Y}_\varrho\right)\ \mathrm{d}\varrho + \frac{\sigma_T}{\gamma_T}\ \mathrm{d}\boldsymbol{W}_\varrho, \tag{132}
$$

where (i) holds by the change-of-variables $\boldsymbol{Y}_t = \frac{\sigma_T\gamma_t}{\gamma_T\sigma_t}\boldsymbol{X}_t$ and (ii) holds by Lemma C.5. $\qquad\square$

### C.2.3 PROOF OF REPARAMETERIZED SDE FOR NOISE PREDICTION MODELS

**Proposition C.7** (Time reparameterization of the reverse-time diffusion SDE for noise prediction models). *The reverse-time SDE in Equation* (3) *can be rewritten in terms of the noise prediction model as*

$$
\mathrm{d}\boldsymbol{Y}_\chi = 2\alpha_T\boldsymbol{x}_{T|\chi}^\theta\left(\frac{\alpha_\chi}{\alpha_T}\boldsymbol{Y}_\chi\right)\ \mathrm{d}\chi + \alpha_T\ \mathrm{d}\overline{\boldsymbol{W}}_{\chi^2}, \tag{133}
$$

*where* $\boldsymbol{Y}_t = \frac{\alpha_t}{\alpha_T}\boldsymbol{X}_t$ *and* $\chi_t \coloneqq \frac{\sigma_t}{\alpha_t}$.

*Proof.* We rewrite Equation (3) in terms of the noise prediction model to find

$$
\mathrm{d}\boldsymbol{X}_t = \left[f(t)\boldsymbol{X}_t + \frac{g^2(t)}{\sigma_t}\boldsymbol{x}_{T|t}^\theta(\boldsymbol{X}_t)\right]\ \mathrm{d}t + g(t)\ \mathrm{d}\overline{\boldsymbol{W}}_t, \tag{134}
$$

where

$$
f(t) = \frac{\dot{\alpha}_t}{\alpha_t}, \quad g^2(t) = \dot{\sigma}_t^2 - 2\frac{\dot{\alpha}_t}{\alpha_t}\sigma_t^2 = -2\sigma_t^2\frac{\mathrm{d}\log\gamma_t}{\mathrm{d}t}. \tag{135}
$$

Next we find the integrating factor to be $\exp-\int_T^t f(u)\ \mathrm{d}u = \frac{\alpha_T}{\alpha_t}$. Moreover, we can further simplify $\frac{g^2(t)}{\sigma_t}$ as

$$
\frac{g^2(t)}{\sigma_t} = -2\sigma_t\frac{\mathrm{d}\log\gamma_t}{\mathrm{d}t}, \tag{136}
$$

$$
= -2\sigma_t\frac{\dot{\gamma}_t}{\gamma_t}, \tag{137}
$$

$$
= -2\frac{\sigma_t}{\gamma_t}\frac{\dot{\alpha}_t\sigma_t - \alpha_t\dot{\sigma}_t}{\sigma_t^2}, \tag{138}
$$

$$
= -2\frac{\sigma_t^2}{\alpha_t}\frac{\dot{\alpha}_t\sigma_t - \alpha_t\dot{\sigma}_t}{\sigma_t^2}, \tag{139}
$$

$$
= 2\frac{\sigma_t^2}{\alpha_t}\frac{\alpha_t\dot{\sigma}_t - \dot{\alpha}_t\sigma_t}{\sigma_t^2}, \tag{140}
$$

$$
= 2\frac{\alpha_t\dot{\sigma}_t - \dot{\alpha}_t\sigma_t}{\alpha_t}, \tag{141}
$$

$$
\tag{142}
$$

Let $\chi_t := \frac{\sigma_t}{\alpha_t} = \frac{1}{\gamma_t}$. Thus we can rewrite the SDE in Equation (134) as

$$\mathrm{d}\left[\frac{\alpha_T}{\alpha_t}\boldsymbol{X}_t\right] = \frac{\alpha_T}{\alpha_t}\frac{g^2(t)}{\sigma_t^2}\boldsymbol{x}_{T|t}^\theta(\boldsymbol{X}_t)\,\mathrm{d}t + \frac{\alpha_T}{\alpha_t}\sqrt{-2\sigma_t^2\frac{\mathrm{d}\log\gamma_t}{\mathrm{d}t}}\,\mathrm{d}\overline{\boldsymbol{W}}_t, \tag{143}$$

$$\mathrm{d}\boldsymbol{Y}_t \stackrel{(i)}{=} \frac{\alpha_T}{\alpha_t}\frac{g^2(t)}{\sigma_t^2}\boldsymbol{x}_{T|t}^\theta\left(\frac{\alpha_t}{\alpha_T}\boldsymbol{Y}_t\right)\,\mathrm{d}t + \frac{\alpha_T}{\alpha_t}\sqrt{-2\sigma_t^2\frac{\mathrm{d}\log\gamma_t}{\mathrm{d}t}}\,\mathrm{d}\overline{\boldsymbol{W}}_t, \tag{144}$$

$$\mathrm{d}\boldsymbol{Y}_t = 2\alpha_T\frac{\alpha_t\dot{\sigma}_t - \dot{\alpha}_t\sigma_t}{\alpha_t^2}\boldsymbol{x}_{T|t}^\theta\left(\frac{\alpha_t}{\alpha_T}\boldsymbol{Y}_t\right)\,\mathrm{d}t + \frac{\alpha_T}{\alpha_t}\sqrt{-2\sigma_t^2\frac{\mathrm{d}\log\gamma_t}{\mathrm{d}t}}\,\mathrm{d}\overline{\boldsymbol{W}}_t, \tag{145}$$

$$\mathrm{d}\boldsymbol{Y}_t \stackrel{(ii)}{=} 2\alpha_T\dot{\chi}_t\boldsymbol{x}_{T|t}^\theta\left(\frac{\alpha_t}{\alpha_T}\boldsymbol{Y}_t\right)\,\mathrm{d}t + \alpha_T\sqrt{-2\frac{\sigma_t^2}{\alpha_t^2}\frac{\mathrm{d}\log\gamma_t}{\mathrm{d}t}}\,\mathrm{d}\overline{\boldsymbol{W}}_t, \tag{146}$$

$$\mathrm{d}\boldsymbol{Y}_t = 2\alpha_T\dot{\chi}_t\boldsymbol{x}_{T|t}^\theta\left(\frac{\alpha_t}{\alpha_T}\boldsymbol{Y}_t\right)\,\mathrm{d}t + \alpha_T\sqrt{\dot{\chi}_t^2}\,\mathrm{d}\overline{\boldsymbol{W}}_t, \tag{147}$$

$$\mathrm{d}\boldsymbol{Y}_\chi \stackrel{(iii)}{=} 2\alpha_T\boldsymbol{x}_{T|\chi}^\theta\left(\frac{\alpha_\chi}{\alpha_T}\boldsymbol{Y}_\chi\right)\,\mathrm{d}\chi + \alpha_T\,\mathrm{d}\overline{\boldsymbol{W}}_{\chi^2}, \tag{148}$$

$$\tag{149}$$

where (i) holds by the change-of-variables $\boldsymbol{Y}_t = \frac{\alpha_T}{\alpha_t}\boldsymbol{X}_t$, (ii) holds by

$$-2\frac{\sigma_t^2}{\alpha_t^2}\frac{\mathrm{d}\log\gamma_t}{\mathrm{d}t} = \frac{\sigma_t^2}{\alpha_t^2}\frac{\mathrm{d}(-2\log\gamma_t)}{\mathrm{d}t}, \tag{150}$$

$$= \frac{\sigma_t^2}{\alpha_t^2}\frac{\mathrm{d}\log\chi_t^2}{\mathrm{d}t}, \tag{151}$$

$$= \frac{\sigma_t^2}{\alpha_t^2}\frac{\dot{\chi}_t^2}{\chi_t^2}, \tag{152}$$

$$= \dot{\chi}_t^2, \tag{153}$$

and (iii) holds by Lemma C.5 *mutatis mutandis* for $\chi_t$. $\qquad\square$

### C.2.4 DERIVATION OF REX (SDE)

We present derivations for both the data prediction and noise prediction formulations.

**Data prediction.** We present this derivation in the form of Lemma C.8 below.

**Lemma C.8** (Rex (SDE) for data prediction models)**.** *Let $\boldsymbol{\Phi}$ be an explicit stochastic Runge-Kutta solver for the additive noise SDE in Equation (10), we construct the following reversible scheme for diffusion models*

$$\begin{aligned}
\boldsymbol{X}_{n+1} &= \frac{\sigma_{n+1}\gamma_n}{\gamma_{n+1}\sigma_n}(\zeta\boldsymbol{X}_n + (1-\zeta)\hat{\boldsymbol{X}}_n) + \frac{\sigma_{n+1}}{\gamma_{n+1}}\boldsymbol{\Psi}_h(\varrho_n, \hat{\boldsymbol{X}}_n, \boldsymbol{W}_\varrho(\omega)), \\
\hat{\boldsymbol{X}}_{n+1} &= \frac{\sigma_{n+1}\gamma_n}{\gamma_{n+1}\sigma_n}\hat{\boldsymbol{X}}_n - \frac{\sigma_{n+1}}{\gamma_{n+1}}\boldsymbol{\Psi}_{-h}(\varrho_{n+1}, \boldsymbol{X}_{n+1}, \boldsymbol{W}_\varrho(\omega)),
\end{aligned} \tag{154}$$

*and the backward step is given as*

$$\begin{aligned}
\hat{\boldsymbol{X}}_n &= \frac{\sigma_n\gamma_{n+1}}{\gamma_n\sigma_{n+1}}\hat{\boldsymbol{X}}_n + \frac{\sigma_n}{\gamma_n}\boldsymbol{\Psi}_{-h}(\varrho_{n+1}, \boldsymbol{X}_{n+1}, \boldsymbol{W}_\varrho(\omega)), \\
\boldsymbol{X}_n &= \frac{\sigma_n\gamma_{n+1}}{\gamma_n\sigma_{n+1}}\zeta^{-1}\boldsymbol{X}_{n+1} + (1-\zeta^{-1})\hat{\boldsymbol{X}}_n - \frac{\sigma_n}{\gamma_n}\zeta^{-1}\boldsymbol{\Psi}_h(\varrho_n, \hat{\boldsymbol{X}}_n, \boldsymbol{W}_\varrho(\omega)),
\end{aligned} \tag{155}$$

*with step size $h := \varrho_{n+1} - \varrho_n$ and where $\boldsymbol{\Psi}$ denotes the following scheme*

$$\hat{\boldsymbol{Z}}_i = \frac{\gamma_n}{\sigma_n}\boldsymbol{X}_n + h\sum_{j=1}^{i-1}\left[a_{ij}\boldsymbol{x}_{0|\varrho_n+c_jh}\left(\frac{\sigma_{\varrho_n+c_jh}}{\gamma_{\varrho_n+c_jh}}\hat{\boldsymbol{Z}}_j\right)\right] + a_i^W\boldsymbol{W}_n + a_i^H\boldsymbol{H}_n,$$

$$\boldsymbol{\Psi}_h(\varrho_n, \boldsymbol{X}_n, \boldsymbol{W}_\varrho(\omega)) = h\sum_{j=1}^{s}\left[b_i\boldsymbol{x}_{0|\varrho_n+c_ih}\left(\frac{\sigma_{\varrho_n+c_ih}}{\gamma_{\varrho_n+c_jh}}\hat{\boldsymbol{Z}}_j\right)\right] + b^W\boldsymbol{W}_n + b^H\boldsymbol{H}_n. \tag{156}$$

*Proof.* We write the SRK scheme for the time-changed reverse-time SDE in Equation (10) to construct the following SRK scheme

$$\boldsymbol{Z}_i = \boldsymbol{Y}_n + h\sum_{j=1}^{i-1}\left[a_{ij}\frac{\sigma_T}{\gamma_T}\boldsymbol{x}_{0|\varrho_n+c_jh}\left(\frac{\gamma_T\sigma_{\varrho_n+c_jh}}{\sigma_T\gamma_{\varrho_n+c_jh}}\boldsymbol{Z}_j\right)\right] + \frac{\sigma_T}{\gamma_T}(a_i^W\boldsymbol{W}_n + a_i^H\boldsymbol{H}_n),$$

$$\boldsymbol{Y}_{n+1} = \boldsymbol{Y}_n + h\sum_{i=1}^{s}\left[b_i\frac{\sigma_T}{\gamma_T}\boldsymbol{x}_{0|\varrho_n+c_ih}\left(\frac{\gamma_T\sigma_{\varrho_n+c_ih}}{\sigma_T\gamma_{\varrho_n+c_ih}}\boldsymbol{Z}_i\right)\right] + \frac{\sigma_T}{\gamma_T}(b^W\boldsymbol{W}_n + b^H\boldsymbol{H}_n),$$

(157)

with step size $h = \varrho_{n+1} - \varrho_n$. Next, we replace $\boldsymbol{Y}_t$ back with $\boldsymbol{X}_t$ which yields

$$\boldsymbol{Z}_i = \frac{\sigma_T}{\gamma_T}\left(\frac{\gamma_n}{\sigma_n}\boldsymbol{X}_n + h\sum_{j=1}^{i-1}\left[a_{ij}\boldsymbol{x}_{0|\varrho_n+c_jh}\left(\frac{\gamma_T\sigma_{\varrho_n+c_jh}}{\sigma_T\gamma_{\varrho_n+c_jh}}\boldsymbol{Z}_j\right)\right]\right)$$

$$+ \frac{\sigma_T}{\gamma_T}(a_i^W\boldsymbol{W}_n + a_i^H\boldsymbol{H}_n),$$

$$\frac{\sigma_T\gamma_{n+1}}{\gamma_T\sigma_{n+1}}\boldsymbol{X}_{n+1} = \frac{\sigma_T\gamma_n}{\gamma_T\sigma_n}\boldsymbol{X}_n$$

$$+ \underbrace{\frac{\sigma_T}{\gamma_T}h\sum_{i=1}^{s}\left[b_i\frac{\sigma_T}{\gamma_T}\boldsymbol{x}_{0|\varrho_n+c_ih}\left(\frac{\gamma_T\sigma_{\varrho_n+c_ih}}{\sigma_T\gamma_{\varrho_n+c_ih}}\boldsymbol{Z}_i\right)\right] + \frac{\sigma_T}{\gamma_T}(b^W\boldsymbol{W}_n + b^H\boldsymbol{H}_n)}_{=\boldsymbol{\Psi}_h(\varrho_n,\boldsymbol{X}_n,\boldsymbol{W}_\varrho)}.$$

(158)

To further simplify let $\frac{\sigma_T}{\gamma_T}\hat{\boldsymbol{Z}}_i = \boldsymbol{Z}_i$, then we construct the reversible scheme with forward pass:

$$\boldsymbol{X}_{n+1} = \frac{\sigma_{n+1}\gamma_n}{\gamma_{n+1}\sigma_n}(\zeta\boldsymbol{X}_n + (1-\zeta)\hat{\boldsymbol{X}}_n) + \frac{\sigma_{n+1}}{\gamma_{n+1}}\boldsymbol{\Psi}_h(\varrho_n,\hat{\boldsymbol{X}}_n,\boldsymbol{W}_\varrho(\omega)),$$

$$\hat{\boldsymbol{X}}_{n+1} = \frac{\sigma_{n+1}\gamma_n}{\gamma_{n+1}\sigma_n}\hat{\boldsymbol{X}}_n - \frac{\sigma_{n+1}}{\gamma_{n+1}}\boldsymbol{\Psi}_{-h}(\varrho_{n+1},\boldsymbol{X}_{n+1},\boldsymbol{W}_\varrho(\omega)),$$

(159)

and backward pass

$$\hat{\boldsymbol{X}}_n = \frac{\sigma_n\gamma_{n+1}}{\gamma_n\sigma_{n+1}}\hat{\boldsymbol{X}}_n + \frac{\sigma_n}{\gamma_n}\boldsymbol{\Psi}_{-h}(\varrho_{n+1},\boldsymbol{X}_{n+1},\boldsymbol{W}_\varrho(\omega)),$$

$$\hat{\boldsymbol{X}}_{n+1} = \frac{\sigma_n\gamma_{n+1}}{\gamma_n\sigma_{n+1}}\zeta^{-1}\boldsymbol{X}_{n+1} + (1-\zeta^{-1})\hat{\boldsymbol{X}}_n - \frac{\sigma_n}{\gamma_n}\zeta^{-1}\boldsymbol{\Psi}_h(\varrho_n,\hat{\boldsymbol{X}}_n,\boldsymbol{W}_\varrho(\omega)),$$

(160)

with step size $h := \varrho_{n+1} - \varrho_n$

$$\hat{\boldsymbol{Z}}_i = \frac{\gamma_n}{\sigma_n}\boldsymbol{X}_n + h\sum_{j=1}^{i-1}\left[a_{ij}\boldsymbol{x}_{0|\varrho_n+c_jh}\left(\frac{\sigma_{\varrho_n+c_jh}}{\gamma_{\varrho_n+c_jh}}\hat{\boldsymbol{Z}}_j\right)\right] + a_i^W\boldsymbol{W}_n + a_i^H\boldsymbol{H}_n,$$

$$\boldsymbol{\Psi}_h(\varrho_n,\boldsymbol{X}_n,\boldsymbol{W}_\varrho(\omega)) = h\sum_{j=1}^{s}\left[b_i\boldsymbol{x}_{0|\varrho_n+c_ih}\left(\frac{\sigma_{\varrho_n+c_ih}}{\gamma_{\varrho_n+c_jh}}\hat{\boldsymbol{Z}}_j\right)\right] + b^W\boldsymbol{W}_n + b^H\boldsymbol{H}_n.$$

(161)

$\square$

**Noise prediction.** We present this derivation in the form of Lemma C.9 below.

**Lemma C.9** (Rex (SDE) for noise prediction models)**.** *Let $\boldsymbol{\Phi}$ be an explicit stochastic Runge-Kutta solver for the additive noise SDE in Equation (133), we construct the following reversible scheme for diffusion models*

$$\boldsymbol{X}_{n+1} = \frac{\alpha_{n+1}}{\alpha_n}(\zeta\boldsymbol{X}_n + (1-\zeta)\hat{\boldsymbol{X}}_n) + \alpha_{n+1}\boldsymbol{\Psi}_h(\chi_n,\hat{\boldsymbol{X}}_n,\boldsymbol{W}_{\chi^2}(\omega)),$$

$$\hat{\boldsymbol{X}}_{n+1} = \frac{\alpha_{n+1}}{\alpha_n}\hat{\boldsymbol{X}}_n - \alpha_{n+1}\boldsymbol{\Psi}_{-h}(\chi_{n+1},\boldsymbol{X}_{n+1},\boldsymbol{W}_{\chi^2}(\omega)),$$

(162)

*and the backward step is given as*

$$\hat{\boldsymbol{X}}_n = \frac{\alpha_n}{\alpha_{n+1}}\hat{\boldsymbol{X}}_n + \alpha_n\boldsymbol{\Psi}_{-h}(\chi_{n+1},\boldsymbol{X}_{n+1},\boldsymbol{W}_{\chi^2}(\omega)),$$

$$\boldsymbol{X}_n = \frac{\alpha_n}{\alpha_n+1}\zeta^{-1}\boldsymbol{X}_{n+1} + (1-\zeta^{-1})\hat{\boldsymbol{X}}_n - \alpha_n\zeta^{-1}\boldsymbol{\Psi}_h(\chi_n,\hat{\boldsymbol{X}}_n,\boldsymbol{W}_{\chi^2}(\omega)),$$

(163)

*with step size $h \coloneqq \chi_{n+1} - \chi_n$ and where $\boldsymbol{\Psi}$ denotes the following scheme*

$$\hat{\boldsymbol{Z}}_i = \frac{1}{\alpha_n}\boldsymbol{X}_n + h\sum_{j=1}^{i-1}\left[2a_{ij}\boldsymbol{x}_{T|\chi_n+c_jh}^{\theta}\left(\alpha_{\chi_n+c_jh}\hat{\boldsymbol{Z}}_j\right)\right] + a_i^W\boldsymbol{W}_n + a_i^H\boldsymbol{H}_n,$$

$$\boldsymbol{\Psi}_h(\chi_n, \boldsymbol{X}_n, \boldsymbol{W}_\chi(\omega)) = h\sum_{j=1}^{s}\left[2b_i\boldsymbol{x}_{T|\chi_n+c_ih}^{\theta}\left(\alpha_{\chi_n+c_ih}\hat{\boldsymbol{Z}}_j\right)\right] + b^W\boldsymbol{W}_n + b^H\boldsymbol{H}_n. \tag{164}$$

*Proof.* We write the SRK scheme for the time-changed reverse-time SDE in Equation (133) to construct the following SRK scheme

$$\boldsymbol{Z}_i = \boldsymbol{Y}_n + h\sum_{j=1}^{i-1}\left[2a_{ij}\alpha_T\boldsymbol{x}_{T|\chi_n+c_jh}\left(\frac{\alpha_{\chi_n+c_jh}}{\alpha_T}\boldsymbol{Z}_j\right)\right] + \alpha_T(a_i^W\boldsymbol{W}_n + a_i^H\boldsymbol{H}_n),$$

$$\boldsymbol{Y}_{n+1} = \boldsymbol{Y}_n + h\sum_{i=1}^{s}\left[2b_i\alpha_T\boldsymbol{x}_{T|\chi_n+c_ih}\left(\frac{\alpha_{\chi_n+c_ih}}{\alpha_T}\boldsymbol{Z}_i\right)\right] + \alpha_T(b^W\boldsymbol{W}_n + b^H\boldsymbol{H}_n), \tag{165}$$

with step size $h = \chi_{n+1} - \chi_n$. Next, we replace $\boldsymbol{Y}_t$ back with $\boldsymbol{X}_t$ which yields

$$\boldsymbol{Z}_i = \alpha_T\left(\frac{1}{\alpha_n}\boldsymbol{X}_n + h\sum_{j=1}^{i-1}\left[2a_{ij}\boldsymbol{x}_{T|\chi_n+c_jh}\left(\frac{\alpha_{\chi_n+c_jh}}{\alpha_T}\boldsymbol{Z}_j\right)\right]\right)$$

$$+ \alpha_T(a_i^W\boldsymbol{W}_n + a_i^H\boldsymbol{H}_n),$$

$$\frac{\alpha_{n+1}}{\alpha_T}\boldsymbol{X}_{n+1} = \frac{\alpha_T}{\alpha_n}\boldsymbol{X}_n \tag{166}$$

$$+ \underbrace{\alpha_T h\sum_{i=1}^{s}\left[2b_i\alpha_T\boldsymbol{x}_{T|\chi_n+c_ih}\left(\frac{\alpha_{\chi_n+c_ih}}{\alpha_T}\boldsymbol{Z}_i\right)\right] + \alpha_T(b^W\boldsymbol{W}_n + b^H\boldsymbol{H}_n)}_{=\boldsymbol{\Psi}_h(\chi_n,\boldsymbol{X}_n,\boldsymbol{W}_\chi)}.$$

To further simplify let $\alpha_T\hat{\boldsymbol{Z}}_i = \boldsymbol{Z}_i$, then we construct the reversible scheme with forward pass:

$$\boldsymbol{X}_{n+1} = \frac{\alpha_{n+1}}{\alpha_n}(\zeta\boldsymbol{X}_n + (1-\zeta)\hat{\boldsymbol{X}}_n) + \alpha_{n+1}\boldsymbol{\Psi}_h(\chi_n, \hat{\boldsymbol{X}}_n, \boldsymbol{W}_\chi(\omega)),$$

$$\hat{\boldsymbol{X}}_{n+1} = \frac{\alpha_{n+1}}{\alpha_n}\hat{\boldsymbol{X}}_n - \alpha_{n+1}\boldsymbol{\Psi}_{-h}(\chi_{n+1}, \boldsymbol{X}_{n+1}, \boldsymbol{W}_\chi(\omega)), \tag{167}$$

and backward pass

$$\hat{\boldsymbol{X}}_n = \frac{\alpha_n}{\alpha_{n+1}}\hat{\boldsymbol{X}}_n + \alpha_n\boldsymbol{\Psi}_{-h}(\chi_{n+1}, \boldsymbol{X}_{n+1}, \boldsymbol{W}_\chi(\omega)),$$

$$\hat{\boldsymbol{X}}_{n+1} = \frac{\alpha_n}{\alpha_{n+1}}\zeta^{-1}\boldsymbol{X}_{n+1} + (1-\zeta^{-1})\hat{\boldsymbol{X}}_n - \alpha_n\zeta^{-1}\boldsymbol{\Psi}_h(\chi_n, \hat{\boldsymbol{X}}_n, \boldsymbol{W}_\chi(\omega)), \tag{168}$$

with step size $h \coloneqq \chi_{n+1} - \chi_n$

$$\hat{\boldsymbol{Z}}_i = \frac{\gamma_n}{\sigma_n}\boldsymbol{X}_n + h\sum_{j=1}^{i-1}\left[2a_{ij}\boldsymbol{x}_{T|\chi_n+c_jh}\left(\alpha_{\chi_n+c_jh}\hat{\boldsymbol{Z}}_j\right)\right] + a_i^W\boldsymbol{W}_n + a_i^H\boldsymbol{H}_n,$$

$$\boldsymbol{\Psi}_h(\chi_n, \boldsymbol{X}_n, \boldsymbol{W}_\chi(\omega)) = h\sum_{j=1}^{s}\left[2b_i\boldsymbol{x}_{T|\chi_n+c_ih}\left(\alpha_{\chi_n+c_ih}\hat{\boldsymbol{Z}}_j\right)\right] + b^W\boldsymbol{W}_n + b^H\boldsymbol{H}_n. \tag{169}$$

*N.B.,* $\boldsymbol{W}_n = \overline{\boldsymbol{W}}_{\chi_{n+1}^2} - \overline{\boldsymbol{W}}_{\chi_n^2}$. □

### C.3   Proof of Proposition 3.3

We now can construct Rex.

**Proposition 3.3** (Rex). *Without loss of generality let $\boldsymbol{\Phi}$ denote an explicit SRK scheme for the SDE in Equation* (10) *with extended Butcher tableau $a_{ij}, b_i, c_i, a_i^W, a_i^H, b^W, b^H$. Fix an $\omega \in \Omega$ and let $\boldsymbol{W}$ be the Brownian motion over time variable $\varsigma$. Then the reversible solver constructed from $\boldsymbol{\Phi}$ in terms of the underlying state variable $\boldsymbol{X}_t$ is given by the forward step*

$$
\begin{aligned}
\boldsymbol{X}_{n+1} &= \frac{w_{n+1}}{w_n}\left(\zeta \boldsymbol{X}_n + (1-\zeta)\hat{\boldsymbol{X}}_n\right) + w_{n+1}\boldsymbol{\Psi}_h(\varsigma_n, \hat{\boldsymbol{X}}_n, \boldsymbol{W}_n(\omega)), \\
\hat{\boldsymbol{X}}_{n+1} &= \frac{w_{n+1}}{w_n}\hat{\boldsymbol{X}}_n - w_{n+1}\boldsymbol{\Psi}_{-h}(\varsigma_{n+1}, \boldsymbol{X}_{n+1}, \boldsymbol{W}_n(\omega)),
\end{aligned}
\tag{12}
$$

*and backward step*

$$
\begin{aligned}
\hat{\boldsymbol{X}}_n &= \frac{w_n}{w_{n+1}}\hat{\boldsymbol{X}}_{n+1} + w_n\boldsymbol{\Psi}_{-h}(\varsigma_{n+1}, \boldsymbol{X}_{n+1}, \boldsymbol{W}_n(\omega)), \\
\boldsymbol{X}_n &= \frac{w_n}{w_{n+1}}\zeta^{-1}\boldsymbol{X}_{n+1} + (1-\zeta^{-1})\hat{\boldsymbol{X}}_n - w_n\zeta^{-1}\boldsymbol{\Psi}_h(\varsigma_n, \hat{\boldsymbol{X}}_n, \boldsymbol{W}_n(\omega)),
\end{aligned}
\tag{13}
$$

*with step size $h := \varsigma_{n+1} - \varsigma_n$ and where $\boldsymbol{\Psi}$ denotes the following scheme*

$$
\hat{\boldsymbol{Z}}_i = \frac{1}{w_n}\boldsymbol{X}_n + h\sum_{j=1}^{i-1}\left[a_{ij}\boldsymbol{f}^\theta\left(\varsigma_n + c_j h, w_{\varsigma_n + c_j h}\hat{\boldsymbol{Z}}_j\right)\right] + a_i^W\boldsymbol{W}_n(\omega) + a_i^H\boldsymbol{H}_n(\omega),
$$

$$
\boldsymbol{\Psi}_h(\varsigma_n, \boldsymbol{X}_n, \boldsymbol{W}_\varrho(\omega)) = h\sum_{j=1}^{s}\left[b_i\boldsymbol{f}^\theta\left(\varsigma_n + c_i h, w_{\varsigma_n + c_i h}\hat{\boldsymbol{Z}}_j\right)\right] + b^W\boldsymbol{W}_n(\omega) + b^H\boldsymbol{H}_n(\omega),
\tag{14}
$$

*where $\boldsymbol{f}^\theta$ denotes the data prediction model, $w_n = \frac{\sigma_n}{\gamma_n}$ and $\varsigma_t = \varrho_t$. The ODE case is recovered for an explicit RK scheme $\boldsymbol{\Phi}$ for the ODE in Equation* (70) *with $w_n = \sigma_n$ and $\varsigma_t = \gamma_t$ For noise prediction models we have $\boldsymbol{f}^\theta$ denoting the noise prediction model with $w_n = \alpha_n$ and $\varsigma_t = \frac{\sigma_n}{\alpha_n}$.*

*Proof.* This follows by Lemmas C.1, C.2, C.8 and C.9 *mutatis mutandis.* □

# D CONVERGENCE ORDER PROOFS

## D.1 ASSUMPTIONS

Beyond the general regularity conditions imposed on the learned diffusion model itself (see Lu et al., 2022b; Blasingame & Liu, 2024a; 2025) we also assert that in the noise prediction setting that $\alpha_T > 0$. In practice most commonly used diffusion noise schedules like the linear or scaled linear schedule satisfy this, (see Appendix H.1; *cf*. Lin et al., 2024).

## D.2 PROOF OF THEOREM 4.1

**Theorem 4.1** (Rex is a $k$-th order solver). *Let $\boldsymbol{\Phi}$ be a $k$-th order explicit Runge-Kutta scheme for the reparameterized probability flow ODE in Equation* (70) *with variance preserving noise schedule $(\alpha_t, \sigma_t)$. Then Rex constructed from $\boldsymbol{\Phi}$ is a $k$-th order solver, i.e., given the reversible solution $\{\boldsymbol{x}_n, \hat{\boldsymbol{x}}_n\}_{n=1}^N$ and true solution $\boldsymbol{x}_{t_n}$ we have*

$$
\|\boldsymbol{x}_n - \boldsymbol{x}_{t_n}\| \leq Ch^k,
\tag{15}
$$

*for constants $C, h_{max} > 0$ and for step sizes $h \in [0, h_{max}]$.*

*Proof.* ' We will prove this for both the data prediction and noise prediction formulations.

**Data prediction.** By Theorem A.1 we have that reversible $\boldsymbol{\Phi}$ is a $k$-th order solver, and thus

$$
\|\boldsymbol{y}_n - \boldsymbol{y}_{t_n}\| \leq Ch^k.
\tag{170}
$$

We use the change of variables from Equation (70) to find

$$
\left\|\frac{\sigma_T}{\sigma_n}\boldsymbol{x}_n - \frac{\sigma_T}{\sigma_n}\boldsymbol{x}_{t_n}\right\| \leq Ch^k,
\tag{171}
$$

which simplifies to

$$\|\boldsymbol{x}_n - \boldsymbol{x}_{t_n}\| \leq \frac{\sigma_n}{\sigma_T} C h^k. \tag{172}$$

Now by definition for variance preserving type diffusion SDEs we have that $\sigma_t \leq 1$ for all $t$. Thus we can write

$$\|\boldsymbol{x}_n - \boldsymbol{x}_{t_n}\| \leq C_1 h^k, \tag{173}$$

where $C_1 = \frac{C}{\sigma_T}$.

**Noise prediction.** By Theorem A.1 we have that reversible $\boldsymbol{\Phi}$ is a $k$-th order solver, and thus

$$\|\boldsymbol{y}_n - \boldsymbol{y}_{t_n}\| \leq C h^k. \tag{174}$$

We use the change of variables from Equation (69) to find

$$\left\| \frac{\alpha_T}{\alpha_n} \boldsymbol{x}_n - \frac{\alpha_T}{\alpha_n} \boldsymbol{x}_{t_n} \right\| \leq C h^k, \tag{175}$$

which simplifies to

$$\|\boldsymbol{x}_n - \boldsymbol{x}_{t_n}\| \leq \frac{\alpha_n}{\alpha_T} C h^k. \tag{176}$$

Now by definition we have $\alpha_t \leq 1$ for all $t$ and we assume that $\alpha_T > 0$. Thus we can write

$$\|\boldsymbol{x}_n - \boldsymbol{x}_{t_n}\| \leq C_1 h^k, \tag{177}$$

where $C_1 = \frac{C}{\sigma_T}$. $\qquad\square$

### D.3 PROOF OF THEOREM 4.2

**Definition D.1** (Strong order of convergence). Suppose an SDE solver admits a numerical solution $\boldsymbol{X}_n$ and we have a true solution $\boldsymbol{X}_{t_n}$. If

$$\sup_{0 \leq n \leq N} \mathbb{E}\|\boldsymbol{X}_n - \boldsymbol{X}_{t_n}\|^2 \leq C h^{2\alpha}, \tag{178}$$

where $C > 0$ is a constant and $h$ is the step size, then the SDE solver strongly converges with order $\alpha$.

**Theorem 4.2** (Convergence order for stochastic $\boldsymbol{\Psi}$). *Let $\boldsymbol{\Phi}$ be a SRK scheme with strong order of convergence $\xi > 0$ for the reparameterized reverse-time diffusion SDE in Equation (10) with variance preserving noise schedule $(\alpha_t, \sigma_t)$ and $\alpha_T > 0$. Then $\boldsymbol{\Psi}$ constructed from $\boldsymbol{\Phi}$ has strong order of convergence $\xi$.*

*Proof.* We will prove this for both the data prediction and noise prediction formulations.

**Data prediction.** By definition we have $\boldsymbol{\Phi}$ has strong order of convergence $\xi$ and thus,

$$\sup_{0 \leq n \leq N} \mathbb{E}\|\boldsymbol{Y}_n - \boldsymbol{Y}_{t_n}\|^2 \leq C h^{2\xi}, \tag{179}$$

where $h = \frac{\sigma_{n+1}^2}{\alpha_{n+1}} - \frac{\sigma_n^2}{\alpha_n}$. We use the change of variables from Equation (10) to find

$$\sup_{0 \leq n \leq N} \mathbb{E} \left\| \frac{\sigma_T^2 \alpha_n}{\sigma_n^2 \alpha_T} \boldsymbol{X}_n - \frac{\sigma_T^2 \alpha_n}{\sigma_n^2 \alpha_T} \boldsymbol{X}_{t_n} \right\|^2 \leq C h^{2\xi}, \tag{180}$$

which simplifies to

$$\sup_{0 \leq n \leq N} \mathbb{E}\|\boldsymbol{X}_n - \boldsymbol{X}_{t_n}\|^2 \leq \frac{\sigma_n \sqrt{\alpha_T}}{\sigma_T \sqrt{\alpha_n}} C h^{2\xi}. \tag{181}$$

Since by definition of $\alpha_n$ is a monotonically decreasing function, $\sigma_n$ is a monotonically increasing function, $\alpha_T > 0$, and $\sigma_T \leq 1$ we can write

$$\sup_{0 \leq n \leq N} \mathbb{E}\|\boldsymbol{X}_n - \boldsymbol{X}_{t_n}\|^2 \leq C h^{2\xi}, \tag{182}$$

as

$$\frac{\sigma_n \sqrt{\alpha_T}}{\sigma_T \sqrt{\alpha_n}} \leq 1. \tag{183}$$

**Noise prediction.** By definition we have $\mathbf{\Phi}$ has strong order of convergence $\xi$ and thus,

$$\sup_{0 \leq n \leq N} \mathbb{E}\|\boldsymbol{Y}_n - \boldsymbol{Y}_{t_n}\|^2 \leq Ch^{2\xi}, \tag{184}$$

where $h = \frac{\sigma_{n+1}}{\alpha_{n+1}} - \frac{\sigma_n}{\alpha_n}$. We use the change of variables from Equation (133) to find

$$\sup_{0 \leq n \leq N} \mathbb{E} \left\| \frac{\alpha_n}{\alpha_T} \boldsymbol{X}_n - \frac{\alpha_n}{\alpha_T} \boldsymbol{X}_{t_n} \right\|^2 \leq Ch^{2\xi}, \tag{185}$$

which simplifies to

$$\sup_{0 \leq n \leq N} \mathbb{E}\|\boldsymbol{X}_n - \boldsymbol{X}_{t_n}\|^2 \leq \frac{\sqrt{\alpha_T}}{\sqrt{\alpha_n}} Ch^{2\xi}. \tag{186}$$

Since by definition of $\alpha_n$ is a monotonically decreasing function strictly less than 1 and $\alpha_T > 0$ we can write

$$\sup_{0 \leq n \leq N} \mathbb{E}\|\boldsymbol{X}_n - \boldsymbol{X}_{t_n}\|^2 \leq Ch^{2\xi}. \tag{187}$$

$\square$

# E   RELATION TO OTHER SOLVERS FOR DIFFUSION MODELS

While this paper primarily focused on Rex and the family of reversible solvers created by it, we wish to discuss the relation between the underlying scheme $\mathbf{\Psi}$ constructed from our method and other existing solvers for diffusion models.

Figure 6: Overview of the construction of $\mathbf{\Psi}$ for the probability flow ODE from an underlying RK scheme $\mathbf{\Phi}$ for the reparameterized ODE. This graph holds for the SDE case *mutatis mutandis*.

Surprisingly, we discover that using Lawson methods outlined in Figure 6 (*cf*. Figure 2 from the main paper) is a surprisingly generalized methodology for construing numerical schemes for diffusion modes, and that it subsumes previous works. This means that several of the reversible schemes we presented here are reversible variants of well known schemes in the literature in diffusion models.

**Theorem 4.3** (Rex subsumes previous solvers). *The underlying scheme used* $\mathbf{\Psi}$ *in Rex given by*

$$\hat{\boldsymbol{Z}}_i = \frac{1}{w_n} \boldsymbol{X}_n + h \sum_{j=1}^{i-1} \left[ a_{ij} \boldsymbol{f}^\theta \left( \varsigma_n + c_j h, w_{\varsigma_n + c_j h} \hat{\boldsymbol{Z}}_j \right) \right] + a_i^W \boldsymbol{W}_n(\omega) + a_i^H \boldsymbol{H}_n(\omega),$$

$$\boldsymbol{X}_{n+1} = \frac{w_{n+1}}{w_n} \boldsymbol{X}_n + w_{n+1} \left( h \sum_{j=1}^{s} \left[ b_i \boldsymbol{f}^\theta \left( \varsigma_n + c_i h, w_{\varsigma_n + c_i h} \hat{\boldsymbol{Z}}_j \right) \right] + b^W \boldsymbol{W}_n(\omega) + b^H \boldsymbol{H}_n(\omega) \right),$$

$$\tag{16}$$

*subsumes the following solvers for diffusion models*

  1. *DDIM (Song et al., 2021a),*

  2. *DPM-Solver-1, DPM-Solver-2, DPM-Solver-12 (Lu et al., 2022b),*

3. *DPM-Solver++1, DPM-Solver++(2S), SDE-DPM-Solver-1, SDE-DPM-Solver++1 (Lu et al., 2022a),*

4. *SEEDS-1 (Gonzalez et al., 2024), and*

5. *gDDIM (Zhang et al., 2023).*

*Proof.* We prove the connection to each solver in the list within a set of separate propositions for easier readability. The statement holds true via Propositions E.1 to E.8 and Corollaries E.1.1 to E.6.1. □

**Corollary 4.3.1** (Rex is reversible version of previous solvers). *Rex is the reversible revision of the well-known solvers for diffusion models in Theorem 4.3.*

**Remark E.1.** The SDE solvers constructed from Foster-Reis-Strange SRK schemes are wholly unique (with the exception of the trivial Euler-Maruyama scheme) and have no existing counterpart in the literature in diffusion models. Thus Rex (ShARK) is not only a novel reversible solver, but a novel solver for diffusion models in general.

## E.1 REX AS REVERSIBLE ODE SOLVERS

Here we discuss Rex as reversible versions for well-known numerical schemes for diffusion models. Recall that the general Butcher tableau for a $s$-stage explicit RK scheme (Stewart, 2022, Section 6.1.4) is written as

$$
\begin{array}{c|ccccc}
c_1 & & & & & \\
c_2 & a_{21} & & & & \\
c_3 & a_{31} & a_{32} & & & \\
\vdots & \vdots & \vdots & \ddots & & \\
c_s & a_{s1} & a_{s2} & \cdots & a_{(s-1)s} & \\
\hline
& b_1 & b_2 & \cdots & b_{s-1} & b_s
\end{array}
= \begin{array}{c|c} c & a \\ \hline & b \end{array}.
\tag{188}
$$

Embedded methods for adaptive step sizing are of the form

$$
\begin{array}{c|ccccc}
c_1 & & & & & \\
c_2 & a_{21} & & & & \\
c_3 & a_{31} & a_{32} & & & \\
\vdots & \vdots & \vdots & \ddots & & \\
c_s & a_{s1} & a_{s2} & \cdots & a_{(s-1)s} & \\
\hline
& b_1 & b_2 & \cdots & b_{s-1} & b_s \\
& b_1^* & b_2^* & \cdots & b_{s-1}^* & b_s^*
\end{array},
\tag{189}
$$

where the lower-order step is given by the coefficients $b_i^*$.

### E.1.1 EULER

In this section we explore the numerical schemes produced by choosing the Euler scheme for $\boldsymbol{\Phi}$. The Butcher tableau for the Euler method is

$$
\begin{array}{c|c} 0 & 0 \\ \hline & 1 \end{array}.
\tag{190}
$$

**Proposition E.1** (Rex (Euler) is reversible DPM-Solver++1). *The underlying scheme of Rex (Euler) for the data prediction parameterization of diffusion models in Equation (70) is the DPM-Solver++1 from Lu et al. (2022a).*

*Proof.* Apply in the Butcher tableau for the Euler scheme to $\boldsymbol{\Psi}$ constructed from Equation (69) to find

$$
\boldsymbol{x}_{n+1} = \frac{\sigma_{n+1}}{\sigma_n} \boldsymbol{x}_n + \sigma_{n+1} h \boldsymbol{x}_{0|\gamma_n}^\theta (\boldsymbol{x}_n),
\tag{191}
$$

with $h = \gamma_{n+1} - \gamma_n$. We can rewrite the step size as

$$\sigma_{n+1}h = \sigma_{n+1}\left(\frac{\alpha_{n+1}}{\sigma_{n+1}} - \frac{\alpha_n}{\sigma_n}\right), \tag{192}$$

$$= \left(\alpha_{n+1} - \alpha_n\frac{\sigma_{n+1}}{\sigma_n}\right), \tag{193}$$

$$= \left(\alpha_{n+1}\frac{\alpha_{n+1}}{\alpha_{n+1}} - \frac{\alpha_n}{\alpha_{n+1}}\frac{\sigma_{n+1}}{\sigma_n}\right), \tag{194}$$

$$= -\alpha_{n+1}\left(\frac{\alpha_n}{\alpha_{n+1}}\frac{\sigma_{n+1}}{\sigma_n} - 1\right), \tag{195}$$

$$= -\alpha_{n+1}\left(\frac{\gamma_n}{\gamma_{n+1}} - 1\right), \tag{196}$$

$$= -\alpha_{n+1}\left(e^{\log\frac{\gamma_n}{\gamma_{n+1}}} - 1\right), \tag{197}$$

$$= -\alpha_{n+1}\left(e^{\log\gamma_n - \log\gamma_{n+1}} - 1\right), \tag{198}$$

$$\overset{(i)}{=} -\alpha_{n+1}\left(e^{\lambda_n - \lambda_{n+1}} - 1\right), \tag{199}$$

$$\overset{(ii)}{=} -\alpha_{n+1}\left(e^{-h_\lambda} - 1\right), \tag{200}$$

where (i) holds by the letting $\lambda_t = \log\gamma_t$ following the notation of Lu et al. (2022b;a) and (ii) holds by letting $h_\lambda = \lambda_{n+1} - \lambda_n$. Plugging this back into Equation (191) yields

$$\boldsymbol{x}_{n+1} = \frac{\sigma_{n+1}}{\sigma_n}\boldsymbol{x}_n - \alpha_{n+1}\left(e^{-h_\lambda} - 1\right)\boldsymbol{x}_{0|t_n}^\theta(\boldsymbol{x}_n), \tag{201}$$

which is the DPM-Solver++1 from Lu et al. (2022a). $\qquad\square$

**Corollary E.1.1** (Rex (Euler) is reversible deterministic DDIM for data prediction models). *The underlying scheme of Rex (Euler) for the data prediction parameterization of diffusion models in Equation (70) is the deterministic DDIM solver from Song et al. (2021a).*

*Proof.* This holds because DPM-Solver++1 is DDIM see Lu et al. (2022a, Equation (21)) with $\eta = 0$. $\qquad\square$

**Proposition E.2** (Rex (Euler) is reversible DPM-Solver-1). *The underlying scheme of Rex (Euler) for the data prediction parameterization of diffusion models in Equation (69) is the DPM-Solver-1 from Lu et al. (2022b, Equation (3.7)).*

*Proof.* Apply in the Butcher tableau for the Euler scheme to $\boldsymbol{\Psi}$ from Rex (see Proposition 3.3) to find

$$\boldsymbol{x}_{n+1} = \frac{\alpha_{n+1}}{\alpha_n}\boldsymbol{x}_n + \alpha_{n+1}h\boldsymbol{x}_{T|\chi_n}^\theta(\boldsymbol{x}_n), \tag{202}$$

with $h = \chi_{n+1} - \chi_n$. We can rewrite step size as

$$\alpha_{n+1} h = \alpha_{n+1} \left( \frac{\sigma_{n+1}}{\alpha_{n+1}} - \frac{\sigma_n}{\alpha_n} \right), \tag{203}$$

$$= \left( \sigma_{n+1} - \sigma_n \frac{\alpha_{n+1}}{\alpha_n} \right), \tag{204}$$

$$= \left( \sigma_{n+1} \frac{\sigma_{n+1}}{\sigma_{n+1}} - \frac{\sigma_n}{\sigma_{n+1}} \frac{\alpha_{n+1}}{\alpha_n} \right), \tag{205}$$

$$= -\sigma_{n+1} \left( \frac{\sigma_n}{\sigma_{n+1}} \frac{\alpha_{n+1}}{\alpha_n} - 1 \right), \tag{206}$$

$$= -\sigma_{n+1} \left( \frac{\chi_n}{\chi_{n+1}} - 1 \right), \tag{207}$$

$$= -\sigma_{n+1} \left( e^{\log \frac{\chi_n}{\chi_{n+1}}} - 1 \right), \tag{208}$$

$$= -\sigma_{n+1} \left( e^{\log \chi_n - \log \chi_{n+1}} - 1 \right), \tag{209}$$

$$\overset{(i)}{=} -\sigma_{n+1} \left( e^{-\lambda_n + \lambda_{n+1}} - 1 \right), \tag{210}$$

$$\overset{(ii)}{=} -\sigma_{n+1} \left( e^{h_\lambda} - 1 \right), \tag{211}$$

where (i) holds by the letting $\lambda_t = \log \gamma_t = -\log \chi_t$ following the notation of Lu et al. (2022b;a) and (ii) holds by letting $h_\lambda = \lambda_{n+1} - \lambda_n$. Plugging this back into Equation (191) yields

$$\boldsymbol{x}_{n+1} = \frac{\alpha_{n+1}}{\alpha_n} \boldsymbol{x}_n - \sigma_{n+1} \left( e^{h_\lambda} - 1 \right) \boldsymbol{x}_{T|t_n}^\theta(\boldsymbol{x}_n), \tag{212}$$

which is the DPM-Solver-1 from Lu et al. (2022b). $\qquad\square$

**Corollary E.2.1** (Rex (Euler) is reversible deterministic DDIM for noise prediction models). *The underlying scheme of Rex (Euler) for the noise prediction parameterization of diffusion models in Equation (69) is the deterministic DDIM solver from Song et al. (2021a).*

*Proof.* This holds because DPM-Solver-1 is DDIM see Lu et al. (2022b, Equation (4.1)). $\qquad\square$

### E.1.2 SECOND-ORDER METHODS

In this section we explore the numerical schemes produced by choosing the explicit midpoint method for $\boldsymbol{\Phi}$. We can write a generic second-order method as

$$
\begin{array}{c|cc}
0 & & \\
\eta & \eta & \\
\hline
& 1 - \frac{1}{2\eta} & \frac{1}{2\eta}
\end{array}, \tag{213}
$$

for $\eta \neq 0$ (Butcher, 2016). The choice of $\eta = \frac{1}{2}$ yields the explicit midpoint, $\eta = \frac{2}{3}$ gives Ralston's second-order method, and $\eta = 1$ gives Heun's second-order method.

**Proposition E.3** (Rex (generic second-order) is reversible DPM-Solver++(2S)). *The underlying scheme of Rex (generic second-order) for the data prediction parameterization of diffusion models in Equation (70) is the DPM-Solver++(2S) from Lu et al. (2022a, Algorithm 1).*

*Proof.* The DPM-Solver++(2S) (Lu et al., 2022a, Algorithm 1) is defined as

$$\boldsymbol{u} = \frac{\sigma_p}{\sigma_n} \boldsymbol{x}_n - \alpha_p \left( e^{-r_\lambda h_\lambda} - 1 \right) \boldsymbol{x}_{0|t_n}^\theta(\boldsymbol{x}_n),$$

$$\boldsymbol{D} = \left( 1 - \frac{1}{2r_\lambda} \right) \boldsymbol{x}_{0|t_n}^\theta(\boldsymbol{x}_n) + \frac{1}{2r_\lambda} \boldsymbol{x}_{0|t_p}^\theta(\boldsymbol{u}), \tag{214}$$

$$\boldsymbol{x}_{n+1} = \frac{\sigma_{n+1}}{\sigma_n} \boldsymbol{x}_n - \alpha_{n+1} \left( e^{-h_\lambda} - 1 \right) \boldsymbol{D},$$

for some intermediate timestep $t_n > t_p > t_{n+1}$ and with $r_\lambda = \frac{\lambda_p - \lambda_n}{\lambda_{n+1} - \lambda_n}$. Notice that $r_\lambda$ describes the location of the midpoint time in the $\lambda$-domain as a ratio, *i.e.*, we could say

$$\lambda_p = \lambda_n + r_\lambda h_\lambda, \tag{215}$$

where $r_\lambda \in (0, 1)$ denotes the interpolation point between the initial timestep $\lambda_n$ and terminal timestep $\lambda_{n+1}$. Thus we fix $\eta = r_\lambda$ as the step size ratio of the intermediate point.

Now we return to the underlying scheme of Rex applied to the generic second-order scheme, see Equation (213), Apply in the Butcher tableau for generic second-order scheme to $\boldsymbol{\Psi}$ constructed from Equation (69) to find

$$\boldsymbol{z} = \frac{1}{\sigma_n} \boldsymbol{x}_n + \eta h \boldsymbol{x}_{0|\gamma_n}^\theta(\boldsymbol{x}_n),$$
$$\boldsymbol{x}_{n+1} = \frac{\sigma_{n+1}}{\sigma_n} \boldsymbol{x}_n + \sigma_{n+1} h \left( \left(1 - \frac{1}{2\eta}\right) \boldsymbol{x}_{0|\gamma_n}^\theta(\boldsymbol{x}_n) + \frac{1}{2\eta} \boldsymbol{x}_{0|\gamma_n + \eta h}^\theta(\sigma_p \boldsymbol{z}) \right), \tag{216}$$

with $h = \gamma_{n+1} - \gamma_n$ and $\sigma_p = \sigma_{\gamma_n + \eta h}$ with $\gamma_p = \gamma_n + \eta h$. We can write

$$\sigma_p \boldsymbol{z} = \frac{\sigma_p}{\sigma_n} \boldsymbol{x}_n + \sigma_p \eta h \boldsymbol{x}_{0|\gamma_n}^\theta(\boldsymbol{x}_n). \tag{217}$$

Plugging this back into Equation (216) yields

$$\sigma_p \boldsymbol{z} = \frac{\sigma_p}{\sigma_n} \boldsymbol{x}_n + \sigma_p \eta h \boldsymbol{x}_{0|\gamma_n}^\theta(\boldsymbol{x}_n),$$
$$\boldsymbol{x}_{n+1} = \frac{\sigma_{n+1}}{\sigma_n} \boldsymbol{x}_n + \sigma_{n+1} h \underbrace{\left( \left(1 - \frac{1}{2\eta}\right) \boldsymbol{x}_{0|\gamma_n}^\theta(\boldsymbol{x}_n) + \frac{1}{2\eta} \boldsymbol{x}_{0|\gamma_n + \eta h}^\theta(\sigma_p \boldsymbol{z}) \right)}_{= \hat{\boldsymbol{D}}}, \tag{218}$$

which is the DPM-Solver++1 from Lu et al. (2022a). Now recall from Proposition E.1 that

$$\sigma_{n+1} h = -\alpha_{n+1} \left( e^{-h_\lambda} - 1 \right), \tag{219}$$

it follows that

$$\sigma_p \eta h = -\alpha_p \left( e^{-r_\lambda h_\lambda} - 1 \right), \tag{220}$$

due to $\lambda_p - \lambda_n = r_\lambda h_\lambda$ and $\eta h = \lambda_p - \lambda_n$. Thus by letting $\sigma_p \boldsymbol{z} = \boldsymbol{u}$ and $\hat{\boldsymbol{D}} = \boldsymbol{D}$ we recover the DPM-Solver++(2S) solver. $\square$

**Proposition E.4** (Rex (generic second-order) is reversible DPM-Solver-2)). *The underlying scheme of Rex (generic second-order) for the noise prediction parameterization of diffusion models in Equation (69) is the DPM-Solver-2 from Lu et al. (2022b, Algorithm 4 cf. Algorithm 1).*

*Proof.* This follows as straightforward derivation from Proposition E.2 and Proposition E.3. $\square$

**Proposition E.5** (Rex (Euler-Midpoint) is DPM-Solver-12). *The underlying scheme of Rex (Euler-Midpoint) for the noise prediction parameterization of diffusion models in Equation (69) is the DPM-Solver-12 from Lu et al. (2022b).*

*Proof.* The explicit midpoint method with embedded Euler method for adaptive step sizing is given by the Butcher tableau

$$
\begin{array}{c|cc}
0 & & \\
\frac{1}{2} & \frac{1}{2} & \\
\hline
& 0 & 1 \\
& 1 & 0
\end{array}. \tag{221}
$$

From Proposition E.2 and Proposition E.4 we have shown that Rex (Euler) and Rex (Midpoint) correspond to DPM-Solver-1 and DPM-Solver-2 respectively. Thus the Butcher tableau above outlines DPM-Solver-12. $\square$

### E.1.3 THIRD-ORDER METHODS

For third-order solvers like DPM-Solver-3 (Lu et al., 2022b, Algorithm 5) our constructed scheme differs from solvers derived using ETD methods due to the presence of $\varphi_2$ terms where

$$\varphi_{k+1}(t) = \int_0^1 e^{(1-\delta)t} \frac{\delta^k}{k!} \, \mathrm{d}\delta, \tag{222}$$

this also reasoning extends to the DPM-Solver-4 from Gonzalez et al. (2024, Algorithm 7).

### E.2 REX AS REVERSIBLE SDE SOLVERS

In this section we discuss the connections between Rex and preexisting SDE solvers for diffusion models.

### E.2.1 EULER-MARUYAMA

The extended Butcher tableau for the Euler-Maruyama scheme is given by

$$
\begin{array}{c|c|c|c}
0 & 0 & 0 & 0 \\
\hline
 & 1 & 1 & 0
\end{array}. \tag{223}
$$

**Proposition E.6** (Rex (Euler-Maruyama) is reversible SDE-DPM-Solver++1). *The underlying scheme of Rex (Euler-Maruyama) for the data prediction parameterization of diffusion models in Equation* (10) *is the SDE-DPM-Solver++1 from Lu et al. (2022a, Equation (18)).*

*Proof.* Apply in the Butcher tableau for the Euler-Maruyama scheme to $\Psi$ constructed from Equation (133) to find

$$\boldsymbol{x}_{n+1} = \frac{\sigma_{n+1}^2 \alpha_n}{\sigma_n^2 \alpha_{n+1}} \boldsymbol{x}_n + \frac{\sigma_{n+1}^2}{\alpha_{n+1}} h \boldsymbol{x}_{0|\varrho_n}^\theta(\boldsymbol{x}_n) + \frac{\sigma_{n+1}^2}{\alpha_{n+1}} \boldsymbol{W}_n, \tag{224}$$

with $h = \varrho_{n+1} - \varrho_n$. We can rewrite the step size as

$$\frac{\sigma_{n+1}^2}{\alpha_{n+1}} h = \frac{\sigma_{n+1}^2}{\alpha_{n+1}} \left( \frac{\alpha_{n+1}^2}{\sigma_{n+1}^2} - \frac{\alpha_n^2}{\sigma_n^2} \right), \tag{225}$$

$$= \left( \alpha_{n+1} - \frac{\alpha_n^2}{\alpha_{n+1}} \frac{\sigma_{n+1}^2}{\sigma_n^2} \right), \tag{226}$$

$$= \alpha_{n+1} \left( 1 - \frac{\alpha_n^2}{\alpha_{n+1}^2} \frac{\sigma_{n+1}^2}{\sigma_n^2} \right), \tag{227}$$

$$= \alpha_{n+1} \left( 1 - \frac{\varrho_n}{\varrho_{n+1}} \right), \tag{228}$$

$$= \alpha_{n+1} \left( 1 - e^{2 \log \frac{\gamma_n}{\gamma_{n+1}}} \right), \tag{229}$$

$$= \alpha_{n+1} \left( 1 - e^{2 \log \gamma_n - 2 \log \gamma_{n+1}} \right), \tag{230}$$

$$\overset{(i)}{=} \alpha_{n+1} \left( 1 - e^{2\lambda_n - 2\lambda_{n+1}} \right), \tag{231}$$

$$\overset{(ii)}{=} \alpha_{n+1} \left( 1 - e^{-2h_\lambda} \right), \tag{232}$$

where (i) holds by the letting $\lambda_t = \log \gamma_t$ following the notation of Lu et al. (2022b;a) and (ii) holds by letting $h_\lambda = \lambda_{n+1} - \lambda_n$. Now recall that

$$\frac{\sigma_{n+1}^2 \alpha_n}{\sigma_n^2 \alpha_{n+1}} = \frac{\sigma_{n+1}}{\sigma_n} e^{-h_\lambda}. \tag{233}$$

Plugging these back into Equation (224) yields

$$\boldsymbol{x}_{n+1} = \frac{\sigma_{n+1}}{\sigma_n} e^{-h_\lambda} \boldsymbol{x}_n + \alpha_{n+1} \left( 1 - e^{-2h_\lambda} \right) \boldsymbol{x}_{0|t_n}^\theta(\boldsymbol{x}_n) + \frac{\sigma_{n+1}^2}{\alpha_n} \boldsymbol{W}_n. \tag{234}$$

Now recall that the Brownian increment $\boldsymbol{W}_n := \boldsymbol{W}_{\varrho_{n+1}} - \boldsymbol{W}_{\varrho_n}$ has variance $h$. Thus via the Itô isometry we can write

$$\boldsymbol{W}_n \sim \sqrt{h}\boldsymbol{\epsilon}, \tag{235}$$

with $\boldsymbol{\epsilon} \sim \mathcal{N}(\boldsymbol{0}, \boldsymbol{I})$. Then we have

$$\frac{\sigma_{n+1}^2}{\alpha_{n+1}}\sqrt{h} = \frac{\sigma_{n+1}^2}{\alpha_{n+1}}\sqrt{\frac{\alpha_{n+1}^2}{\sigma_{n+1}^2} - \frac{\alpha_n^2}{\sigma_n^2}}, \tag{236}$$

$$= \sqrt{\sigma_{n+1}^2 - \frac{\alpha_n^2}{\alpha_{n+1}^2}\frac{\sigma_{n+1}^4}{\sigma_n^2}}, \tag{237}$$

$$= \sigma_{n+1}\sqrt{1 - \frac{\alpha_n^2}{\alpha_{n+1}^2}\frac{\sigma_{n+1}^2}{\sigma_n^2}}, \tag{238}$$

$$= \sigma_{n+1}\sqrt{1 - \frac{\varrho_n}{\varrho_{n+1}}}, \tag{239}$$

$$= \sigma_{n+1}\sqrt{1 - e^{-2h_\lambda}}. \tag{240}$$

Thus we have re-derived the noise term of the SDE-DPM-Solver++1, and putting everything together we have obtained the SDE-DPM-Solver++1 from Lu et al. (2022a) which is

$$\boldsymbol{x}_{n+1} = \frac{\sigma_{n+1}}{\sigma_n}e^{-h_\lambda}\boldsymbol{x}_n + \alpha_{n+1}\left(1 - e^{-2h_\lambda}\right)\boldsymbol{x}_{0|t_n}^\theta(\boldsymbol{x}_n) + \sigma_{n+1}\sqrt{1 - e^{-2h_\lambda}}\boldsymbol{\epsilon}. \tag{241}$$

Thus we have shown that the SDE-DPM-Solver++1 is the same as the underlying scheme of Rex (Euler-Maruyama). $\qquad\square$

**Corollary E.6.1** (Rex (Euler-Maruyama) is reversible stochastic DDIM). *The underlying scheme of Rex (Euler-Maruyama) for the data prediction parameterization of diffusion models in Equation* (10) *is the stochastic DDIM solver from Song et al. (2021a) with $\eta = \sigma_t\sqrt{1 - e^{-2h_\lambda}}$.*

*Proof.* This holds because SDE-DPM-Solver-1 is DDIM see Lu et al. (2022a, Section 6.1). $\qquad\square$

**Proposition E.7** (Rex (Euler-Maruyama) is reversible SDE-DPM-Solver-1). *The underlying scheme of Rex (Euler-Maruyama) for the noise prediction parameterization of diffusion models in Equation* (133) *is the SDE-DPM-Solver-1 from Lu et al. (2022a, Equation (17)).*

*Proof.* Apply in the Butcher tableau for the Euler scheme to $\boldsymbol{\Psi}$ from Rex (see Proposition 3.3) to find

$$\boldsymbol{x}_{n+1} = \frac{\alpha_{n+1}}{\alpha_n}\boldsymbol{x}_n + 2\alpha_{n+1}h\boldsymbol{x}_{T|\chi_n}^\theta(\boldsymbol{x}_n) + \alpha_{n+1}\boldsymbol{W}_n, \tag{242}$$

with $h = \chi_{n+1} - \chi_n$. Recall from Proposition E.2 that we can rewrite the step size

$$\alpha_{n+1}h = -\sigma_{n+1}\left(e^{h_\lambda} - 1\right). \tag{243}$$

Now recall that the Brownian increment $\boldsymbol{W}_n := \overline{\boldsymbol{W}}_{\chi_{n+1}^2} - \overline{\boldsymbol{W}}_{\chi_n^2}$ has variance $\chi_n^2 - \chi_{n+1}^2$.[20] Thus via the Itô isometry we can write

$$\boldsymbol{W}_n \sim \sqrt{\chi_n^2 - \chi_{n+1}^2}\boldsymbol{\epsilon}, \tag{244}$$

---

[20]This is because $\overline{\boldsymbol{W}}_\chi^2$ is defined in reverse-time.

with $\epsilon \sim \mathcal{N}(\mathbf{0}, \boldsymbol{I})$. Then we have

$$\alpha_{n+1}\sqrt{\chi_n^2 - \chi_{n+1}^2} = \alpha_{n+1}\sqrt{\frac{\sigma_n^2}{\alpha_n^2} - \frac{\sigma_{n+1}^2}{\alpha_{n+1}^2}}, \tag{245}$$

$$= \sqrt{\frac{\sigma_n^2 \alpha_{n+1}^2}{\alpha_n^2} - \sigma_{n+1}^2}, \tag{246}$$

$$= \sigma_{n+1}\sqrt{\frac{\sigma_n^2 \alpha_{n+1}^2}{\sigma_{n+1}^2 \alpha_n^2} - 1}, \tag{247}$$

$$= \sigma_{n+1}\sqrt{\frac{\chi_n^2}{\chi_{n+1}^2} - 1}, \tag{248}$$

$$= \sigma_{n+1}\sqrt{e^{\log \frac{\chi_n^2}{\chi_{n+1}^2}} - 1}, \tag{249}$$

$$= \sigma_{n+1}\sqrt{e^{\log \chi_n^2 - \log \chi_{n+1}^2} - 1}, \tag{250}$$

$$= \sigma_{n+1}\sqrt{e^{-2\log \gamma_n + 2\log \gamma_{n+1}} - 1}, \tag{251}$$

$$= \sigma_{n+1}\sqrt{e^{2\log \lambda_{n+1} - 2\log \lambda_n} - 1}, \tag{252}$$

$$= \sigma_{n+1}\sqrt{e^{2h_\lambda} - 1}. \tag{253}$$

Plugging Equations (243) and (253) back into Equation (242) yields

$$\boldsymbol{x}_{n+1} = \frac{\alpha_{n+1}}{\alpha_n}\boldsymbol{x}_n - 2\sigma_{n+1}\left(e^{h_\lambda} - 1\right)\boldsymbol{x}_{T|\chi_n}^\theta(\boldsymbol{x}_n) + \sigma_{n+1}\sqrt{e^{2h_\lambda} - 1}\,\epsilon, \tag{254}$$

which is the SDE-DPM-Solver-1 from Lu et al. (2022a). $\qquad\square$

**Corollary E.7.1** (Rex (Euler-Maruyama) is reversible stochastic DDIM for noise prediction models)**.** *The underlying scheme of Rex (Euler-Maruyama) for the noise prediction parameterization of diffusion models in Equation* (133) *is the stochastic DDIM solver from Song et al. (2021a) with* $\eta = \sigma_t\sqrt{e^{-2h_\lambda} - 1}$.

*Proof.* This follows from a straightforwardly from Corollary E.6.1 and Lu et al. (2022b, Equation (4.1)). $\qquad\square$

### E.3 REX AS REVERSIBLE SEEDS-1

**Proposition E.8** (Rex is reversible SEEDS-1)**.** *The choice of Euler or Euler-Maruyama for the underlying scheme of Rex with either the noise prediction parameterization of diffusion models in Equations* (69) *and* (133) *or data prediction in Equations* (10) *and* (69) *yields the four variants of SEEDS-1 outlined in Gonzalez et al. (2024, Equations (28-31)).*

*Proof.* This follows straightforwardly from Propositions E.1, E.2, E.6 and E.7 by definition of SEEDS-1. $\qquad\square$

**Corollary E.8.1** (Rex (Euler-Maruyama) is reversible gDDIM)**.** *The underlying scheme of Rex (Euler-Maruyama) for the data prediction parameterization of diffusion models in Equation* (10) *is the gDDIM solver in Zhang et al. (2023, Theorem 1) for* $\ell = 1$.

*Proof.* This follows as an immediate consequence of Proposition E.8 since by Gonzalez et al. (2024, Proposition 4.5) gDDIM is SEEDS-1. $\qquad\square$

As mentioned earlier in Section A.4.1 high-order variants of SEEDS use a Markov-preserving noise decomposition to approximate the iterated stochastic integrals. However, we follow Foster et al. (2024) and use the space-time Lévy area resulting in numerical schemes that are quite different beyond the first-order case, albeit that Rex exhibits better convergence properties.

## F    A BRIEF NOTE ON THE THEORY OF ROUGH PATHS

To perform reversibility it is useful to consider the pathwise interpretation of SDEs (Lyons, 1998), as such we introduce a few notations from rough path theory. Let $\{\boldsymbol{W}_t\}$ be a $d_w$-dimensional Brownian motion and let $\boldsymbol{W}$ be enhanced by

$$\mathbb{W}_{s,t} = \int_s^t \boldsymbol{W}_{s,r} \otimes \mathrm{od}\boldsymbol{W}_r, \tag{255}$$

where $\otimes$ is the tensor product. Then, the pair $\mathcal{W} := (\boldsymbol{W}, \mathbb{W})$ is the *Stratonovich enhanced Brownian rough path*.[21] Thus consider the $d_x$-dimensional *rough differential equation* RDE of the form:

$$\mathrm{d}\boldsymbol{X}_t = \boldsymbol{\mu}(t, \boldsymbol{X}_t) \, \mathrm{d}t + \boldsymbol{\sigma}(t, \boldsymbol{X}_t) \, \mathrm{d}\mathcal{W}_t, \qquad \boldsymbol{X}_0 = \boldsymbol{x}_0. \tag{256}$$

where $\boldsymbol{\mu} : [0, T] \times \mathbb{R}^{d_x} \to \mathbb{R}^{d_x}$ is Lipschitz continuous in its second argument and $\boldsymbol{\sigma} \in \mathcal{C}_b^{1,3}([0, T] \times \mathbb{R}^{d_x}; \mathcal{L}(\mathbb{R}^{d_w}, \mathbb{R}^{d_x}))$ (Friz & Hairer, 2020, Theorem 9.1).[22] Fix an $\omega \in \Omega$, then almost surely $\mathcal{W}(\omega)$ admits a unique solution to the RDE $(\boldsymbol{X}_t(\omega), \boldsymbol{\sigma}(t, \boldsymbol{X}_t(\omega)))$ and $\boldsymbol{X}_t = \boldsymbol{X}_t(\omega)$ is a strong solution to the Stratonovich SDE[23] started at $\boldsymbol{X}_0 = \boldsymbol{x}_0$. To elucidate, consider the commutative diagram below

$$\boldsymbol{W} \overset{\Psi}{\longmapsto} (\boldsymbol{W}, \mathbb{W}) \overset{S}{\longmapsto} \boldsymbol{X}, \tag{257}$$

where $\Psi$ is a map which merely lifts Brownian motion into a rough path (could be Itô or Stratonovich), the second map, $S$, is known as the *Itô-Lyons map* (Lyons, 1998); this map is purely deterministic and is also a *continuous map* w.r.t. to initial condition and driving signal. Thus for a fixed realization of the Brownian motion we have a pathwise interpretation of the Stratonovich SDE.

## G    NUMERICAL SIMULATION OF BROWNIAN MOTION

Earlier we mentioned that for reversible methods we need to be able to compute both the *same* realization of the Brownian motion. Now sampling Brownian motion is quite simple—recall Lévy's characterization of Brownian motion (Øksendal, 2003, Theorem 8.6.1)—and can be sampled by drawing independent Gaussian increments during the numerical solve of an SDE. A common choice for an adaptive solver is to use Lévy's Brownian bridge formula (Revuz & Yor, 2013).

**Definition G.1** (Lévy's Brownian bridge). Given the standard $d_w$-dimensional Brownian motion $\{\boldsymbol{W}_t : t \geq 0\}$ and for any $0 \leq s < t < u$, the Brownian bridge is defined as

$$\boldsymbol{W}_t | \boldsymbol{W}_s, \boldsymbol{W}_u \sim \mathcal{N}\left(\boldsymbol{W}_s + \frac{t-s}{u-s}(\boldsymbol{W}_u - \boldsymbol{W}_s), \frac{(u-t)(t-s)}{u-s}\boldsymbol{I}\right), \tag{258}$$

and this quantity is conditionally independent of $\boldsymbol{W}_v$ for $v < s$ or $v > u$.

Sampling the Brownian motion in reverse-time, however, is more complicated as it is only adapted to the natural filtration defined in forward time. The naïve approach to sampling Brownian motion, called the *Brownian path*, is to simply store the entire realization of the Brownian motion from the forward pass in memory and use Equation (258) when necessary (for adaptive step size methods). This results in a query time of $\mathcal{O}(1)$, but with a memory cost of $\mathcal{O}(nd_w)$, where $n$ is the number of samples.

**Virtual Brownian Tree.**    Seminal work on neural SDEs by Li et al. (2020) introduced the *Virtual Brownian Tree* which extends the concept of Brownian trees introduced by Gaines & Lyons (1997). The Brownian tree recursively applies Equation (258) to sample the Brownian motion at any midpoint, constructing a tree structure; however, storing such a tree would be memory intensive. By making

---

[21]See, Friz & Hairer (2020, Chapter 3) for more details.

[22]Here $\mathcal{L}(V, W)$ denotes the set of continuous maps from $V$ to $W$, a Banach space.

[23]If $\boldsymbol{X}_t$ and $\partial_{\boldsymbol{x}}\boldsymbol{X}_t$ are adapted and $\langle \boldsymbol{X}, \boldsymbol{W}\rangle_t$ exists, then almost surely

$$\int_0^T \boldsymbol{X} \mathrm{d}\mathcal{W}_t = \int_0^T \boldsymbol{X} \circ \mathrm{d}\boldsymbol{W}_t.$$

use of splittable *pseudo-random number generators* PRNGs (Salmon et al., 2011; Claessen & Pałka, 2013) which can deterministically generate two random seeds given an existing seed. Then making use of a splittable PRNG one can evaluate the Brownian motion at any point by recursively applying the Brownian tree constructing to rebuild the tree until the recursive midpoint time $t_r$ is suitable *close* to the desired timestep $t$, *i.e.*, $|t - t_r| < \epsilon$ for some fixed error threshold $\epsilon > 0$. This requires constant $\mathcal{O}(1)$ memory but takes $\mathcal{O}(\log(1/\epsilon))$ time and is only *approximate*.

**Brownian Interval.** Closely related work by Kidger et al. (2021) introduces the *Brownian Interval* which offers exact sampling with $\mathcal{O}(1)$ query times. The primary difference between this method and Virtual Brownian Trees is that this method focuses on intervals rather than particular sample points. To elucidate, let $\boldsymbol{W}_{s,t} = \boldsymbol{W}_t - \boldsymbol{W}_s$ denote an interval of Brownian motion. Then the formula for Lévy's Brownian bridge (258) can be rewritten in terms of Brownian intervals as

$$\boldsymbol{W}_{s,t}|\boldsymbol{W}_{s,u} \sim \mathcal{N}\left(\frac{t-s}{u-s}\boldsymbol{W}_{s,u}, \frac{(u-t)(s-u)}{u-s}\boldsymbol{I}\right). \tag{259}$$

Then, the method constructs a tree with stump being the global interval $[0, T]$ and a random seed for a splittable PRNG. New leaf nodes are constructed when queries over intervals are made; this provides the advantage of the tree being query-dependent unlike the Virtual Brownian Tree which has a fixed dyadic structure. Further computational improvements are made to improve implementation with the details being found in Kidger (2022, Section 5.5.3). Beyond the numerical efficiency in computing intervals over points is that we regularly need use intervals in numeric schemes and not single sample points. Often, solvers which approximate higher-order integrals (*e.g.*, stochastic Runge-Kutta) require samples of the Lévy area[24] which would require the Brownian interval to construct.[25]

**Updated Virtual Brownian Tree.** Recent work by Jelinčič et al. (2024) improves upon the Virtual Brownian Tree (Li et al., 2020) by using an interpolation strategy between query points.[26] This enables the updated algorithm to exactly match the distribution of Brownian motion and Lévy areas at all query times as long as each query time is at least $\epsilon$ apart.

## H IMPLEMENTATION DETAILS

### H.1 CLOSED FORM EXPRESSIONS OF THE NOISE SCHEDULE

In practice, popular libraries like the `diffusers` library define the noise schedule for diffusion models as a discrete schedule $\{\beta_n\}_{n=1}^N$ following Ho et al. (2020); Song et al. (2021a) as an arithemetic sequence of the form

$$\beta_n = \frac{\beta_0}{N} + \frac{n-1}{N(N-1)}(\beta_1 - \beta_0), \tag{260}$$

with hyperparameters $\beta_0, \beta_1 \in \mathbb{R}_{\geq 0}$. Song et al. (2021b) defines the continuous-time schedule as

$$\beta_t = \beta_0 + t(\beta_1 - \beta_0), \tag{261}$$

for all $t \in [0, 1]$ in the limit of $N \to \infty$. Thus one can write the forward-time diffusion (variance preserving) SDE as

$$\mathrm{d}\boldsymbol{X}_t = -\frac{1}{2}\beta_t\boldsymbol{X}_t\,\mathrm{d}t + \sqrt{\beta_t}\,\mathrm{d}\boldsymbol{W}_t. \tag{262}$$

Thus we can express the noise schedule $(\alpha_t, \sigma_t)$ as

$$\alpha_t = \exp\left(-\frac{1}{2}\int \beta_t\,\mathrm{d}t\right),$$
$$\sigma_t = \sqrt{1 - \alpha_t^2}. \tag{263}$$

---

[24]*I.e.*, for a $d_w$-dimensional Brownian motion over $[s, t]$ the Lévy area is

$$2\boldsymbol{L}_{s,t}^{i,j} := \int_s^t \boldsymbol{W}_{s,u}^i \mathrm{d}\boldsymbol{W}_u^j - \int_s^t \boldsymbol{W}_{s,u}^j \mathrm{d}\boldsymbol{W}_u^i.$$

[25]The interested reader can find more details in James Foster's thesis (Foster, 2020).

[26]This algorithm is a part of the popular `Diffrax` library.

*N.B.*, often the hyperparmeters in libraries like `diffusers` are expressed as $\hat{\beta}_0 = \frac{\beta_0}{N}$ and $\hat{\beta}_1 = \frac{\beta_1}{N}$, often with $N = 1000$.

### H.1.1 LINEAR NOISE SCHEDULE

For the linear noise schedule in Equation (261) used by DDPMs (Ho et al., 2020), the schedule $(\alpha_t, \sigma_t)$ is written as

$$\alpha_t = \exp\left(-\frac{\beta_1 - \beta_0}{4}t^2 - \frac{\beta_0}{2}t\right),$$
$$\sigma_t = \sqrt{1 - \alpha_t^2}, \tag{264}$$

for $t \in [0, 1]$ with hyperparameters $\beta_0$ and $\beta_1$.

**Proposition H.1** (Inverse function of $\gamma_t$ for linear noise schedule)**.** *For the linear noise schedule used by DDPMs (Ho et al., 2020) the inverse function of $\gamma_t$ denoted $t_\gamma$ can be expressed in closed form as*

$$t_\gamma(\gamma) = \frac{-\beta_0 + \sqrt{\beta_0^2 + 2(\beta_1 - \beta_0)\log(\gamma^{-2} + 1)}}{\beta_1 - \beta_0}. \tag{265}$$

*Proof.* Let $\alpha_t$ be denoted by $\alpha_t = e^{a_t}$ where

$$a_t = -\frac{\beta_1 - \beta_0}{4}t^2 - \frac{\beta_0}{2}t. \tag{266}$$

Then by definition of $\gamma_t$ we can write

$$\gamma_t = \frac{e^{a_t}}{\sqrt{1 - e^{2a_t}}}, \tag{267}$$

and with a little more algebra we find

$$\sqrt{1 - e^{2a_t}} = \frac{e^{a_t}}{\gamma_t}, \tag{268}$$

$$1 - e^{2a_t} = \frac{e^{2a_t}}{\gamma_t^2}, \tag{269}$$

$$e^{-2a_t} - 1 = \gamma_t^{-2}, \tag{270}$$

$$e^{-2a_t} = \gamma_t^{-2} + 1, \tag{271}$$

$$-2a_t = \log(\gamma_t^{-2} + 1). \tag{272}$$

Then by substituting in the definition of $a_t$ and letting $\gamma$ denote the variable produced by $\gamma_t$ we have

$$\frac{\beta_1 - \beta_0}{2}t^2 + \beta_0 t - \log(\gamma^{-2} + 1) = 0. \tag{273}$$

We then use the quadratic formula to find the roots of the polynomial of $t$ to find

$$t = \frac{-\beta_0 \pm \sqrt{\beta_0^2 + 2(\beta_1 - \beta_0)\log(\gamma^{-2} + 1)}}{\beta_1 - \beta_0}. \tag{274}$$

Since $t \in [0, 1]$ we only take the positive root and thus

$$t = \frac{-\beta_0 + \sqrt{\beta_0^2 + 2(\beta_1 - \beta_0)\log(\gamma^{-2} + 1)}}{\beta_1 - \beta_0}. \tag{275}$$

$\square$

**Corollary H.1.1** (Inverse function of $\chi_t$ for linear noise schedule)**.** *It follows by a straightforward substitution from Proposition H.1 that $t_\chi$ can be written as*

$$t_\chi(\chi) = \frac{-\beta_0 + \sqrt{\beta_0^2 + 2(\beta_1 - \beta_0)\log(\chi^2 + 1)}}{\beta_1 - \beta_0}. \tag{276}$$

**Corollary H.1.2** (Inverse function of $\varrho_t$ for linear noise schedule)**.** *It follows by a straightforward substitution from Proposition H.1 that $t_\varrho$ can be written as*

$$t_\varrho(\varrho) = \frac{-\beta_0 + \sqrt{\beta_0^2 + 2(\beta_1 - \beta_0)\log(\varrho^{-1} + 1)}}{\beta_1 - \beta_0}. \tag{277}$$

### H.1.2 SCALED LINEAR SCHEDULE

The *scaled linear schedule* is used widely by *latent diffusion models* (LDMs) (Rombach et al., 2022) and takes the discrete form of

$$\beta_n = \left( \sqrt{\hat{\beta}_0} + \frac{n-1}{N-1} \left( \sqrt{\hat{\beta}_1} - \sqrt{\hat{\beta}_0} \right) \right)^2. \tag{278}$$

Thus following a similar approach to Song et al. (2021b) we write the scaled linear schedule as a function of $t$,

$$\beta_t = (\beta_1 - 2\sqrt{\beta_1\beta_0} + \beta_0)t^2 + 2t(\sqrt{\beta_1\beta_0} - \beta_0) + \beta_0. \tag{279}$$

Then using Equation (263) we find the noise schedule $(\alpha_t, \sigma_t)$ to be defined as

$$\alpha_t = \exp\left( -\frac{\beta_1 - 2\sqrt{\beta_1\beta_0} + \beta_0}{6}t^3 - \frac{\sqrt{\beta_1\beta_0} - \beta_0}{2}t^2 - \frac{\beta_0}{2}t \right),$$
$$\sigma_t = \sqrt{1 - \alpha_t^2}. \tag{280}$$

Next we will derive the inverse function for $\gamma_t$

**Proposition H.2** (Inverse function of $\gamma_t$ for scaled linear noise schedule). *For the scaled linear noise schedule commonly used by LDMs (Rombach et al., 2022) the inverse function of $\gamma_t$ denoted $t_\gamma$ can be expressed in closed form as*

$$t_\gamma(\gamma) = \frac{\beta_0 - \sqrt{\beta_1\beta_0} - \sqrt[3]{2(\sqrt{\beta_1\beta_0} - \beta_0)^3 - 3\beta_0\Delta(\sqrt{\beta_1\beta_0} - \beta_0) - 3\Delta^2\log(\gamma^{-2} + 1)}}{\Delta}, \tag{281}$$

*where*

$$\Delta = \beta_1 - 2\sqrt{\beta_1\beta_0} + \beta_0. \tag{282}$$

*Proof.* Let $\alpha_t$ be denoted by $\alpha_t = e^{a_t}$ where

$$a_t = -\frac{\beta_1 - 2\sqrt{\beta_1\beta_0} + \beta_0}{6}t^3 - \frac{\sqrt{\beta_1\beta_0} - \beta_0}{2}t^2 - \frac{\beta_0}{2}t. \tag{283}$$

Then by definition of $\gamma_t$ we can write

$$\gamma_t = \frac{e^{a_t}}{\sqrt{1 - e^{2a_t}}}, \tag{284}$$

and with a little more algebra we find

$$\sqrt{1 - e^{2a_t}} = \frac{e^{a_t}}{\gamma_t}, \tag{285}$$

$$1 - e^{2a_t} = \frac{e^{2a_t}}{\gamma_t^2}, \tag{286}$$

$$e^{-2a_t} - 1 = \gamma_t^{-2}, \tag{287}$$

$$e^{-2a_t} = \gamma_t^{-2} + 1, \tag{288}$$

$$-2a_t = \log(\gamma_t^{-2} + 1). \tag{289}$$

Then by substituting in the definition of $a_t$ and letting $\gamma$ denote the variable produced by $\gamma_t$ we have

$$\frac{\beta_1 - 2\sqrt{\beta_1\beta_0} + \beta_0}{3}t^3 + (\sqrt{\beta_1\beta_0} - \beta_0)t^2 + \beta_0 t - \log(\gamma^{-2} + 1) = 0. \tag{290}$$

We then use the cubic formula (Cardano, 1545) to find the roots of the polynomial of $t$. The only real root is given by

$$t_\gamma(\gamma) = \frac{\beta_0 - \sqrt{\beta_1\beta_0} - \sqrt[3]{2(\sqrt{\beta_1\beta_0} - \beta_0)^3 - 3\beta_0\Delta(\sqrt{\beta_1\beta_0} - \beta_0) - 3\Delta^2\log(\gamma^{-2} + 1)}}{\Delta}, \tag{291}$$

where

$$\Delta = \beta_1 - 2\sqrt{\beta_1\beta_0} + \beta_0. \tag{292}$$

$\square$

**Corollary H.2.1** (Inverse function of $\chi_t$ for scaled linear noise schedule)**.** *It follows by a straightforward substitution from Proposition H.2 that $t_\chi$ can be written as*

$$t_\chi(\chi) = \frac{\beta_0 - \sqrt{\beta_1 \beta_0} - \sqrt[3]{2(\sqrt{\beta_1 \beta_0} - \beta_0)^3 - 3\beta_0 \Delta(\sqrt{\beta_1 \beta_0} - \beta_0) - 3\Delta^2 \log(\chi^2 + 1)}}{\Delta}, \quad (293)$$

*where*

$$\Delta = \beta_1 - 2\sqrt{\beta_1 \beta_0} + \beta_0. \quad (294)$$

**Corollary H.2.2** (Inverse function of $\varrho_t$ for scaled linear noise schedule)**.** *It follows by a straightforward substitution from Proposition H.2 that $t_\varrho$ can be written as*

$$t_\varrho(\varrho) = \frac{\beta_0 - \sqrt{\beta_1 \beta_0} - \sqrt[3]{2(\sqrt{\beta_1 \beta_0} - \beta_0)^3 - 3\beta_0 \Delta(\sqrt{\beta_1 \beta_0} - \beta_0) - 3\Delta^2 \log(\varrho^{-1} + 1)}}{\Delta}, \quad (295)$$

*where*

$$\Delta = \beta_1 - 2\sqrt{\beta_1 \beta_0} + \beta_0. \quad (296)$$

### H.2 SOME OTHER INVERSE FUNCTIONS

**Gamma to sigma.** Additionally, we need to be able to extract the weighting terms from the time integration variable. For the ODE case we need the function $\sigma_\gamma(\gamma)$ which describes the map $\gamma \mapsto \sigma$. By the definition of $\gamma$ we have

$$\gamma = \frac{\alpha}{\sigma}, \quad (297)$$

$$\gamma \overset{(i)}{=} \frac{\sqrt{1 - \sigma^2}}{\sigma}, \quad (298)$$

$$\sigma\gamma = \sqrt{1 - \sigma^2}, \quad (299)$$

$$\sigma^2\gamma^2 = 1 - \sigma^2, \quad (300)$$

$$\sigma^2\gamma^2 = 1 - \sigma^2, \quad (301)$$

$$\gamma^2 = \sigma^{-2} - 1, \quad (302)$$

$$\gamma^2 + 1 = \sigma^{-2}, \quad (303)$$

$$\sigma^2 = \frac{1}{\gamma^2 + 1} \quad (304)$$

$$\sigma_\gamma(\gamma) = \frac{1}{\sqrt{\gamma^2 + 1}}, \quad (305)$$

where (i) hold by $\sigma^2 = 1 - \alpha^2$ for VP type diffusion SDEs.

**Rho to sigma over gamma.** Likewise, for the SDE case we need the function which maps $\varrho \mapsto \frac{\sigma}{\gamma}$. Recall that (note we drop the subscript $t$ for the derivation)

$$\varrho = \frac{\alpha^2}{\sigma^2}, \quad (306)$$

thus we have

$$\varrho \overset{(i)}{=} \frac{\alpha^2}{1 - \alpha^2}, \quad (307)$$

$$(1 - \alpha^2)\varrho = \alpha^2, \quad (308)$$

$$\alpha^{-2} - 1 = \varrho^{-1}, \quad (309)$$

$$\alpha^{-2} = \varrho^{-1} + 1, \quad (310)$$

$$\alpha = \frac{1}{\sqrt{\varrho^{-1} + 1}}, \quad (311)$$

where (i) hold by $\sigma^2 = 1 - \alpha^2$ for VP type diffusion SDEs. Then we can write

$$\frac{\sigma}{\gamma} = \frac{\sigma^2}{\alpha}, \tag{312}$$

$$= \frac{\sigma^2}{\alpha}\frac{\alpha}{\alpha}, \tag{313}$$

$$= \frac{\sigma^2}{\alpha^2}\alpha, \tag{314}$$

$$= \varrho^{-1}\alpha, \tag{315}$$

$$= \frac{1}{\rho\sqrt{\rho^{-1} + 1}}. \tag{316}$$

**Chi to alpha.** Lastly, for the noise prediction models we need the map $\chi \mapsto \alpha$ denoted $\alpha_\chi(\chi)$. By definition of $\chi$ we have

$$\chi = \frac{\sigma}{\alpha}, \tag{317}$$

$$\chi \overset{(i)}{=} \frac{\sqrt{1 - \alpha^2}}{\alpha}, \tag{318}$$

$$\alpha_\chi(\chi) \overset{(ii)}{=} \frac{1}{\sqrt{\chi^2 + 1}}, \tag{319}$$

where (i) hold by $\sigma^2 = 1 - \alpha^2$ for VP type diffusion SDEs and (ii) holds by the derivation for $\sigma_\gamma(\gamma)$ *mutatis mutandis*.

### H.3 BROWNIAN MOTION

We used the Brownian interval (Kidger et al., 2021) provided by the `torchsde` library. In general we would recommend the virtual Brownian tree from Jelinčič et al. (2024) over the Brownian interval, an implementation of this can be found in the `diffrax` library. However, as our code base made extensive used of prior projects developed in pytorch and `diffrax` is a jax library it made more sense to use `torchsde` for this project.

## I EXPERIMENTAL DETAILS

We provide additional details for the empirical studies conducted in Section 5. *N.B.*, for all experiments we used fixed random seeds between the different software components to ensure a fair comparision.

### I.1 UNCONDITIONAL IMAGE GENERATION

#### I.1.1 DIFFUSION MODEL

We make use of a pre-trained DDPM (Ho et al., 2020) model trained on the CelebA-HQ $256 \times 256$ dataset (Karras et al., 2018). The linear noise schedule from (Ho et al., 2020) is given as

$$\beta_i = \frac{\hat{\beta}_0}{T} + \frac{i - 1}{T(T - 1)}(\hat{\beta}_1 - \hat{\beta}_0). \tag{320}$$

We convert this into a continuous time representation via the details in Appendix H.1 following Song et al. (2021b). For this experiment we used $\hat{\beta}_0 = 0.0001$ and $\hat{\beta}_1 = 0.2$. To ensure numerical stability due to $\frac{1}{\sigma_t}$ terms we solve the probability flow ODE in reverse-time on the time interval $[\epsilon, 1]$ with $\epsilon = 0.0002$. This is a common choice to make in practice see Song et al. (2023).

#### I.1.2 METRICS

We use several metrics to assess the performance in unconditional image generation following Stein et al. (2023) by using a DINOv2 feature extractor (Oquab et al., 2023), all of which are calculated

using the 10k generated samples and 30k real samples from the CelebA-HQ dataset. Throughout this section we will let $\{x_i\}_{i=1}^n$ denote an empirical distribution drawn from our generated distribution $\mathbb{P}_\theta$ and let $\{\hat{x}_i\}_{i=1}^m$ denote an empirical distribution drawn from the data distribution $\mathbb{P}_{data}$.

**FD.** The *Fréchet distance* (FD) (Dowson & Landau, 1982) is measured using the sample mean and covariance of the real $\mathbb{P}_{data}$ and generated $\mathbb{P}_\theta$ distributions denoted

$$\text{FD}(\mathbb{P}_{data}\|\mathbb{P}_\theta) = \|\mu_{data} - \mu_\theta\|_2^2 + \text{Tr}\left(\Sigma_{data} + \Sigma_\theta - 2(\Sigma_{data}\Sigma_\theta)^{\frac{1}{2}}\right), \quad (321)$$

where $(\mu_\cdot, \Sigma_\cdot)$ denote the sample mean and covariances. This metric corresponds two the 2-Wasserstein distance between two multivariate Gaussians and is thus a valid metric between the first two moments. Heusel et al. (2017) popularized the use of this metric within the feature layer of an Inception-V3 network (Szegedy et al., 2016) to assess the fidelity of unconditional image generation, this metric is referred to as the *Fréchet inception distance* or FID. Recent works have challenged the use of the Inception-V3 network as the feature extractor (Stein et al., 2023; Jayasumana et al., 2024; Kynkäänniemi et al., 2023) showing that the Inception-V3 network is poorly suited for capturing a semantic view of images which correlates well to human judgment. In particular, Stein et al. (2023) shows that using DINOv2 (Oquab et al., 2023) for the feature extractor results in a metric which is significantly more aligned with human judgment.

**FD$_\infty$.** FD$_\infty$ proposed by Chong & Forsyth (2020) is a modification of FD which aims to remove the inherent bias induced by using a finite number of empirical samples. The samples is determined by evaluating FD over 15 regular intervals over the number of total samples and fitting a linear trend to the 15 data points to infer a trend for FD as the number of empirical samples, $N \to \infty$.

**Precision, recall, density and coverage.** The density metric (Naeem et al., 2020) is used as a proxy to measure sample fidelity and improves upon the earlier precision metric (Kynkäänniemi et al., 2019; Sajjadi et al., 2018). The metric is based upon nearest neighbours distance computed in a representation space and counts how many real-sample neighbourhood balls contain the generated sample. Likewise to quantify sample diversity we use the coverage metric (Naeem et al., 2020) which improves upon the earlier recall metric (Kynkäänniemi et al., 2019; Sajjadi et al., 2018). The density metric is given by

$$\text{density}(\mathbb{P}_{data}, \mathbb{P}_\theta) = \frac{1}{kn}\sum_{i=1}^n\sum_{j=1}^m 1_{B(\hat{x}_j, \delta^k(\hat{x}_j))}(x_i), \quad (322)$$

where $1_A(\cdot)$ denotes the indicator function for set $A$, $B(x, r)$ constructs a Euclidean ball centered at $x$ with radius $r$, and $\delta^k(\hat{x}_j)$ is the distance to the $k$-th nearest neighbour in $\{\hat{x}_i\}_{i=1}^m$, excluding itself. The precision metric is given by

$$\text{precision}(\mathbb{P}_{data}, \mathbb{P}_\theta) = \frac{1}{n}\sum_{i=1}^n 1_{\bigcup_{j=1}^m B(\hat{x}_j, \delta^k(\hat{x}_j))}(x_i). \quad (323)$$

Similarly, coverage is given by

$$\text{coverage}(\mathbb{P}_{data}, \mathbb{P}_\theta) = \frac{1}{m}\sum_{j=1}^m \max_{i=1,\ldots,n} 1_{B(\hat{x}_j, \delta^k(\hat{x}_j))}(x_i). \quad (324)$$

Likewise, the recall metric is given by

$$\text{recall}(\mathbb{P}_{data}, \mathbb{P}_\theta) = \frac{1}{m}\sum_{j=1}^m 1_{\bigcup_{i=1}^n B(x_i, \delta^k(x_i))}(\hat{x}_j). \quad (325)$$

We used $k = 5$ and 10k samples throughtout, as standard.

**On reporting.** When reporting on these metrics like in Table 1 we use **bold font** to denote the best performance with a 1% error range. More formally, suppose we have a series of $n$ data points $\{x_i\}_{i=1}^n$ that is totally ordered by some relation $R$. We say will denote a query point $x_i$ with **bold font** if the *range-normalized absolute percentage error* is less than $\epsilon > 0$, *i.e.*,

$$\frac{|\max_j x_j - x_i|}{\max_j x_j - \min_k x_k} < \epsilon. \quad (326)$$

In our experiments we report $\epsilon = 0.01$.

### I.1.3 HYPERPARAMETERS

We follow the suggestion of Wallace et al. (2023) and report results with EDICT using the hyper-parameter $p = 0.93$. For BDIA, the original paper recommends $\gamma = 1.0$ for unconditional image generation (Zhang et al., 2024, Section 6.1). However, we found $\gamma = 0.5$ to yield better performance, this corroborates with the findings of Wang et al. (2024).

## I.2 CONDITIONAL IMAGE GENERATION

### I.2.1 DIFFUSION MODEL

We make use of Stable Diffusion v1.5 (Rombach et al., 2022) a pre-trained *latent diffusion model* (LDM) model. We also use the scaled linear noise schedule given as

$$\beta_i = \left( \sqrt{\frac{\hat{\beta}_0}{T}} + \frac{i-1}{\sqrt{T}(T-1)} \left( \sqrt{\hat{\beta}_1} - \sqrt{\hat{\beta}_0} \right) \right)^2. \tag{327}$$

We convert this into a continuous time representation via the details in Appendix H.1 following Song et al. (2021b). For this experiment we used $\hat{\beta}_0 = 0.00085$ and $\hat{\beta}_1 = 0.012$. To ensure numerical stability due to $\frac{1}{\sigma_t}$ terms we solve the probability flow ODE in reverse-time on the time interval $[\epsilon, 1]$ with $\epsilon = 0.0002$. This is a common choice to make in practice see Song et al. (2023).

**Numerical schemes.** We set the last two steps of Rex schemes to be either Euler or Euler-Maruyama for better stability near time 0.

### I.2.2 METRICS

As mentioned in the main paper we use the CLIP Score (Hessel et al., 2021) PickScore (Kirstain et al., 2023), and Image Reward metrics (Xu et al., 2023) to asses the ability of the text-to-image conditional generation task. We calculate each by comparing the sampled image and the given text prompt used to produce the image. We then report the average over the 1000 samples.

**CLIP score.** The CLIP score measures the cosine similarity between the text and visual embeddings with pretrained CLIP model (Radford et al., 2021) denoted as

$$\text{CLIPScore}(\boldsymbol{x}, \boldsymbol{c}) = \max \left\{ \frac{\langle \mathcal{E}_I(\boldsymbol{x}), \mathcal{E}_C(\boldsymbol{c}) \rangle}{\|\mathcal{E}_I(\boldsymbol{x})\| \|\mathcal{E}_C(\boldsymbol{c})\|}, 0 \right\}, \tag{328}$$

where $\mathcal{E}_I : \mathbb{R}^d \to V$ is the image embedder and $\mathcal{E}_C : \mathbb{R}^{d'} \to V$ is the caption embedder; and where $\boldsymbol{x}$ is the query image and $\boldsymbol{c}$ is the query caption. Thus this metric aims to measure how well our generated images align with their prompt. In particular, we use the `ViT-L/14` backbone trained by OpenAI.

**PickScore.** Similar to CLIP score, PickScore finetunes a CLIP-H model on their proposed Pick-a-Pic dataset which purportedly aligns better with human preference over CLIP score.

**Image Reward.** Image Reward (Xu et al., 2023) is the newest of the three metrics and uses BLIP (Li et al., 2022) over CLIP as the backbone and finetunes the model using reward model training. The resulting metrics achieves state-of-the-art alignment with human preferences.

**On reporting.** When reporting on these metrics like in Table 2 we use **bold font** to denote the best performance with a 1% error range. In our experiments we report $\epsilon = 0.01$.

### I.2.3 HYPERPARAMETERS

We follow the suggestion of Wallace et al. (2023) and report results with EDICT using the hyperpa-rameter $p = 0.93$. For BDIA, the original paper recommends $\gamma = 0.5$ for text-to-image generation (Zhang et al., 2024, Section 6.1). We also ran BDIA with $\gamma = 0.96$ as suggested by Wang et al. (2024).

### I.3 INTERPOLATION

**Diffusion model.** We make use of a pre-trained DDPM (Ho et al., 2020) model trained on the CelebA-HQ $256 \times 256$ dataset (Karras et al., 2018). We used linear noise schedule from (Ho et al., 2020). We convert this into a continuous time representation via the details in Appendix H.1 following Song et al. (2021b). For this experiment we used $\hat{\beta}_0 = 0.0001$ and $\hat{\beta}_1 = 0.2$. For the face pairings we followed Blasingame & Liu (2024a;c) and used the FRLL (DeBruine & Jones, 2017) dataset.

Notably, we used the noise prediction parameterization rather than data prediction as we found that it performed better for editing. This is likely due to the singularity of the $\frac{1}{\sigma_t}$ terms as $t \to 0$. Within this parameterization we could use the time interval $[0, 1]$ instead of $[\epsilon, 1]$ like in previous experiments with data prediction models.

### I.4 HARDWARE

All experiments were run using a single NVIDIA H100 80 GB GPU.

### I.5 REPOSITORIES

In our empirical studies we made use of the following resources and repositories:

1. `google/ddpm-celebahq-256` (DDPM Model)

2. `stable-diffusion-v1-5/stable-diffusion-v1-5` (Stable Diffusion v1.5)

3. `zituitui/BELM` (Implementation of BELM, EDICT, and BDIA)

4. `google-research/torchsde` (Brownian Interval)

5. `layer6ai-labs/dgm-eval` (FD, FD$_\infty$, KD, Density, and Coverage metrics)

6. `torchmetrics` (CLIP score)

7. `zai-org/ImageReward` (Image Reward)

## J CODE

In this section we provide some example code for the core components of the model to help illustrate the core ideas.

```
Code J.1: Rex forward step

def rex_forward(model_func, scheduler, xt, xt_hat, timesteps, solver='euler', coupling=0.999,
↪   low_order_final_n_steps=0, bm=None, pred_type='data', sched_type='linear'):
    """
    Based on McCallum & Foster's reversible ODE solver and adapted for diffusion models.
    """

    # Choose underlying solver
    is_sde = (solver in SDE_SOLVERS)
    psi = SOLVER_DICT[solver]

    if not is_sde:
        _t_to_gamma, _gamma_to_t = _gen_time_funcs(sched_type=sched_type, pred_type=pred_type)
        t_to_gamma = _t_to_gamma
        gamma_to_t = _gamma_to_t
        gamma_to_sigma = _gamma_to_sigma if pred_type == 'data' else _chi_to_alpha
    else:
        _t_to_rho, _rho_to_t = _gen_time_funcs(sched_type=sched_type, rho=True, pred_type=pred_type)
        t_to_gamma = _t_to_rho
        gamma_to_t = _rho_to_t
        gamma_to_sigma = _rho_to_siggamma if pred_type == 'data' else _chi_to_alpha

    # create timesteps in gamma, alt gamma^2 = rho for SDEs
    gammas = t_to_gamma(scheduler, timesteps)

    # Push gamma reparam back to time t and convert noise pred to data pred
    if pred_type == 'data':
        wrap_model = lambda gamma, x: _convert_noise_to_data(scheduler, model_func,
        ↪   gamma_to_t(scheduler, gamma), x, sched_type=sched_type)
    else:
        p = 2 if is_sde else 1
        wrap_model = lambda gamma, x: p * model_func(gamma_to_t(scheduler, gamma), x)

    xt.to(torch.float32)
    xt_hat.to(torch.float32)

    for n in tqdm(range(len(gammas)-1)):
        gamma_n = gammas[n]
        gamma_n1 = gammas[n+1]
        h = gamma_n1 - gamma_n

        sigma_n = gamma_to_sigma(gamma_n)
        sigma_n1 = gamma_to_sigma(gamma_n1)

        if n < (len(gammas) - 1 - low_order_final_n_steps):
            if not is_sde:
                _psi = lambda t, x, h: psi(wrap_model, t, x, h)
            else:
                _psi = lambda t, x, h: psi(wrap_model, t, x, h, bm, pred_type=pred_type)
        else:
            if not is_sde:
                _psi = lambda t, x, h: euler(wrap_model, t, x, h)
            else:
                _psi = lambda t, x, h: euler_maruyama(wrap_model, t, x, h, bm, pred_type=pred_type)

        xt = (sigma_n1 / sigma_n) * (coupling * xt + (1-coupling) * xt_hat) + sigma_n1 * _psi(gamma_n,
        ↪   xt_hat, h)
        xt_hat = (sigma_n1 / sigma_n) * xt_hat - sigma_n1 * _psi(gamma_n1, xt, -h)

    return xt, xt_hat
```

### Code J.2: Rex backward step

```python
def rex_backward(model_func, scheduler, xt, xt_hat, timesteps, solver='euler', coupling=0.999,
↪   low_order_final_n_steps=0, bm=None, pred_type='data', sched_type='linear'):
    """
    Based on McCallum & Foster's reversible ODE solver and adapted for diffusion models.
    """

    # Choose underlying solver
    is_sde = (solver in SDE_SOLVERS)
    psi = SOLVER_DICT[solver]

    if not is_sde:
        _t_to_gamma, _gamma_to_t = _gen_time_funcs(sched_type=sched_type, pred_type=pred_type)
        t_to_gamma = _t_to_gamma
        gamma_to_t = _gamma_to_t
        gamma_to_sigma = _gamma_to_sigma if pred_type == 'data' else _chi_to_alpha
    else:
        _t_to_rho, _rho_to_t = _gen_time_funcs(sched_type=sched_type, rho=True, pred_type=pred_type)
        t_to_gamma = _t_to_rho
        gamma_to_t = _rho_to_t
        gamma_to_sigma = _rho_to_siggamma if pred_type == 'data' else _chi_to_alpha

    # create timesteps in gamma, alt gamma^2 = rho for SDEs
    gammas = t_to_gamma(scheduler, timesteps)

    # Push gamma reparam back to time t and convert noise pred to data pred
    if pred_type == 'data':
        wrap_model = lambda gamma, x: _convert_noise_to_data(scheduler, model_func,
        ↪   gamma_to_t(scheduler, gamma), x, sched_type=sched_type)
    else:
        p = 2 if is_sde else 1
        wrap_model = lambda gamma, x: p * model_func(gamma_to_t(scheduler, gamma), x)

    xt.to(torch.float32)
    xt_hat.to(torch.float32)

    coupling_inv = 1. / coupling

    for n in tqdm(range(len(gammas) - 2, -1, -1)):
        gamma_n = gammas[n]
        gamma_n1 = gammas[n+1]
        h = gamma_n1 - gamma_n

        sigma_n = gamma_to_sigma(gamma_n)
        sigma_n1 = gamma_to_sigma(gamma_n1)

        if n < (len(gammas) - 1 - low_order_final_n_steps):
            if not is_sde:
                _psi = lambda t, x, h: psi(wrap_model, t, x, h)
            else:
                _psi = lambda t, x, h: psi(wrap_model, t, x, h, bm, pred_type=pred_type)
        else:
            if not is_sde:
                _psi = lambda t, x, h: euler(wrap_model, t, x, h)
            else:
                _psi = lambda t, x, h: euler_maruyama(wrap_model, t, x, h, bm, pred_type=pred_type)

        xt_hat = (sigma_n / sigma_n1) * xt_hat + sigma_n * _psi(gamma_n1, xt, -h)
        xt = (sigma_n / sigma_n1) * (coupling_inv * xt) + (1 - coupling_inv) * xt_hat - sigma_n *
        ↪   coupling_inv * _psi(gamma_n, xt_hat, h)

    return xt, xt_hat
```

In Code J.3 we provide an implementation of the ShARK method. The official implementation can be found at https://github.com/patrick-kidger/diffrax/blob/main/diffrax/_solver/shark.py.

Code J.3: ShARK

```python
def ShARK(model, time_var, x, h, bm, pred_type='data'):
    t_to_w = _rho_to_siggamma if pred_type == 'data' else _chi_to_alpha

    x_sg = x / t_to_w(time_var)

    if pred_type == 'data':
        a, b = time_var, time_var + h
    else:
        a, b = time_var.pow(2), (time_var + h).pow(2)

    if h < 0:
        a, b = b, a

    h_corr = h if pred_type == 'data' else (time_var + h).pow(2) - time_var.pow(2)

    W, U = bm(a, b, return_U=True)
    W, U = W.to(x.device), U.to(x.device)

    if h < 0:
        H = U / (-h_corr) - 0.5 * W
        W = -W
    else:
        H = U / (-h_corr) - 0.5 * W

    Z1 = x_sg + H

    f1 = model(time_var, t_to_w(time_var) * Z1)

    Z2 = x_sg + h * (5/6) * f1 + (5/6) * W + H
    f2 = model(time_var + 5/6 * h, t_to_w(time_var + 5/6 * h) * Z2)

    return h * (0.4 * f1 + 0.6 * f2) + W
```

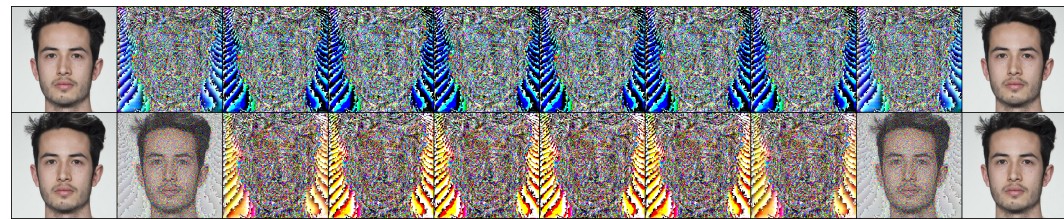

Figure 7: Inversion followed by sampling with Rex (Euler) 5 steps, $\zeta = 0.999$. Data prediction. Top row tracks $\boldsymbol{x}_n$, bottom row $\hat{\boldsymbol{x}}_n$.

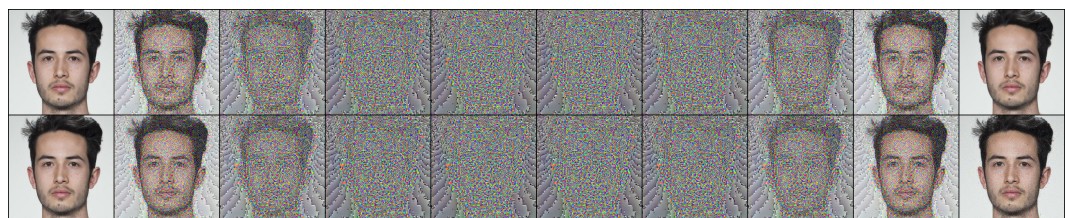

Figure 8: Inversion followed by sampling with Rex (Euler) 5 steps, $\zeta = 0.999$. Noise prediction. Top row tracks $\boldsymbol{x}_n$, bottom row $\hat{\boldsymbol{x}}_n$.

## K  VISUALIZATION OF INVERSION AND THE LATENT SPACE

We conduct a further qualitative study of the latent space produced by inversion and the impact various design parameters play. First in Figure 7 we show the process of inverting and then reconstructing a real sample. Notice that while the data prediction formulation worked great in sampling and still possesses the correct reconstruction, *i.e.*, it is still reversible, the latent space is all messed up. The variance of $(\boldsymbol{x}_n, \hat{\boldsymbol{x}}_n)$ tends to about $10^7$, many orders of magnitude too large! We did observe that raising $\zeta = 1 - 10^{-9}$ did help reduce this, but it was still relatively unstable. *N.B.*, these trends hold in a large number of discretization steps (we tested up to 250); however, for visualization purposes we chose fewer steps.

Conversely, the noise prediction formulation is much more stable, see Figure 8. The variance of $(\boldsymbol{x}_n, \hat{\boldsymbol{x}}_n)$ is on the right order of magnitude this time, however, there are strange artefacting and it is clear the latent variables are not normally distributed.

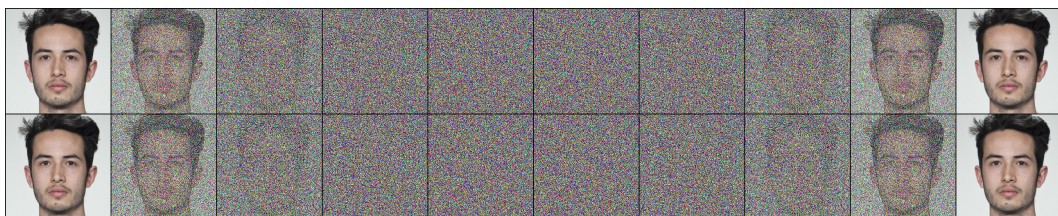

Figure 9: FAILURE CASE! Inversion followed by sampling with Rex (ShARK) 5 steps, $\zeta = 0.999$. Data prediction. Top row tracks $\boldsymbol{x}_n$, bottom row $\hat{\boldsymbol{x}}_n$.

Figure 10: Inversion followed by sampling with Rex (ShARK) 5 steps, $\zeta = 0.999$. Noise prediction. Top row tracks $\boldsymbol{x}_n$, bottom row $\hat{\boldsymbol{x}}_n$.

Moving to the SDE case with ShARK in Figure 9, we see that the data prediction formulation is so unstable in forward-time that we ran into overflow errors and can no longer achieve algebraic reversibility. However, the noise parameterization with ShARK, see Figure 10, works very well with the latent variables appearing to be close to normally distributed.

## L ADDITIONAL RESULTS

### L.1 UNCONDITIONAL IMAGE GENERATION

We present some additional ablations on the underlying solver for Rex in Table 4.

Table 4: Quantitative comparison of different underlying schemes $\boldsymbol{\Phi}$ used in Rex in terms of FID ($\downarrow$) for unconditional image generation with a pre-trained DDPM model on CelebA-HQ ($256 \times 256$).

| Steps | Euler | Midpoint | RK4 | Euler-Maruyama | ShARK |
|---|---|---|---|---|---|
| 10 | 36.65 | x | 31.00 | 40.79 | 59.89 |
| 20 | 24.63 | 23.36 | 23.49 | 27.80 | 32.18 |
| 50 | 21.45 | 21.45 | 21.35 | 19.77 | 21.93 |

### L.2 CONDITIONAL IMAGE GENERATION

We present some uncrated samples using Rex with various underlying solvers and discretization steps.

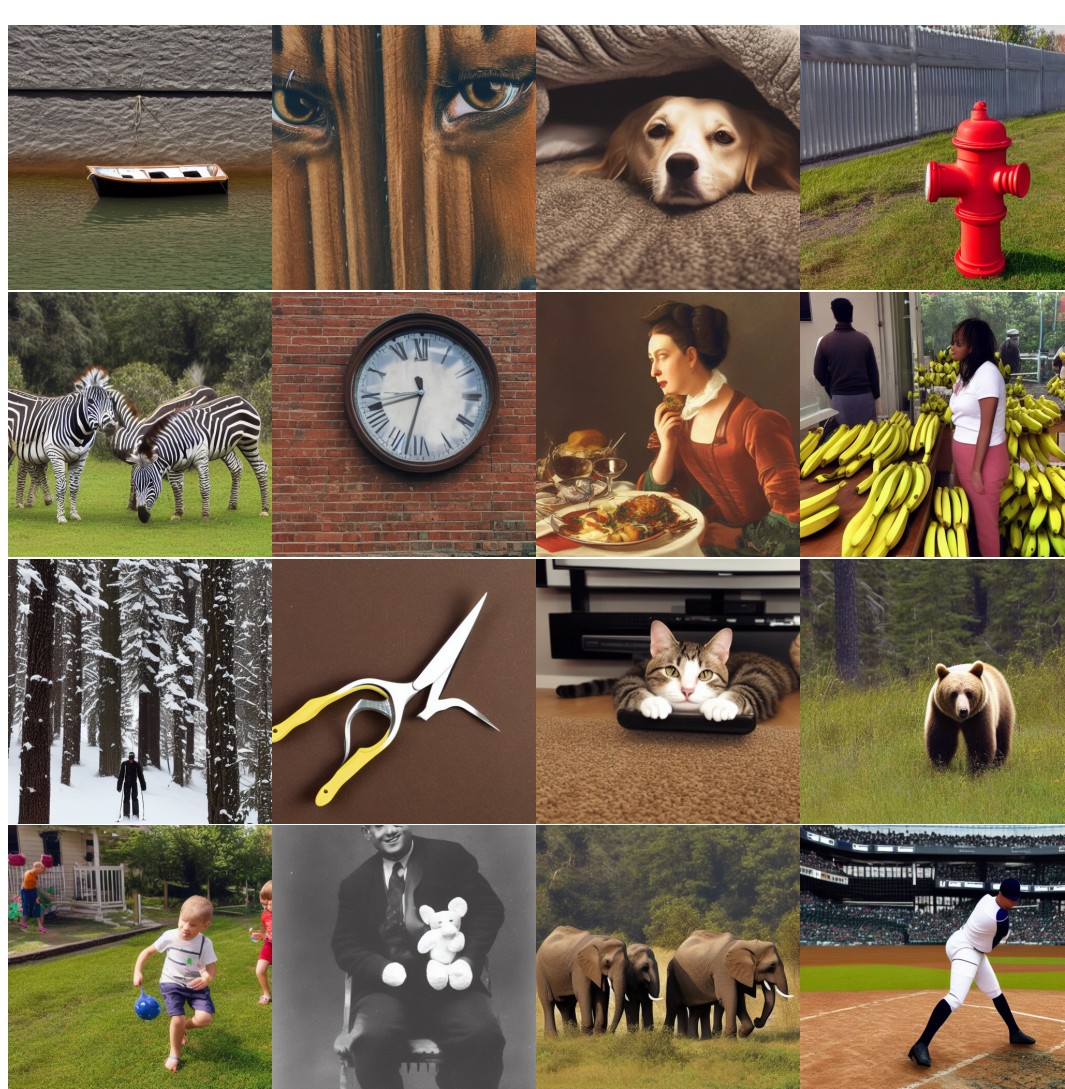

Figure 11: Uncurated samples created using Rex (RK4) and Stable Diffusion v1.5 ($512 \times 512$) and 10 discretization steps.

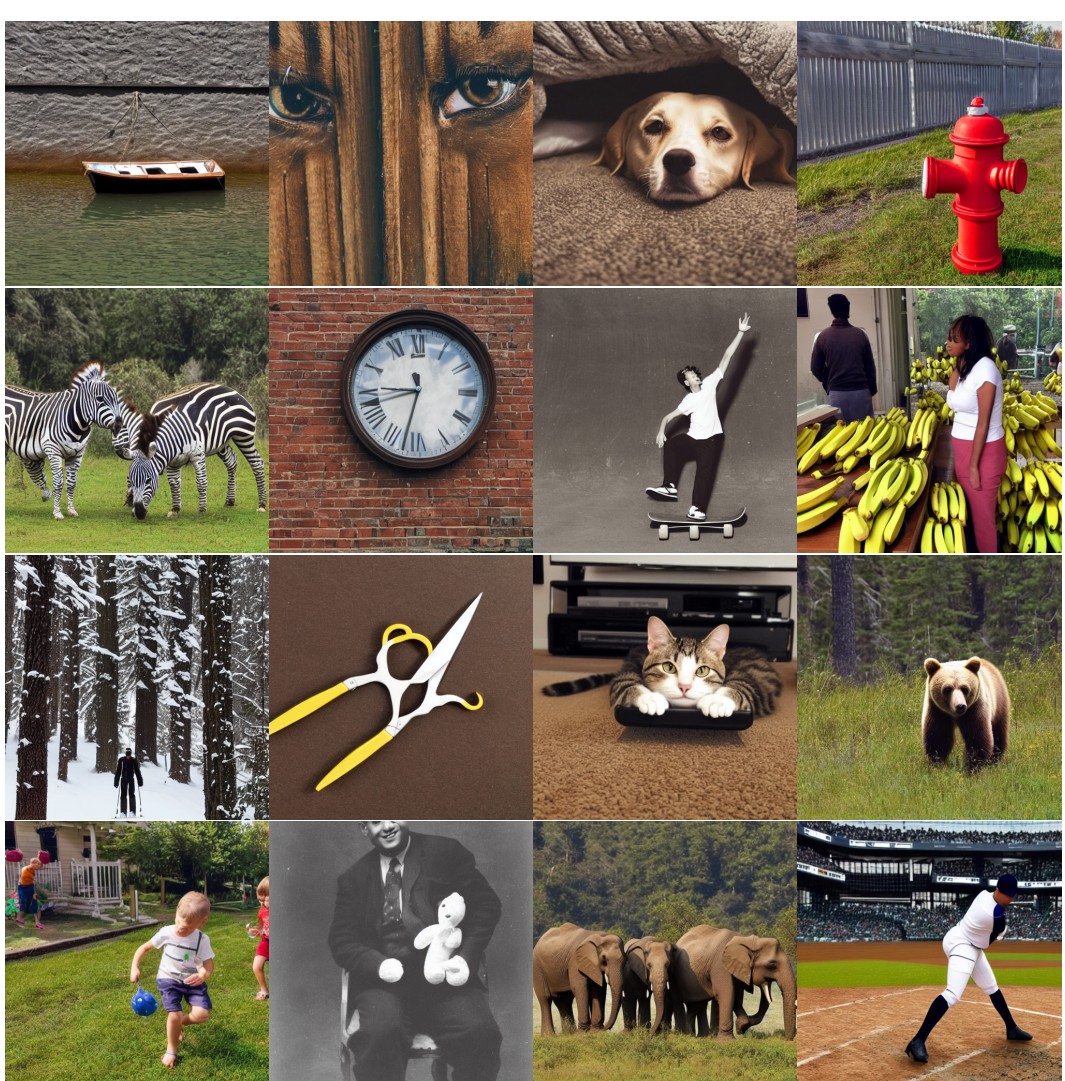

Figure 12: Uncurated samples created using Rex (RK4) and Stable Diffusion v1.5 ($512 \times 512$) and 50 discretization steps.

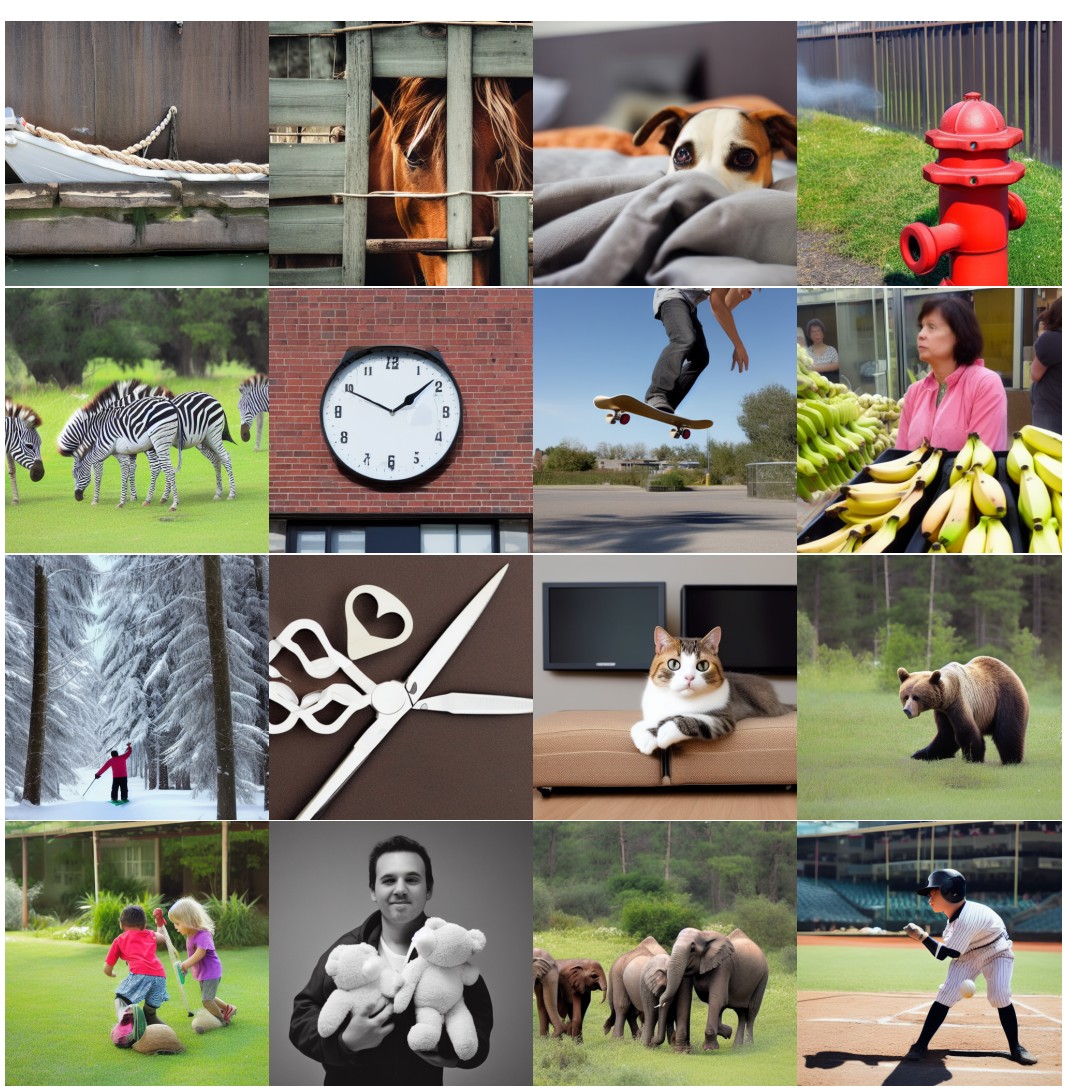

Figure 13: Uncurated samples created using Rex (ShARK) and Stable Diffusion v1.5 ($512 \times 512$) and 10 discretization steps.

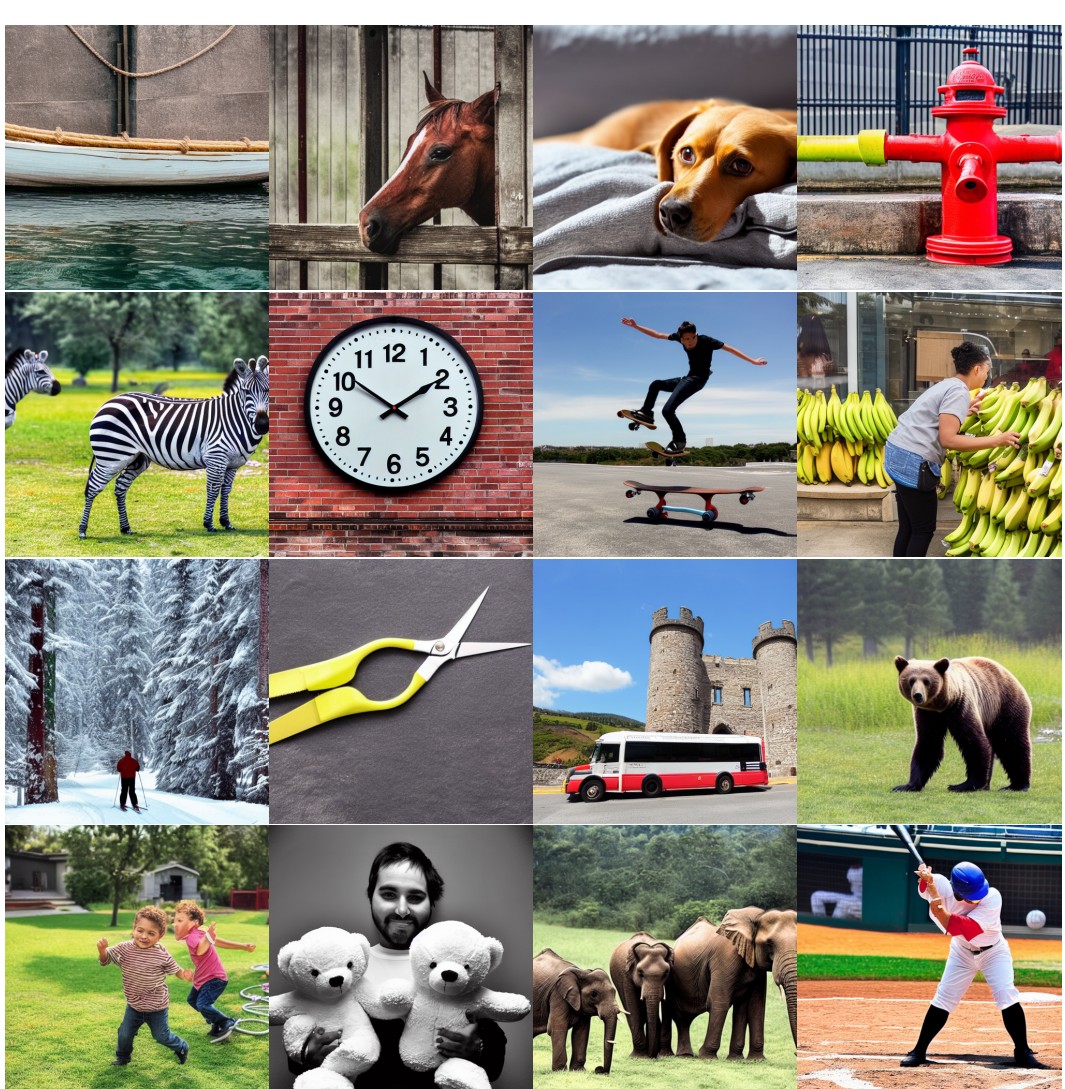

Figure 14: Uncurated samples created using Rex (ShARK) and Stable Diffusion v1.5 ($512 \times 512$) and 50 discretization steps.

