# OpenReview forum: "Rex: Reversible Solvers for Diffusion Models"
_ICLR.cc/2026/Conference — Submitted to ICLR 2026_

### Official Review · Reviewer_qLBS · 2025-10-26

**Soundness:** 2
**Presentation:** 3
**Contribution:** 3
**Rating:** 4
**Confidence:** 3

**Summary:**

This paper addresses the important and challenging problem of creating algebraically reversible numerical solvers for diffusion models, which is crucial for tasks like data inversion, editing, and interpolation. The authors propose "Rex," a new family of solvers based on a novel combination of the McCallum-Foster (2024) reversible solver framework and exponential integrators (Lawson methods) tailored for diffusion models.

**Strengths:**

1. It proposes Rex-ODE, a reversible ODE solver that, by applying the McCallum-Foster method in a reparameterized space, inherits its non-trivial linear stability and high-order convergence properties —a clear advantage over prior work.

2. It introduces (to my knowledge) the first practical, algebraically reversible SDE solver (Rex-SDE) that does not require storing the entire $\mathcal{O}(N)$ Brownian motion path in memory. It cleverly achieves this using splittable PRNGs for noise reconstruction.

3. It provides a deep and valuable theoretical analysis (primarily in Appendix A) demonstrating that recent competing reversible solvers, namely BDIA and O-BELM, are fundamentally variants of the leapfrog/midpoint method and are thus nowhere linearly stable.

**Weaknesses:**

1. The paper is deeply contradictory. The main derivation in Section 3.1 is for the "data prediction" parameterization. However, Appendix I (Figs 6, 8) explicitly shows this is pathologically unstable for inversion, with latent variances exploding to $\approx 10^7$. Appendix I and G.3 state that the "noise prediction" parameterization is stable and was used for key experiments. This makes it unclear what method was actually used for the main results in Tables 1 & 2 and invalidates the focus of the main paper's derivation.

2. A key selling point is the ability to achieve "arbitrarily high order of convergence". Yet, the empirical results in Tables 1, 2, and 4 consistently show that the high-order Rex (RK4) performs worse than its low-order counterparts (Rex-Euler, Rex-EM). The paper completely fails to discuss or analyze this significant negative finding.

3. For the standard unconditional sampling task (Table 1, FID), the Rex-ODE solvers are not SOTA. They are outperformed by the O-BELM baseline at 10 and 20 steps —the very baseline this paper proves is unstable. This suggests that for pure sampling (not inversion), the theoretical stability of Rex does not necessarily translate to superior FID.

4. What is the computational overhead (i.e., extra latency) of this "re-compute" strategy during the backward pass (inversion or gradient back-propagation) compared to the naive approach of simply storing the $\mathcal{O}(N)$ path? Is there a significant practical trade-off between memory and speed?

5. If an "unstable" method can achieve superior FID in a pure sampling task, why should the community adopt Rex-ODE for this purpose? Does this imply that the stability advantage of Rex is primarily relevant for inversion-dependent tasks (as asked in Q3) and offers no clear benefit for standard, forward-pass generation?

**Questions:**

See weakness.

---

> ### Author Response · Authors · 2025-11-21
> **Part 1**
>
> We thank reviewer qLBS for their time and helpful feedback. We hope that the points below address all the reviewer's concerns.
>
> ## Weaknesses / Questions
> 1. We thank the reviewer for pointing out our over emphasis on the *data prediction* formulation. Rex is a family of solvers covering four scenarios:
>
>      i. Data / ODE
>
>      ii. Data / SDE
>
>      iii. Noise / ODE
>
>      iv. Noise / SDE
>
> After reading your feedback and rereading our submission we realized we clearly favoured i and ii over the others. We have now adjusted our paper to reflect that we study both data and noise. In particular, we have made the following changes
>
>      a. Updated Figure 2 to reflect both data/noise prediction scenarios,
>      b. Rewrote all of Section 3 and in particular Section 3.1 to reflect that we consider both noise/data scenarios.
>      c. We rewrote Proposition 3.3 to be generally stated for both data and noise prediction scenarios.
>
> Yes one of our findings in this work is that depending on the type of reversibility we want certain scenarios are better suited due to the stability of the scheme. We did find that for reversible tasks starting from the image domain $X_0 \sim q(X)$ that the noise prediction reparameterization was better suited (ie, forward is $X_0 \to X_T$ and backward is $X_T \to X_0$) Conversely, if we wanted reversibility starting from the latent space $X_T \sim p(X)$ (e.g., for gradient guidance) the data parameterization would be better suited.
>
> 2. At the suggestion of multiple reviewers we reran these experiments with a more comprehensive set of metrics. In both our updates Tables 1 and 2 we see that Rex does well across the board, however, Rex (RK4) does not perform as well as our SDE solvers. We don't believe this detracts from our contributions as a rather larger part of our work is getting this scheme to work with diffusion SDEs, all other methods for exact inversion with diffusion models only focused on ODEs, and the only reversible scheme for SDEs is Kidger's reversible Heun which has poor stability in the ODE version. We view the SDE contribution as the larger half over the ODE contribution, we more or less included it because we get it for free with all the maths necessary to get the reversible SDE version.
>
> 3. Upon rerunning and more thoroughly exploring the unconditional image generation experiments we find that the ODE version of Rex performs better than we had initially thought. In particular, we see that Rex (ODE) does well in the precision, recall, density, and coverage metrics. Moreover, we did not optimize the coupling parameter $\zeta = 0.999$ for sampling, like prior works. As mentioned in [2, Section 2.5] we get better stability as $\zeta \to 0$, but worse reversibility.
>
> 4. Are you referring to storing $\\{\boldsymbol x_n\\}\_{n=1}^N$ or $\\{\boldsymbol W_n\\}\_{n=1}^N$?. For the former it would depend on the task. As others in the neural differential equation community have pointed out the preferred way to do backprop through a neural differential equation is DTO / DTO with Checkpointing > OTD/DTO with reversible solvers > OTD with non-reversible solvers (avoid at all costs!), see https://docs.kidger.site/diffrax/api/adjoints/. So the choice depends on what you want do with the trajectory, the size of the neural network, and number of discretization steps.
>
> 5. As we mentioned earlier we have rerun our unconditional generation experiments which shows the utility of Rex. But to further answer your question we have added Section 4.2 and in particular Theorem 4.3 which shows that Rex is actually the reversible version of many pre-existing and popular solvers for diffusion models. With regards to stability as proved in [2, Theorem 2.3] the stability region of the reversible version is a *strict a subset* of the stability region of the original non-reversible scheme $\boldsymbol \Psi$, however, unlike prior reversible solvers we have a region of stability with non-zero measure. Or in other words for stability we have: $EDICT = BDIA = BELM = Reversible Heun = Asynchronous Leapfrog \subsetneq Rex \subsetneq \boldsymbol \Psi$. So yes for non-inversion tasks it is not recommended to use Rex as the reversible version offers no benefits if you don't care about the reversibility. We hope to run some additional experiments on our non-reversible ShARK scheme if we have time, but currently we have not added them to the pdf.
>
> We hope this answers the reviewer's questions and we would be happy to provide more details or answer future questions.
>
> ### References
> [1] Cheng Lu, Yuhao Zhou, Fan Bao, Jianfei Chen, Chongxuan Li, and Jun Zhu. Dpm-solver++: Fast
> solver for guided sampling of diffusion probabilistic models. arXiv preprint arXiv:2211.01095,
> 2022a
>
> [2] Sam McCallum and James Foster. Efficient, accurate and stable gradients for neural odes. arXiv
> preprint arXiv:2410.11648, 2024

---

### Official Review · Reviewer_GcU8 · 2025-10-27

**Soundness:** 3
**Presentation:** 3
**Contribution:** 2
**Rating:** 4
**Confidence:** 3

**Summary:**

This paper proposes a family of reversible SDE solvers named "Rex" based on the McCallum-Foster method, which can be reduced to reversible ODE  solvers by proper parameter-setup. To facilitate the derivation of Rex, the "change-of-variables" and Lawson method are utilised for reformulation of ODE and ODE solver. Experimental results on unconditional and conditional sampling are provided in the paper.

**Strengths:**

The paper proposes, for the first time, a family of reversible SDE solvers based on the McCallum-Foster method. Convergence analysis is conducted for Rex, showing the kth order convergence behavior. To facilitate exact inversion, random seeds for generating the Gaussian noises in the process of clean image to latent noises are stored in the memory. In the process from latent noise to clean image, the stored random seeds are utilized to recomputed the same Gaussian noises.

**Weaknesses:**

(1) The main weakness is that the proposed Rex performs worse than O-BELM in Table 1. The authors show in Theorem 4.1 that Rex has high-order convergence behavior. However, the experiments do not show its advantages.  If so, why would other researchers use Rex for sampling?

(2) The experiments in the main paper only considers sampling and interpolation. I highly suggest the authors to evaluate Rex for round-trip image editing and compare with existing methods such as EDICT, BDIA, and O-BELM.  Note that reversible solvers are primarily used for image editing.  The authors state that both BDIA and O-BELM has poor stability.  Without the experiment on image editing, it is not clear at all if the proposed method provides any new benefits in terms of stability.

(3) In the original BDIA paper, the authors also perform experiments over Stable Diffusion v1.5, and show that BDIA performs better than DDIM. Why in Table, does DDIM perform better than BDIA? I highly suspect if the optimal hyper-parameters were chosen for BDIA in this paper.

**Questions:**

(1) The authors states that "Following Wang et al. (2024) we choose the optimal hyperparameters for BDIA, EDICT, and BELM". The authors should state what the optimal hyper-parameters are for those methods.

(2) Right before (2), I would think that alpha_t is monotonically decreasing over t and sigma_t is monotonically increasing over t. Correct me if I am wrong.

---

> ### Author Response · Authors · 2025-11-21
>
> We thank reviewer GcU8 for their time and helpful feedback. We hope that the points below address all the reviewer's concerns.
>
> ## Weaknesses
> 1. We realized after comments from several reviewers that we needed to do a more thorough job of exploring that experiment. We reran BDIA with the recommended parameters from their paper [Section 6.1, 1] ($\gamma = 1.0$) in additional to reporting a more comprehensive suite of metrics (FD, FD$_\infty$, precision, recall, density, and coverage) which are all reported in our updated Table 1. We also more explicitly draw out the connections between Rex and preexisting solvers for diffusion models, *i.e.*, Rex is the *reversible* version of well-known solvers, (*cf.* Theorem 4.3).
>
> 2. We plan on running more experiments along this direction, however, they are not yet done. We will update the reviewer as those experiments finish.
>
> 3. We thank the reviewer for their careful observation. There was indeed a discrepancy between the optimal parameters reported in the BELM paper [2] and the ones reported in the BDIA paper [1]. We reran both unconditional generation with the BDIA recommended $\gamma = 1.0$ and text-to-image generation with $\gamma = 0.5$.  The unconditional run with $\gamma = 1.0$ performed noticeably worse than our original run. The updated text-to-image run with $\gamma = 0.5$ performed a little better in the $< 10$ steps regime and a little bit worse in the $> 10$ steps regime, but the performance differences weren't too large. We provide two possible justifications for why BDIA performed worse relative to DDIM in Table 2:
>
>    a. In our experiments for text-to-image generation we measure prompt alignment and not FID,
>
>    b. The poor performance of BDIA wrt DDIM seems to align with [Table 4, 2]
>
>
> ## Questions
> 1. We agree that these should be explicity stated. We have now explicitly stated the hyperparameters in Appendices I.1.3 and I.2.3
> 2. Yes, you are correct, thank you for catching that typo. We have fixed it in our revised pdf.
>
> We hope this answers the reviewer's questions and we would be happy to provide more details or answer future questions.
>
> ### References
> [1] Guoqiang Zhang, J. P. Lewis, and W. Bastiaan Kleijn. Exact diffusion inversion via bidirectional
> integration approximation. In Computer Vision – ECCV 2024: 18th European Conference, Milan,
> Italy, September 29–October 4, 2024, Proceedings, Part LVII, pp. 19–36, Berlin, Heidelberg,
> 2024. Springer-Verlag. ISBN 978-3-031-72997-3. doi: 10.1007/978-3-031-72998-0_2. URL
> https://doi.org/10.1007/978-3-031-72998-0_2.
>
> [2] Fangyikang Wang, Hubery Yin, Yue-Jiang Dong, Huminhao Zhu, Chao Zhang, Hanbin Zhao, Hui
> Qian, and Chen Li. BELM: Bidirectional explicit linear multi-step sampler for exact inversion
> in diffusion models. In The Thirty-eighth Annual Conference on Neural Information Processing
> Systems, 2024. URL https://openreview.net/forum?id=ccQ4fmwLDb.

---

### Official Review · Reviewer_5BAq · 2025-11-01

**Soundness:** 4
**Presentation:** 3
**Contribution:** 3
**Rating:** 8
**Confidence:** 2

**Summary:**

This paper introduces REX, a family of algebraically reversible numerical solvers for diffusion models that enable exact inversion, mapping samples from the data distribution back to the prior distribution without reconstruction errors. The key innovation is applying Lawson methods combined with the McCallum-Foster reversible scheme to construct solvers that work for both the probability flow ODE and reverse-time SDE formulations of diffusion models.

**Strengths:**

The paper makes significant theoretical contributions by constructing the first known (to the best of my knowledge) method for exact SDE inversion without storing complete Brownian motion trajectories, which is a non-trivial achievement.

**Weaknesses:**

The only issue I see is the lack of concrete motivation for why exact inversion is important: the authors claim it is "invaluable for many downstream applications" but provide no citations or specific examples of these applications in the introduction or related work. This makes it difficult to assess the practical impact of the work beyond being a theoretically interesting problem.

**Questions:**

Can you provide specific examples with citations of downstream applications where exact inversion is crucial? What fails when using approximate inversion methods, and how does REX specifically enable these applications?

---

> ### Author Response · Authors · 2025-11-21
>
> We thank reviewer 5BAq for their helpful feedback and insightful questions.
>
> > Can you provide specific examples with citations of downstream applications where exact inversion is crucial? What fails when using approximate inversion methods, and how does REX specifically enable these applications?
>
> An important application for these methods is calculating the gradients through a neural ODE/CDE/SDE/RDE solve [1-3]. When performing backpropagation through the solve it is important to accurately **reconstruct** the solution trajectory to ensure we get accurate gradient information. This is especially relevant because the numerical stability the backward pass is often disastrously bad. For a good example we refer to [Example 5.6, 1].
>
> Now outside of training neural differential equations, there is interest in backproping through flow/diffusion models for **guided generation** with examples in molecule generation, adversarial attacks, and more. In some scenarios it may be possible to use standard *discretize-then-optimize* (DTO), *i.e.*, standard backprop. An example of this in practice is [4]. However, as the models get larger or number of discretization steps increases (some AI4science applications requiring many more integration steps than image generation) this strategy gets prohibitively intractable due to memory constraints. Others [5,6] have looked at using the adjoint method due to its $\mathcal O(1)$ memory cost; however, this has it's drawbacks of poor numerical stability as discussed in [1]. Some works circumvent this by storing the solution trajectory in memory [6, 7] but this again runs into memory issues with larger models or more discretization steps.
>
> We hope this answers the reviewer's questions and we would be happy to provide more details or answer future questions.
>
>
> ### References
>
> [1] Patrick Kidger. On Neural Differential Equations. Ph.d. thesis, Oxford University, 2022. Available at
> https://arxiv.org/abs/2202.02435.
>
> [2] Patrick Kidger, James Foster, Xuechen Chen Li, and Terry Lyons. Efficient and accurate gradients
> for neural sdes. Advances in Neural Information Processing Systems, 34:18747–18761, 2021.
>
> [3] Sam McCallum and James Foster. Efficient, accurate and stable gradients for neural odes. arXiv
> preprint arXiv:2410.11648, 2024.
>
> [4] Ben-Hamu, H., Puny, O., Gat, I., Karrer, B., Singer, U., and Lipman, Y. (2024). “D-Flow: Differ-
> entiating through Flows for Controlled Generation”. In: Forty-first International Conference on
> Machine Learning. URL: https://openreview.net/forum?id=SE20BFqj6J
>
> [5] Pan, J., Liew, J. H., Tan, V., Feng, J., and Yan, H. (2024). “AdjointDPM: Adjoint Sensitivity Method
> for Gradient Backpropagation of Diffusion Probabilistic Models”. In: The Twelfth International
> Conference on Learning Representations. URL: https://openreview.net/forum?id=y33lDRBgWI
>
> [6] Zander W. Blasingame and Chen Liu. Adjointdeis: Efficient gradients for diffusion models. In
> A. Globerson, L. Mackey, D. Belgrave, A. Fan, U. Paquet, J. Tomczak, and C. Zhang (eds.), Ad-
> vances in Neural Information Processing Systems, volume 37, pp. 2449–2483. Curran Associates,
> Inc., 2024a. URL https://proceedings.neurips.cc/paper_files/paper/2024/file/04badd3b048315c8c3a0ca17eff723d7-Paper-Conference.pdf.
>
> [7] Domingo-Enrich, C., Drozdzal, M., Karrer, B., and Chen, R. T. Q. (2025). “Adjoint Matching: Fine-
> tuning Flow and Diffusion Generative Models with Memoryless Stochastic Optimal Control”.
> In: The Thirteenth International Conference on Learning Representations. URL: https://openreview.net/forum?id=xQBRrtQM8u

---

### Official Review · Reviewer_MTWv · 2025-11-09

**Soundness:** 3
**Presentation:** 2
**Contribution:** 2
**Rating:** 4
**Confidence:** 3

**Summary:**

This paper introduce REX, a unified family of algebraically-reversible ODE/SDE solvers for diffusion models. By applying Lawson methods to construct exponential Runge-Kutta methods, a family of reversible solvers has been developed to meet the needs of diffusion models. Empirical results on image generation and interpolation show the effectiveness of the proposed method.

**Strengths:**

1. This paper extends the McCallum-Foster framework to stochastic diffusions using exponential integrators and time reparameterization.
2. Rex provides a reversible solution for stochastic differential equations.
3. The Appendix is quite comprehensive, containing rich theoretical validations.

**Weaknesses:**

1. The current submission spends too much content to preliminary discussions (including Sections 3.1 and 3.2). The exact introduction of the REX solver does not begin until page 6, which restricts the space available for experiments that could demonstrate the superiority of the proposed methods.
2. The current submission lacks experiments on image editing and reconstruction, which are commonly included in related literature.

**Questions:**

1. What are the advantages of REX compared to other related solvers? It is mentioned that one of the advantages of REX is its ability to perform exact inversion for diffusion SDEs without the need to store the entire trajectory. What are the exact costs of storing the entire trajectory?

2. As mentioned above, existing inversion works have made significant progress in applications such as image editing[1,2,3]. Would combining them with REX reduce the loss in inversion?

3. It would be better to provide results on more cutting-edge text-to-image models, e.g., SDXL, SD3 and Flux.

References

[1]Hertz, Amir, et al. "Prompt-to-Prompt Image Editing with Cross-Attention Control." *ICLR*. 2023.

[2]Tim Brooks, Aleksander Holynski, and Alexei A Efros. Instructpix2pix: Learning to follow image editing instructions. CVPR 2023.

[3]Narek Tumanyan, Michal Geyer, Shai Bagon, and Tali Dekel. Plug-and-play diffusion features for text-driven image-to-image translation.CVPR 2023.

---

> ### Author Response · Authors · 2025-11-22
>
> We thank reviewer MTWv for their time and helpful feedback. We hope that the points below address all the reviewer's concerns.
>
> ## Weaknesses
> 1. We thank the reviewer for pointing this out. We have now rewritten Section 2 & 3 to be more concise and introduce Rex earlier in page 5. We shortened some of our discussion on the probability flow ODE and SRK schemes to make room for this. We think the flow of the paper is better now with these changes.
>
> 2. We are currently running more experiments along this direction and will update the reviewer when they are done.
>
> ## Questions
> 1. The primary advantages of Rex are:
>
>    i) We create the **first** reversible solver for diffusion SDEs
>
>    ii) we can construct a $k$-th order ODE solver, now in practice for diffusion models $k$ doesn't necessarily need to be very high, but it's a cool theoretical result and could have more applications in ai4science applications where higher order schemes like Dormand-Prince 5 are used. *NB,* Rex also works with adaptive step-size solvers!
>
>    iii) For the ODE solvers we have a non-zero region of stability!
>
>    iv) Rex SDE uses a high-order SRK scheme based on the Foster-Reis-Strange scheme which uses Levy areas to approximate the higher-order terms. Importantly $\boldsymbol \Psi$ has *strong order* converge guarantees prior to earlier work which had *weak order* convergence guarantees. Constructing higher-order SDE schemes is tricky because we need to decompose the iterated Ito-Taylor expansions which preserve the Markov property. Prior work [1] found a clever scheme for this but could only obtain weak order convergence as result. We discuss this more in our newly added **Appendix A.4.1**.
>
>     v) Rex is actually the *reversible* version of many popular solvers for diffusion models. We discuss this in our newly added **Section 4.2** and specifically **Theorem 4.3**.
>
> The cost of storing the entire realization of the Brownian motion in memory would be $\mathcal O(nd)$ for diffusion models where $n$ is the number of discretization steps and $d$ is the dimensionality.
>
> 2. Yes, combining Rex would remove the inversion error entirely, however, once we perform edits this where the stability of the method would come into play.
>
> 3. As a part of our updated experiments we would like to include these. We will provide an update once they are done.
>
>
> We hope this answers the reviewer's questions and we would be happy to provide more details or answer future questions.
>
> ### References
> [1] Martin Gonzalez, Nelson Fernandez Pinto, Thuy Tran, Hatem Hajri, Nader Masmoudi, et al. Seeds:
> Exponential sde solvers for fast high-quality sampling from diffusion models. Advances in Neural
> Information Processing Systems, 36, 2024.

---

### Author Response · Authors · 2025-12-04
**Author Final Remarks**

We thank all the reviewers for their time and constructive feedback. We are glad that the reviewers appreciate our "significant theoretical contributions" **reviewer 5BAq** which has "non-trivial linear stability and high-order convergence properties" **reviewer qLBS**. We are also glad to find that the reviewers appreciate our detailed "convergence analysis... conducted for Rex, showing the kth order convergence behavior" **reviewer GcU8** which is found in our "Appendix [which] is quite comprehensive, containing rich theoretical validations." **reviewer MTWv**.

For the AC we provide a brief summary of the discussion period.

---
## List of major concerns addressed in the discussion period
1. **Advantages of Rex over other reversible solvers.** We clarify the advantages of Rex, namely, i) We create the **first** reversible solver for diffusion SDEs, ii) we can construct a $k$-th order ODE solver, iii) for the ODE solvers we have a non-zero region of stability, iv) Rex SDE uses a high-order SRK scheme based on the Foster-Reis-Strange scheme which uses Levy areas to approximate the higher-order term, v) Rex is actually the reversible version of many popular solvers for diffusion models.
2. **Experimental results.** We reran BDIA with the recommended parameters from their paper [Section 6.1, 1] ($\gamma = 1.0$) in additional to reporting a more comprehensive suite of metrics (FD, FD$_\infty$, precision, recall, density, and coverage) which are all reported in our updated **Table 1.** We also reran the conditional generation experiments and updated **Table 2** to include the Pick score metric. We updated the writing of the Section 5 to improve the discussion comparing to prior works and discussing the differences between the ODE and SDE solve.
3. **Clarity.** We redesigned **Figure 2** to more clearly show that we consider both data and noise prediction scenarios and both ODE and SDE scenarios. We also rewrote **Proposition 3.3** to more clearly show this result. Additionally we rewrote **Sections 2 and 3** to introduce Rex earlier and cut down on unnecessary bloat.
4. **Relation of Rex to other non-reversible solvers.** To address the question of how Rex relates to other non-reversible solvers we show that Rex is actually the *reversible* version of *many* widely known pre-existing solvers for diffusion models. We discuss this in our newly added **Section 4.2** and specifically **Theorem 4.3.**
5. **Errata.** Additionally, we have fixed several small minor errors, such as the omission of particular hyperparameters, *&c.* We refer to the our original responses for more details.

---

Due to the helpful suggestions from the reviewers our empirical contributions are now significantly more compelling and further supplement our core theoretical contributions.
Due to the changes in the ICLR 2026 policy no longer allowing the reviewers to provide follow-up comments to our responses we were unable to engage with a dialogue with the reviewers; however, it is our belief that we managed to address most of the primary concerns raised by the reviewers, thereby significantly improving the quality of our manuscript.

Best,
The authors

---

### Meta-Review · Area_Chair_H87H · 2025-12-17

**Summary:**

The paper proposes "Rex," a family of algebraically reversible solvers for diffusion models, encompassing both ODE and SDE formulations. The core contribution is the application of Lawson methods (exponential integrators) combined with the McCallum-Foster framework to create reversible schemes. Notably, this includes a reversible SDE solver that does not require storing the full Brownian path in memory, utilizing splittable PRNGs instead.

While the reviewers acknowledged the theoretical novelty, particularly the reversible SDE formulation and the rigorous stability analysis, the method's practical utility and empirical robustness fall short of the bar for acceptance. Besides, the critical gaps remain regarding the method's performance in high-order settings and its application to standard tasks like image editing.

**Reviewer Concerns:**

**Resolved:**
- **Reviewer 5BAq:** The authors clarified the importance of exact inversion for memory-efficient gradient backpropagation (adjoint methods) in scientific ML and guided generation, satisfying the request for concrete motivation.
- **Reviewer qLBS:** The authors rewrote Sections 2 and 3 to explicitly handle both data and noise prediction formulations, directly addressing the confusion raised regarding the derivation focusing on "data prediction" while experiments required "noise prediction."
- **Reviewer GcU8:** The authors reran experiments with correct hyperparameters for BDIA and included a wider suite of metrics, providing the fair comparison requested by the reviewer.
- **Reviewers GcU8, qLBS:** The addition of Theorem 4.3, showing that Rex essentially "reversifies" standard solvers like DDIM and SEEDS, strengthened the theoretical positioning and addressed questions about the method's relationship to existing non-reversible solvers.

**Outstanding:**
- **Reviewer qLBS:** A primary theoretical claim of the paper is the achievement of arbitrarily high-order convergence. However, experiments consistently show that high-order variants (e.g., RK4) perform worse than lower-order counterparts (Euler). The authors acknowledged this but did not resolve the contradiction.
- **Reviewers GcU8, qLBS:** In standard sampling tasks, Rex is often outperformed by baselines like O-BELM, which the authors critique as "unstable." As noted by Reviewer qLBS, if the stable/reversible method performs worse in general generation, its utility is limited strictly to specialized inversion tasks, which significantly narrows the contribution compared to what was claimed.
- **Reviewers MTWv, GcU8:** Despite explicit requests, the paper lacks evaluations on cutting-edge models (e.g., SDXL, Flux) and, crucially, lacks standard image editing/reconstruction experiments. Since image editing is the primary use-case for reversible solvers, this omission prevents a clear assessment of practical utility.
- **Reviewer qLBS:** The reviewer pointed out that the "data prediction" parameterization is pathologically unstable for inversion. While the authors pivoted to "noise prediction" for experiments, the disconnect between the primary theoretical derivation and the working practical implementation remains a fundamental weakness.

**Reviewer Scores:**

- **Reviewer 5BAq (8):** Remains positive the score  as the primary concern regarding motivation was addressed or decrease the score since key concerns from other reviewers remain unresolved.
- **Reviewer MTWv (4 $\to$ 4):** Maintained their score. While structural complaints were addressed, the critical request to validate the method on modern models (SDXL, Flux) and perform image editing tasks remains unfulfilled, limiting the assessment of practical relevance.
- **Reviewer GcU8 (4 $\to$ 4):** Maintained their score. Although the BDIA baseline was fixed, the absence of image editing and reconstruction experiments, which are standard benchmarks for reversible solvers, prevents a clear assessment of the method's superiority over existing techniques.
- **Reviewer qLBS (4 $\to$ 4):** Maintained their score. This reviewer likely found the rebuttal insufficient because it mostly "admitted" weaknesses rather than fixing them. Specifically: 1) The "high-order" theoretical advantage is negated by poor experimental performance; 2) The method lacks competitiveness in standard sampling against "unstable" baselines; and 3) The admitted instability of the data-prediction parameterization undermines the paper's primary mathematical derivation.

---

### Decision · Program_Chairs · 2026-01-26

Reject